# Reducing Belief Deviation in Reinforcement Learning for Active Reasoning of LLM Agents

**Deyu Zou[1][†], Yongqiang Chen[1][†], Jianxiang Wang[2], Haochen Yang[1], Mufei Li[3],**
**Qing Da[2], James Cheng[1][‡], Pan Li[3][‡], Yu Gong[2]**

[1]The Chinese University of Hong Kong, [2]ByteDance, [3]Georgia Institute of Technology
{dyzou24,jcheng}@cse.cuhk.edu.hk,
panli@gatech.edu, yuxiaofei@bytedance.com

## Abstract

Active reasoning requires large language model (LLM) agents to interact with external sources and strategically gather information to solve problems in multiple turns. Central to this process is *belief tracking*: maintaining an accurate representation of the underlying state and uncertainty in understanding and solving the problem. However, due to limited reasoning capabilities, LLM-based agents often suffer *belief deviation*: their internal beliefs drift from the true problem state, leading to loss of state awareness and uninformative or repetitive actions. Once this happens, errors compound in the trajectories used for reinforcement learning (RL), leading to misattributed credits and limited exploration. To address this issue, we propose to track belief deviation and develop $\mathbf{T^3}$, a simple yet principled method that detects excessive deviation and truncates training trajectories to suppress uninformative tail effects. Hence, $\mathbf{T^3}$ preserves credits for informative prefixes and systematically improves policy optimization. Across 5 challenging tasks, $\mathbf{T^3}$ consistently enhances training stability and yields performance gains of up to 30 points while cutting token cost by up to $34\%$. These results highlight *belief control* as a key principle for building robust LLM agents capable of active reasoning.[*]

## 1 Introduction

Large language models (LLMs) have demonstrated remarkable reasoning capabilities across diverse domains (Huang & Chang, 2022; Plaat et al., 2024; Li et al., 2025b), further advanced by reinforcement learning (RL) with outcome rewards (Wang et al., 2024; Srivastava & Aggarwal, 2025; Xu et al., 2025; Guo et al., 2025; OpenAI, 2025; Team et al., 2025). Recently, along with the increasing agentic applications of LLMs (Zhang et al., 2025a; Plaat et al., 2025), the community seeks to extend the success of RL to long-horizon and multi-turn reasoning (Wu et al., 2025; Laban et al., 2025; Li et al., 2025a). A key capability of LLM agents in multi-turn reasoning is *active reasoning* where the agent must *strategically* raise questions and actively acquire missing information to complete the task through multi-turn interactions with the external environment (Zhou et al., 2025; Badola et al., 2025).

However, LLM-based agents often struggle in multi-turn and active reasoning settings: as interactions unfold, they generate redundant, irrelevant, or uninformative actions (Yuan et al., 2025; Fu et al., 2025; Zhang et al., 2025b), and may even collapse into unproductive loops (Zhou et al., 2025). Moreover, RL training alone does not fully resolve these issues. Empirically, the learned policies can still yield globally suboptimal outcomes (Wang et al., 2025) or exhibit poor robustness to unseen tasks (Zhang et al., 2025b). Hence, it raises an intriguing research question:

*Why do LLM agents get trapped in active reasoning, and how can we mitigate it?*

To answer the question, we start by modeling active reasoning as a Partially Observable Markov Decision Process (POMDP). Classical POMDP formulations assume perfect belief estimate (*e.g.*,

---

[†]These authors contributed equally to this work.
[‡]Corresponding Authors.
[*]Our implementation is available at https://github.com/unimpor/T3.

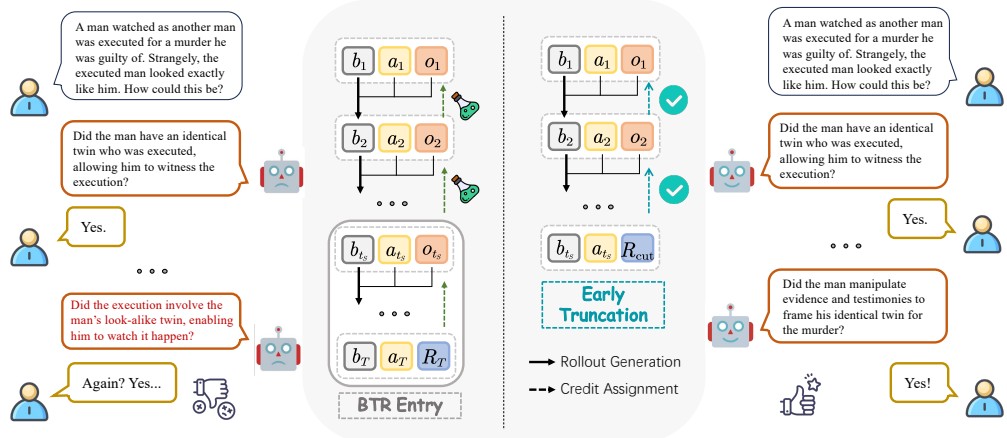

Figure 1: Overall framework of $\mathbf{T^3}$, where $(b_t, a_t, o_t)$ denote the agent's internal belief, its chosen action, and the resulting feedback at turn $t$, respectively. By truncating belief-trapped trajectories, we prevent the agent from entering the belief-trap region (BTR) where credit assignment is contaminated in RL training, allowing learning signals to concentrate on genuinely informative actions. As a result, policy optimization becomes more stable and effective under complex active reasoning.

via Bayesian filtering) conditioned on past observations (Kaelbling et al., 1998). In contrast, when instantiated with LLM agents, belief tracking must be approximated by the model itself, which is inherently imperfect due to the limited reasoning capabilities of LLMs. Under mild assumptions, we show that such imperfect belief updates can lead the rollout trajectories into a *Belief-Trap Region* (BTR, Def. 1), where actions cease to be informative, errors accumulate, and the reasoning progress stagnates (Thm. 1). Moreover, we show that standard policy optimization will be systematically misled by such belief-trap dynamics: once trapped, the uninformative tail of the trajectory can contaminate the credit assigned to crucial early-stage actions, and even *invert their estimated gradients* (Thm. 2), thereby hindering effective exploration and leading to suboptimal policies.

To mitigate belief-trap dynamics, we propose $\mathbf{T^3}$ (Truncating Belief-Trapped Trajectories), a simple yet principled method that halts trajectories upon detecting entry into the BTR. Since the exact onset of the BTR is intractable in LLM agents, we introduce the $\mathbf{T^3}$ condition (Def. 2), a theory-grounded criterion that characterizes entry into the BTR. In practice, this condition is instantiated via observable proxy signals within the reasoning trace. Empirically, we find that even simple signals, *e.g.*, redundant queries, proved to be effective indicators of belief trapping. By truncating the uninformative tail, $\mathbf{T^3}$ preserves the credit assigned to informative prefixes, resulting in lower-variance and less-biased gradient estimates (Cor. 1). Owing to its simplicity, $\mathbf{T^3}$ can be seamlessly integrated into standard policy optimization frameworks (e.g., PPO, GPRO, and GSPO) without altering the underlying algorithm, providing a practical drop-in solution to the credit assignment problem.

We evaluate $\mathbf{T^3}$ on 4 datasets and 5 tasks from recent challenging active reasoning benchmarks, including AR-Bench (Zhou et al., 2025) and Multi-Turn Puzzles (Badola et al., 2025). Across all settings, $\mathbf{T^3}$ consistently improves training stability, token efficiency, and final performance, achieving gains of up to 30 points while cutting rollout tokens by up to 34%. It further shows robust benefits across LLM sizes, architectures, and even under out-of-distribution scenarios. These results demonstrate that controlling belief traps not only systematically improves policy optimization but also provides a principled path toward building reliable active reasoning agents.

## 2 REINFORCEMENT LEARNING FOR ACTIVE REASONING

### 2.1 THEORETICAL FORMULATIONS

Due to space limits, in this section, we will state the necessary setup to derive our theoretical results and leave the details to Appendix B. To strengthen the connection between our theoretical analysis and the practical behavior of LLM-based agents, we conduct empirical studies that directly examine the key theoretical components and summarize the findings in Appendix C (an overview in Fig. 2).

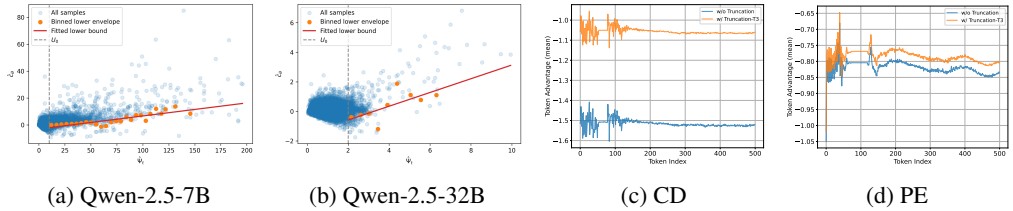

(a) Qwen-2.5-7B    (b) Qwen-2.5-32B    (c) CD    (d) PE

Figure 2: Overview of empirical verification for key theoretical components (details in Appendix C). (a)(b) Empirical lower-bound fitting for Asmp. 1. We visualize the fitted lower bound (red line) $\hat{c}_\theta \approx \hat{m}_\theta \hat{\Psi} - \hat{c}_0$, over the region $\hat{\Psi} \geq \hat{U}_0$ (vertical dashed line) for the PE task across Qwen-2.5-7B and 32B models. Both models exhibit a clear positive lower-bound slope. (c)(d) Empirical validation of advantage drift (Thm. 2). We report token-wise mean GAE values on failed rollouts for the CD and PE tasks (Qwen-2.5-7B), comparing *without* vs. *with* $\mathbf{T^3}$. In both tasks, early-token advantages display negative drift, and this drift is attenuated under $\mathbf{T^3}$, consistent with Cor. 1.

We model *active reasoning* as a Partially Observable Markov Decision Process (POMDP) (Kaelbling et al., 1998) $(\mathcal{S}, \mathcal{A}, \mathcal{O}, T, O, R, \gamma)$, where $\mathcal{S}$ is the space of latent states, $\mathcal{A}$ the action space, $\mathcal{O}$ the observation space, $T$ the transition dynamics, $O$ the observation model, $R$ the reward function, and $\gamma$ the discount factor. At each step, the agent selects an action (question) $a_t \in \mathcal{A}$ based on a belief state $b_t \in \Delta(\mathcal{S})$, *i.e.*, a distribution over latent states summarizing the interaction history. Note that the true latent state $s^\star \in \mathcal{S}$ is **unobservable** to the agent ($s^\star$ is introduced solely for theoretical analysis); for analytical clarity, we assume $s^\star$ is **fixed** within an episode. The environment returns an observation $o_t \in \mathcal{O}$ via $O(\cdot \mid s^\star, a_t)$, and the agent updates its belief to $b_{t+1}$ accordingly.

**Belief Updates.** We consider two agentic reasoners operating under the same interaction protocol but differing in their belief-update mechanisms: an *oracle* reasoner and an imperfect *LLM* reasoner. The oracle maintains an *oracle belief* $b_t^\star$ and updates it via the Bayesian operator $B^\star$:

$$b_{t+1}^\star(s) := B^\star(b_t^\star, a_t, o_t) = \frac{O(o_t \mid s, a_t)b_t^\star(s)}{p_b(o_t \mid a_t)}, \tag{1}$$

where $p_b(o_t \mid a_t) := \sum_{s' \in \mathcal{S}} O(o_t \mid s', a_t)b_t^\star(s')$ is the Bayes normalizer. In contrast, the LLM agent maintains an *LLM belief* $b_t$ (its internal estimate of the latent state) and updates it through a potentially imperfect rule $B_\theta$, where $\theta$ denotes the LLM parameters.

**Task Progress.** We analyze how the agent's belief updates influence task progress during interaction. To quantify progress, we introduce a truth-anchored potential function $\Psi(b) := -\log b(s^\star)$, which measures the negative log-belief mass assigned to the true latent state $s^\star$. We have $\Psi(b) \in [0, \infty)$, with $\Psi(b) = 0$ iff $b(s^\star) = 1$ (task completion), and smaller values indicate higher confidence in $s^\star$. For brevity, we write $\Psi_t^\star := \Psi(b_t^\star)$ and $\Psi_t := \Psi(b_t)$ when analyzing their dynamics. We then define the *belief-update discrepancy* as the expected gap in $\Psi$ after one update between the LLM update rule $B_\theta$ and the Bayesian operator $B^\star$:

$$c_\theta(b_t) := \mathbb{E}_{a_t} \mathbb{E}_{o_t} \Big[ \Psi\big(B_\theta(b_t, a_t, o_t)\big) - \Psi\big(B^\star(b_t, a_t, o_t)\big) \Big]. \tag{2}$$

Accurately modeling belief states in active reasoning requires the agent to maintain a precise estimate of the underlying problem state and the remaining uncertainty, which is inherently challenging for LLMs. To formalize imperfect belief modeling, we introduce the following assumption.

**Assumption 1** (Update-Error Growth). *There exist constants $m_\theta > 0$, $c_0 \geq 0$, and a threshold $U_0 \geq 0$ such that for all beliefs $b$ satisfying $\Psi(b) \geq U_0$, $c_\theta(b) \geq m_\theta \Psi(b) - c_0$.*

Intuitively, Assumption 1 states that belief-update errors are amplified as the deviation increases. In high-uncertainty regimes, the update error grows at least linearly with $\Psi$. Then, we have

**Theorem 1** (Informal). *Assume (i) non-degenerate observations, (ii) an $L_\pi$-Lipschitz policy w.r.t. beliefs, and (iii) Asmp. 1. Define $U := \max\{U_0, (\Psi_1^\star + \bar{B} + c_0)/m_\theta\}$ with $\bar{B} := 2(-L_\pi \log \eta + 1/\eta)$, and let $t_S := \inf\{t : \Psi_t \geq U\}$. If $t_S < \infty$, then for all $t \geq t_S$, the expected potential ceases to decrease: $\mathbb{E}[\Psi_{t+1} \mid b_t] \geq \Psi_t$. Moreover, additionally assuming $U_0 = 0$ and $\Psi_t^\star \geq \mu > 0$ for all $t < t_S$, it holds that $t_S \leq 1 + \lceil \log_{1+m_\theta} \frac{m_\theta U + \delta}{m_\theta \Delta_1 + \delta} \rceil$ for $\delta := m_\theta \mu - (c_0 + \bar{B}) > 0$ and $\Delta_1 := \Psi_1 - \Psi_1^\star$.*

A formal statement and proof are given in Appendix B.3. Intuitively, Thm. 1 indicates that, once $t_S$ is reached, the belief trajectory enters an absorbing region in which expected task progress becomes non-positive. We refer to such regions as Belief Trap Regions and define them formally below.

**Definition 1** (Belief Trap Region, BTR). *A set $\mathcal{R}_\theta \subseteq \Delta(\mathcal{S})$ is called a belief trap region for an agent parameterized by $\theta$ if it is absorbing and induces non-positive progress: for any belief $b \in \mathcal{R}_\theta$ and all subsequent times $t$ once entered, $\mathbb{E}[\Psi(b_{t+1}) \mid b_t = b] \geq \Psi(b)$.*

**Misguided credit assignment.** Within BTRs, the potential sequence $\Psi_t$ becomes non-decreasing in expectation, *i.e.*, $\mathbb{E}[\Psi_{t+1} \mid b_t] \geq \Psi_t$. Consequently, once a trajectory enters the BTR, subsequent steps contribute little task progress and reinforce the stalled dynamics. This degrades sample efficiency, as extended uninformative interactions provide limited learning signal. More critically, entry into the BTR distorts *credit assignment*: the uninformative tail of a trajectory contaminates the credit attributed to earlier exploratory actions, and may even invert their estimated advantages. This mechanism discourages exploration and leads to suboptimal policies.

We formalize this by analyzing the generalized advantage estimator (GAE) (Schulman et al., 2015), $\widehat{A}_t = \sum_{j=0}^{T-t-1} (\gamma\lambda)^j \delta_{t+j}$, where $\gamma \in (0, 1)$ is the discount factor, $\lambda \in [0, 1]$ is the GAE parameter, and the TD-error is defined as $\delta_t = r_t + \gamma V_{t+1} - V_t$ with $r_t$ the intermediate reward and $V_t$ the value function at step $t$. We consider the outcome-based RL setting, in which only the terminal step yields a non-zero reward. The following theorem characterizes how entry into the BTR can drive the expected advantage of early actions negative, thereby inverting the gradient direction.

**Theorem 2** (Informal). *Under the same setup as Thm. 1, assume (i) the value function in policy optimization satisfies $V_t = g(b_t(s^*))$ for an increasing, differentiable $g$ with $\inf_x g'(x) \geq \kappa_V > 0$, and (ii) belief drop in BTRs: there exists $\rho_b > 0$ such that $\mathbb{E}\left[b_{k+1}(s^\star) - b_k(s^\star) \mid \mathcal{F}_k\right] \leq -\rho_b$ for $k \geq t_S$. Then, for any $t < t_S$, the expected advantage is bounded: $\mathbb{E}[\widehat{A}_t] \leq \gamma\left(S_{\mathrm{pre}}(t) - \kappa_V \rho_b S_{\mathrm{tail}}^\ominus(t)\right)$, where $S_{\mathrm{pre}}(t) = \sum_{j=0}^{t_S-t-1} (\gamma\lambda)^j$ and $S_{\mathrm{tail}}^\ominus(t) = \sum_{j=t_S-t}^{T-t-2} (\gamma\lambda)^j$. Therefore, a sufficient condition for $\mathbb{E}[\widehat{A}_t] < 0$ is: $\kappa_V \rho_b > S_{\mathrm{pre}}(t)/S_{\mathrm{tail}}^\ominus(t)$. In particular, when $\gamma\lambda \to 1$ (often used in practice for long-horizon agentic RL), this reduces to $\kappa_V \rho_b > \Delta/L$, where $\Delta = t_S - t$ and $L = T - 1 - t_S$ are the prefix and tail lengths, respectively.*

A formal statement is given in Appendix B.4. Thm. 2 quantifies the credit assignment failure: a sufficiently long uninformative tail (large $L$) induces a negative drift that can dominate the positive contribution from the informative prefix, causing its overall gradient to point in the wrong direction and penalizing earlier exploratory actions. This analysis directly motivates $\mathbf{T^3}$: truncating a rollout upon entering the BTR preserves the credit assigned to informative prefix actions and eliminates the adverse effect of the uninformative tail.

**Corollary 1** (Value of Truncation). *Let $\widehat{A}_t^{\mathrm{pre}}$ denote the advantage estimator truncated at $t_S$. Under the assumptions of Thm. 2, early truncation yields a less biased gradient estimate: $\mathbb{E}[\widehat{A}_t^{\mathrm{pre}}] \geq \mathbb{E}[\widehat{A}_t] + \gamma\kappa_V \rho_b S_{\mathrm{tail}}^\ominus(t)$.*

Corollary 1 indicates that truncating the trajectory at $t_S := \inf\{t : \Psi_t \geq U\}$ removes the uninformative tail and yields a less biased policy optimization update. However, this idealized truncation rule is not directly implementable in practice for two-fold reasons. 1) *Belief modeling complexity:* the belief state $b$ is defined over the latent state space $\mathcal{S}$, which is often high-dimensional, structured, and intricate. In LLM agents, belief is not explicitly represented; instead, it is only implicitly encoded in intermediate reasoning traces or internal activation status, and thus precisely recovering their underlying belief states is infeasible in practice. 2) *Unobservable thresholds:* Although Thm. 1 provides sufficient conditions for entry into the BTR, the critical threshold $U$ and its related parameters (*e.g.*, $m_\theta, c_0, \bar{B}$) are agent-specific and cannot be directly measured.

## 2.2 FROM THEORY TO PRACTICE: PROXY SIGNALS

**Operational criterion – $\mathbf{T^3}$ condition.** We now translate the theoretical characterization of belief trapping into an operational criterion. Although the exact BTR entry time is unobservable, its defining feature, *i.e.*, *stalling of epistemic progress*, can be approximated through observable surrogates. This motivates a general truncation principle based on detecting sustained stalls of progress:

**Definition 2** ($\mathbf{T^3}$ Condition). *Let $\mathcal{H}_t$ denote the hypothesis space at step $t$. The $\mathbf{T^3}$ condition for trajectory truncation at step $t$ is defined as follows: there exists a minimum progress threshold $\Delta_{\min} \geq 0$ such that for all steps $\tau$ in the window $[t - k, t)$, $d(\mathcal{H}_\tau, \mathcal{H}_{\tau+1}) \leq \Delta_{\min}$, where $k$ is the window size and $d(\cdot, \cdot)$ is a refinement measure capturing the degree to which the hypothesis set contracts between two consecutive steps.*

$\mathbf{T^3}$ truncates the trajectory at step $t$ when the condition is satisfied. In goal-directed active reasoning tasks, the space of latent states $\mathcal{S}$ could correspond to the set of candidate solutions, where we could interpret $\mathcal{H}_t$ as the subset of states that remain plausible given the interaction history up to step $t$. Its concrete instantiation may vary across tasks and can be finite or infinite (*cf.* Sec. 3.1). In particular, for tasks with a finite and enumerable hypothesis space $\mathcal{H}_t$, if one models the agent's belief as uniform over $\mathcal{H}_t$ (assuming $s^\star \in \mathcal{H}_t$), then the identity $\Psi(b_t) = \log |\mathcal{H}_t|$ follows, which provides an exact observable surrogate for the potential dynamics in this setting.

**Relation to the BTR formalism.** Conceptually, the $\mathbf{T^3}$ principle is structurally aligned with the BTR formalism: BTRs are characterized by stalled progress in the potential function, *i.e.*, $\mathbb{E}[\Delta\Psi_t] \geq 0$. In goal-directed reasoning tasks, such stagnation typically manifests as a persistent lack of contraction in the hypothesis spaces. Def. 2 formalizes this point by introducing: *1)* a measure $d(\mathcal{H}_t, \mathcal{H}_{t+1})$ to quantify incremental contraction of the hypothesis representation; *2)* a threshold $\Delta_{\min}$ to capture the notion of a minimally informative update; and *3)* a window of length $k$ that enforces temporal persistence, reflecting that BTRs arise from sustained stalls rather than a single noisy fluctuation.

To further quantify this alignment, the following proposition establishes a guarantee under a standard *biased noisy* model, linking $\mathbf{T^3}$ ingredients to an upper bound on false-truncation probability.

**Proposition 1.** *Let the true single-step potential progress be $g_t := \Psi(b_t) - \Psi(b_{t+1})$ and define the observable refinement signal $d_t := d(\mathcal{H}_t, \mathcal{H}_{t+1})$. Assume that (i) outside the BTR, single-step potential progress admits a uniform positive margin: $g_t \geq \rho > 0$, and (ii) the proxy follows a biased Gaussian-noise model: $d_t = g_t + \beta_t + \xi_t$, where $|\beta_t| \leq M_d$, $\xi_t \sim \mathcal{N}(0, \sigma^2)$ are independent across $t$. If $\Delta_{\min} < \rho - M_d$, then a sufficient condition for the $\mathbf{T^3}$ rule to keep the false-truncation probability on any $k$-step non-BTR segment below $\delta \in (0, 1)$ is $k(\rho - M_d - \Delta_{\min})^2 \geq 2\sigma^2 \log(1/\delta)$.*

A proof is given in Appendix B.9. This proposition shows that, even under both systematic bias and stochastic noise in the proxy, the $\mathbf{T^3}$ rule remains statistically robust. In particular, the choice of $\mathcal{H}$ and metric $d(\cdot, \cdot)$ determines the bias bound $M_d$. Reducing this bias, increasing $k$, or decreasing $\Delta_{\min}$ reduces the probability of false truncation at an exponential rate. We additionally present an analysis on the effect of false-truncation in Appendix C.3.

**Practical instantiation and toward general-purpose detectors.** In practice, since the structure of hypothesis spaces and notions of progress differ across tasks, constructing $\mathcal{H}_t$ and $d(\cdot, \cdot)$ naturally leverages *task-level structure* to define observable proxies that track epistemic progress. We show how to instantiate it for practical tasks in Sec. 3.1. Moreover, guided by $\mathbf{T^3}$, we can further reduce the reliance on task-specific structures by designing *general-purpose* truncation detectors. We conduct preliminary explorations and find that these surrogates can be incorporated into $\mathbf{T^3}$ while still yielding improvements across multiple tasks. Details and discussion are provided in Appendix E.1.

**Key advantages.** This principle functions as a *meta-wrapper*: it provides structured guidance for designing effective proxy signals grounded in progress-based criteria that capture the essence of belief-trap dynamics, rather than relying on complex heuristics or heavy engineering. Importantly, the resulting truncation rules integrate seamlessly into standard policy optimization frameworks (e.g., PPO, GRPO, GSPO) without altering their algorithms, making $\mathbf{T^3}$ a practical drop-in solution for mitigating credit assignment distortion in active reasoning.

## 3 EXPERIMENTS

### 3.1 TASK-SPECIFIC INSTANTIATIONS OF THE $\mathbf{T^3}$ CRITERION

We evaluate $\mathbf{T^3}$ on five interactive reasoning tasks from AR-Bench (Zhou et al., 2025) and Multi-Turn Puzzles (Badola et al., 2025). The $\mathbf{T^3}$ criterion (Def. 2) provides a task-agnostic principle. In practice, its components ($\mathcal{H}$, $d$, etc.) are instantiated using observable proxies tailored to each task. Note that we do adaptations to some of these datasets for RL training. See more details in Appendix F.1.

**GuessNumbers (GN).** The agent aims to identify a hidden number through iterative guesses, receiving structured feedback that indicates the number of digits in the correct position or misplaced. The hypothesis space $\mathcal{H}_t$ consists of all candidate numbers consistent with the interaction history $\{a_{\leq t}, o_{\leq t}\}$. We naturally define the refinement metric as $d(\mathcal{H}_\tau, \mathcal{H}_{\tau+1}) := |\mathcal{H}_\tau| - |\mathcal{H}_{\tau+1}|$, which directly measures reduction in the candidate set. *Early truncation:* a trajectory is cut at step $t$ if the agent's guess $a_t$ lies outside $\mathcal{H}_{t-1}$, corresponding to the case $k = 1$ where we treat $d(\mathcal{H}_{t-1}, \mathcal{H}_t) \leq 0$.

Table 1: Main results across active reasoning tasks (all metrics are scaled by 100). ↑ indicates absolute improvement (in points) over the vanilla RL baseline. We report the average rank across all metrics.

| | CD | SP | | GN | PE | MR | Avg. |
|---|---|---|---|---|---|---|---|
| | EM | F1-word | F1-char | EM | Binary Sim | EM | Rank |
| **Direct Inference** | | | | | | | |
| o3-mini | 92.67 | 20.64 | 39.35 | 95.28 | 44.67 | 83.33 | 4.67 |
| Gemini-2.5-Pro | 92.23 | 24.12 | 49.28 | 90.84 | 16.67 | 83.00 | 5.67 |
| Qwen-2.5-7B-Inst. | 12.50 | 19.46 | 41.62 | 20.94 | 23.67 | 27.67 | 8.17 |
| **Reinforcement Learning** | | | | | | | |
| PPO | 61.67 | 28.77 | 74.56 | 91.62 | 42.00 | 24.33 | 6.50 |
| PPO w/ $\mathbf{T^3}$ | 77.83 ↑16.2 | 36.85 ↑8.1 | 81.50 ↑6.9 | 93.98 ↑2.4 | 49.00 ↑7.0 | 38.00 ↑13.6 | 4.50 |
| GRPO | 79.33 | 36.46 | 83.73 | 61.26 | 51.67 | 12.00 | 5.50 |
| GRPO w/ $\mathbf{T^3}$ | 81.33 ↑2.0 | 39.45 ↑3.0 | 84.58 ↑0.8 | 91.36 ↑30.1 | 52.33 ↑0.7 | 32.67 ↑20.7 | 3.17 |
| GSPO | 77.67 | 36.63 | 82.17 | 96.07 | 59.00 | 14.67 | 4.33 |
| GSPO w/ $\mathbf{T^3}$ | 81.00 ↑3.3 | 36.96 ↑0.3 | 82.08 ↓0.1 | 99.74 ↑3.7 | 62.00 ↑3.0 | 55.67 ↑41.0 | 2.50 |

Such guesses violate the logical constraints accumulated from previous observations and reflect a failure to correctly track the candidate set.

**SituationPuzzles (SP).** The agent resolves a paradoxical puzzle by posing yes/no questions to a judge model. Here $\mathcal{H}_t$ denotes the set of plausible explanations consistent with the dialogue history. Since $\mathcal{H}_t$ can be complex or unbounded, we approximate stalled refinement using judge feedback: a step is considered uninformative if the judge responds with "unknown," which serves as a proxy for $d(\mathcal{H}_\tau, \mathcal{H}_{\tau+1}) < \Delta_{\min}$. *Early truncation:* if this occurs for $k = 5$ consecutive steps, we truncate the trajectory, signaling entrapment in an unproductive line of questioning. We employ an LLM-based judge proxy in the main experiments and additionally evaluate a judge-free proxy in Sec. 3.3.3.

**CircuitDecoding (CD).** The agent identifies hidden boolean circuits from a large candidate pool. At each step, the agent queries a candidate circuit with a binary input and eliminates inconsistent candidates based on the feedback. The hypothesis space $\mathcal{H}_t$ consists of all surviving candidates consistent with the interaction history, and we define the refinement metric analogously to GN: $d(\mathcal{H}_\tau, \mathcal{H}_{\tau+1}) := |\mathcal{H}_\tau| - |\mathcal{H}_{\tau+1}|$. *Early truncation:* we monitor $|\mathcal{H}_t|$ and truncate if it fails to contract, *i.e.*, $d(\mathcal{H}_\tau, \mathcal{H}_{\tau+1}) \leq 0$, for $k = 3$ turns, indicating that queries no longer reduce uncertainty.

**PreferenceEstimation (PE) / MovieRecommendation (MR).** In PE, the agent infers a hidden vector $v^\star$ about user preference on movies by iteratively raising pairwise comparisons over the given reference movies. In MR, the agent recommends unseen movies to the user based on the inferred preference vector, requiring generalization beyond the training distribution. Here $\mathcal{H}_t$ corresponds to the subspace of plausible preference vectors consistent with past feedback. Since this space is continuous and not explicitly enumerable, we approximate its epistemic refinement progress via the LLM's explicit estimate $v_t$. Concretely, we prompt the agent to report its current estimate $v_t$ in a fixed format at each turn. *Early truncation:* we approximate the refinement signal $d(\mathcal{H}_\tau, \mathcal{H}_{\tau+1})$ by the change in similarity between the agent's estimate and the ground-truth preference, *i.e.*, $\text{Sim}(v_{\tau+1}, v^\star) - \text{Sim}(v_\tau, v^\star)$. If similarity decreases for $k = 2$ consecutive steps, the trajectory is truncated, reflecting persistent divergence in the inferred preference representation. As the proxy depends on access to the ground-truth preference $v^\star$ during training, we also explore alternative proxies that do not require ground-truth information and demonstrate the promise of $\mathbf{T^3}$ in Appendix D.3.

## 3.2 EXPERIMENTAL SETUP

**Baselines.** To evaluate the effectiveness of $\mathbf{T^3}$, we compare it against the following baselines: 1) Direct Inference without Training, where we evaluate representative proprietary reasoning LLMs, including o3-mini and Gemini-2.5-Pro; 2) PPO (Schulman et al., 2017); 3) GRPO (Shao et al., 2024); and 4) GSPO (Zheng et al., 2025). See more details of the adopted RL algorithms in Appendix F.2.

**Implementation Details.** The main experiments of RL training are conducted on Qwen2.5-7B-Instruct (Yang et al., 2024). Analyses on other architecture scales and types can be seen in Sec. 3.3.4. For the GN, CD, PE, and MR tasks, the interactive feedback is rule-based; for the SP dataset, a Qwen2.5-14B-Instruct model simulates the "user" and provides the interactive feedback. See more implementation details in Appendix F.3.

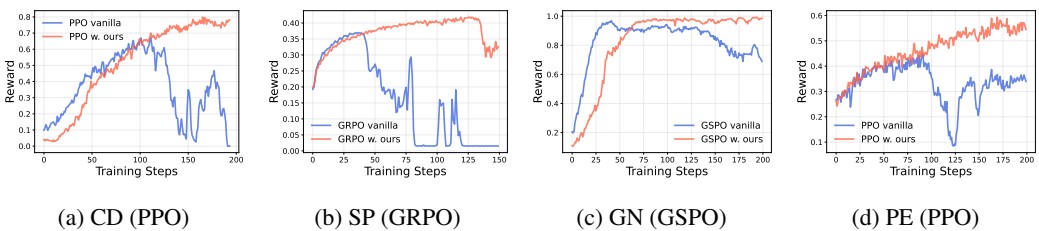

Figure 3: Training dynamics of rewards *w.r.t.* training steps.

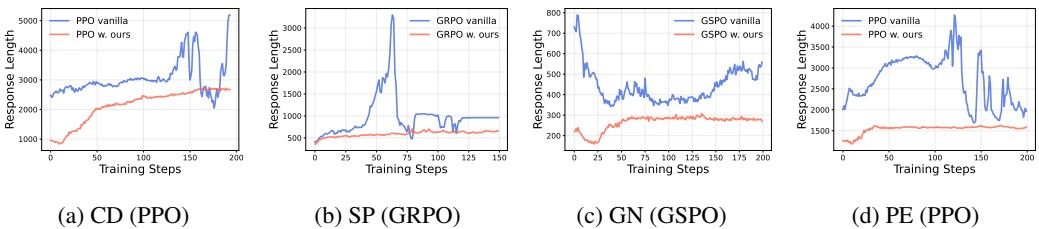

Figure 4: Training dynamics of response length *w.r.t.* training steps.

**Evaluation Metrics.** For the GN, CD, and MR tasks, we report *Exact Match* (EM), which measures whether the final prediction made by the LLM exactly matches the hidden number, ground-truth circuit, or the correct movie recommendation. For the SP task, we use the *F1* score (both word-level and character-level) to assess the similarity between the ground-truth explanation and the solution produced by the LLM. For PE, we report *Binary Similarity*, which compares the LLM-estimated vector against the ground-truth preference vector using cosine similarity. Specifically, we threshold the cosine score at $0.88$: values above the threshold are labeled as $1$, and values below as $0$. In Appendix D.1, we also explore the sensitivity with other thresholds.

### 3.3 EXPERIMENTAL RESULTS AND ANALYSES

In this part, we first present overall performance, followed by analyses of $\mathbf{T^3}$ on out-of-distribution generalization, ablation studies of truncation conditions, and the impact of LLM architectures.

#### 3.3.1 OVERALL PERFORMANCE

**Overall Performance.** The main experimental results are summarized in Table 1. Across tasks, all RL-trained agents, both with and without $\mathbf{T^3}$, substantially outperform the zero-shot baseline, confirming the necessity of RL in incentivizing active-reasoning capabilities. Compared to vanilla RL methods, incorporating $\mathbf{T^3}$ consistently improves final performance across datasets and algorithms, with non-marginal gains observed in 14 out of 18 reported metrics. On CD, PPO+$\mathbf{T^3}$ boosts EM by 16.2 points and GRPO+$\mathbf{T^3}$ yields further gains. On SP, GRPO+$\mathbf{T^3}$ achieves the best F1-word and F1-char scores. On GN, $\mathbf{T^3}$ leads to substantial improvements, raising GRPO by 30.1 points and enabling GSPO to reach a near-perfect 99.74 EM. In PE and MR, $\mathbf{T^3}$ also brings steady gains, with GSPO+$\mathbf{T^3}$ improving movie recommendation accuracy by 41.0 points. Overall, these results indicate that $\mathbf{T^3}$ provides consistent benefits across diverse active reasoning tasks.

**Comparing to frontier reasoning models.** We can also find that advanced reasoning LLMs perform strongly on tasks where the hypothesis space $\mathcal{H}$ is finite and enumerable (*e.g.*, GN and CD). However, their performance degrades on tasks with large, infinite, or continuous hypothesis spaces (*e.g.*, SP and PE), where they lag behind RL-trained Qwen-7B models equipped with $\mathbf{T^3}$. These observations suggest that large-scale RL with outcome reward training alone may be insufficient for effective active reasoning over unbounded hypothesis spaces, and that mechanisms explicitly addressing credit assignment, *e.g.*, $\mathbf{T^3}$, could provide complementary benefits.

**Better Training Stability and Optimization Behavior.** Beyond final performance, $\mathbf{T^3}$ improves training dynamics. As shown in Fig. 3, vanilla RL methods exhibit higher variance and instability, with rewards prone to collapsing after partial convergence. By contrast, incorporating $\mathbf{T^3}$ leads to more stable training trajectories, with largely monotonic or near-monotonic reward improvement and much fewer abrupt drops, and enables better optimization behavior. These results suggest the dual benefit of $\mathbf{T^3}$: stabilizing optimization while encouraging more informative exploration.

**Higher Token Efficiency**. Although the reward curves *wrt.* training steps (Fig. 3) suggest slightly slower reward growth in the early stage when incorporating $\mathbf{T^3}$, early truncation reduces the average number of tokens per rollout (*cf.*, Fig. 4). As a result, when measured against token consumption, $\mathbf{T^3}$ achieves higher training efficiency. For example, under PPO on CD, to reach a reward level of 0.65, our method consumes 66.4% of the total tokens compared to vanilla on average; under GSPO on GN, to reach 0.96, it requires 76.3% of the tokens. More importantly, while vanilla methods often stagnate and fail to improve further, incorporating $\mathbf{T^3}$ continues to enhance rewards, achieving up to 0.8 on CD and 0.99 on GN.

### 3.3.2 OUT-OF-DISTRIBUTION ANALYSIS

To better understand whether the agents learn the generalizable policies for active reasoning, we further evaluate $\mathbf{T^3}$ under distribution shifts in two representative tasks: CircuitDecoding (CD) and Preference Estimation (PE). In CD, we vary two key factors relative to training: the number of hidden circuits (training uses 2, we test up to 4) and the candidate pool size (training uses 10, we test up to 30). In PE, we vary the number of reference movies (training uses 10, we test 5-30) and the sampling distribution of their scores (training uses uniform, we test skewed side distributions).

The results are given in Table 2. Across all OOD settings, $\mathbf{T^3}$ consistently improves over vanilla PPO. In CD, although accuracy drops as the task becomes harder with larger candidate pools or more hidden circuits, the gains from $\mathbf{T^3}$ remain substantial, reaching ↑ 10.8 points with 25 candidates and ↑ 15.0 points with 3 circuits. In PE, performance varies non-monotonically with the reference size, where moderate contexts (*e.g.*, $S = 20$) achieve the best results (↑ 12.7 points). We conjecture that too few references increase the ambiguity of preference estimation, while too many may introduce noise and redundancy, which may in turn exacerbate belief-trap dynamics. See Appendix D.2 for an empirical evidence. Similarly, for reference sampling, $\mathbf{T^3}$ yields improvements across all conditions, with the largest margin observed under max-skewed sampling. Overall, these results show that $\mathbf{T^3}$ consistently enhances OOD robustness across diverse settings, even in more challenging regimes where the distribution deviates substantially from the training.

Table 2: Evaluations of $\mathbf{T^3}$ on out-of-distribution (OOD) scenarios of PE (Qwen-2.5-7B-Inst.) and CD (Qwen-2.5-14B-Inst.) tasks under the PPO algorithm.

| | PE (PPO) | | | CD (PPO) | |
|---|---|---|---|---|---|
| | Vanilla | w/ $\mathbf{T^3}$ | | Vanilla | w/ $\mathbf{T^3}$ |
| Reference Size ($S$) | | | Candidate Size ($S$) | | |
| $S = 5$ | 40.0 | 44.3 ↑ 4.3 | $S = 10$ | 67.8 | 86.3 ↑ 18.5 |
| $S = 10$ | 42.0 | 49.0 ↑ 7.0 | $S = 15$ | 61.7 | 74.7 ↑ 13.0 |
| $S = 15$ | 39.3 | 47.0 ↑ 7.7 | $S = 20$ | 48.2 | 55.8 ↑ 7.7 |
| $S = 20$ | 41.0 | 53.7 ↑ 12.7 | $S = 25$ | 35.2 | 46.0 ↑ 10.8 |
| $S = 30$ | 42.3 | 46.3 ↑ 4.0 | $S = 30$ | 31.5 | 35.7 ↑ 4.2 |
| Reference Sampling | | | Hidden Circuit Size ($C$) | | |
| min-max | 45.7 | 56.0 ↑ 10.3 | $C = 2$ | 67.8 | 86.3 ↑ 18.5 |
| uniform | 42.0 | 49.0 ↑ 7.0 | $C = 3$ | 60.3 | 75.3 ↑ 15.0 |
| max | 50.7 | 61.3 ↑ 10.7 | $C = 4$ | 42.7 | 49.3 ↑ 6.6 |

### 3.3.3 ABLATION STUDY ON TRUNCATION CONDITIONS

The effectiveness of $\mathbf{T^3}$ depends on the design of the proxy signal for truncating the BTR tail. We therefore conduct ablation studies to examine the robustness of different truncation conditions and their associated trade-offs. First, we vary the window size $k$ to evaluate the effect of temporal persistence in detecting stalls. Furthermore, we consider alternative truncation strategies. For the SP task, we evaluate *Question Semantic Similarity (Sim-$\alpha$)*: a trajectory is truncated if the cosine similarity between the embedding of the current query and any previous query exceeds a threshold $\alpha$, where we leverage the `E5-large-v2` model (Wang et al., 2022) to calculate embeddings. This proxy detects redundant or circular questioning, and we evaluate $\alpha \in \{0.9, 0.93, 0.96\}$. For the CD and PE tasks, we include a *random truncation (Rand-$\beta$)* baseline, where each step is truncated independently with probability $\beta$. We test $\beta \in \{0.1, 0.2, 0.5\}$ for CD and $\{0.2, 0.5, 0.8\}$ for PE.

Table 3: Ablation Study of Truncation Conditions on the SP, CD, and PE tasks. Beyond the window size $k$ as seen in Def. 2, we consider alternative truncation methods, described in $\alpha$ and $\beta$.

| SP (GRPO) | | CD (PPO) | | PE (PPO) | |
|---|---|---|---|---|---|
| Method | F1-word | Method | EM | Method | Binary Sim |
| Vanilla | 36.46 | Vanilla | 61.67 | Vanilla | 42.00 |
| $k = 3$ | 38.62 ↑ 2.16 | $k = 2$ | 69.17 ↑ 7.50 | $k = 2$ | 49.00 ↑ 7.00 |
| $k = 5$ | 39.45 ↑ 2.99 | $k = 3$ | 77.83 ↑ 16.2 | $k = 4$ | 44.33 ↑ 2.33 |
| $k = 9$ | 36.96 ↓ 0.50 | $k = 4$ | 79.33 ↑ 17.6 | $k = 7$ | 42.00 ↑ 0.00 |
| $\alpha = 0.9$ | 39.44 ↑ 2.98 | $\beta = 0.1$ | 69.00 ↑ 7.33 | $\beta = 0.2$ | 43.33 ↑ 1.33 |
| $\alpha = 0.93$ | 38.81 ↑ 2.35 | $\beta = 0.2$ | 57.50 ↓ 4.17 | $\beta = 0.5$ | 44.67 ↑ 2.67 |
| $\alpha = 0.96$ | 37.93 ↑ 1.47 | $\beta = 0.5$ | 13.17 ↓ 48.5 | $\beta = 0.8$ | 39.00 ↓ 3.00 |

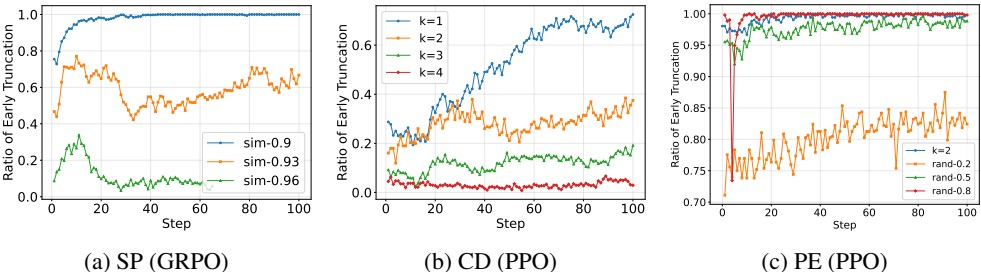

(a) SP (GRPO)    (b) CD (PPO)    (c) PE (PPO)

Figure 5: Training dynamics of the ratio of early truncation *w.r.t.* training steps under different truncation conditions for the SP (a), CD (b), and PE (c) tasks.

The results are reported in Table 3. For SP, increasing $k$ improves performance up to around $k = 5$, after which the gains diminish. The similarity-based proxy also improves over vanilla GRPO, suggesting that $\mathbf{T^3}$ is robust to different proxy formulations as long as they can detect the BTR entry reasonably. For CD, varying $k$ shows stable improvements, with $k = 3, 4$ yielding the largest gains over vanilla PPO. We further observe that even random truncation can produce a mild improvement when the ratio $\beta$ is appropriately chosen. This suggests the significance of the BTR issue: mitigating long uninformative tails, even via simple truncation heuristics, partially improves optimization quality. For PE, $k = 2$ achieves the best performance, while the gains diminish as the condition becomes looser. Overall, these results indicate that the proxy condition should be calibrated at a moderate level. If it is too loose (*e.g.*, $k = 9$ for SP), truncation has limited effect, causing belief-tracking errors to accumulate. If it is too strict (e.g., $\beta = 0.2, 0.5$ for CD), it may terminate trajectories prematurely, suppressing early-stage exploratory actions and reducing the effective learning signal.

**Training Dynamics of Early Truncation.** Furthermore, we examine the temporal evolution of the early-truncation frequency during training, as shown in Fig. 5. For clarity, the truncation ratio at training step $t$ is defined as $\text{ratio}_t = \frac{\text{\# rollouts truncated at step } t}{\text{\# total rollouts at step } t}$. This quantity tracks how frequently trajectories satisfy the truncation condition throughout optimization. Combining these dynamics with the final performance (Table 3) yields a clear pattern. For tasks where the hypothesis space $\mathcal{H}$ is *unbounded* (SP and PE), stronger performance is associated with relatively *high and stable* truncation ratios from early training steps. For example, in SP, the query-similarity proxy with $\alpha = 0.9$ quickly reaches near $1.0$ and achieves the best F1; in PE, $k = 2$ likewise achieves both a higher truncation ratio and the highest performance. These observations suggest that, in unbounded hypothesis spaces, promptly removing uninformative tails could contribute to improved training performance. Notably, in PE, random truncation with $\beta = 0.5, 0.8$ yields truncation ratios comparable to $k = 2$ but leads to inferior final performance. This underscores the importance of proper truncation condition design: it should meaningfully approximate the BTR entry rather than cut indiscriminately.

By contrast, for tasks with *finite and enumerable* spaces (CD), a *low-to-moderate* truncation ratio would be preferable: settings such as $k = 3, 4$ maintain low truncation frequencies throughout training and yield the largest EM gains; more aggressive truncation (e.g., $k = 1, 2$) increases the truncation ratio and is associated with reduced performance, consistent with premature termination of potentially informative trajectories. In summary, these dynamics suggest that, given a properly designed truncation condition, the appropriate truncation intensity depends on the structural properties of the hypothesis space and the task: relatively aggressive truncation could be beneficial in unbounded settings, while moderate truncation would be preferable in finite settings.

### 3.3.4 IMPACT OF LLM ARCHITECTURE

We further evaluate $\mathbf{T^3}$ across different LLM scales and architectures, including Qwen-2.5 (3B, 7B, and 14B) and multiple variants of LLaMA-3.1-8B. As shown in Fig. 6a and 6b, across Qwen-2.5 3B, 7B, and 14B, we observe that the 3B model shows only limited improvements, whereas the 7B and 14B variants achieve clear gains under RL. Moreover, larger models tend to benefit more substantially from $\mathbf{T^3}$ compared to the 3B variant. One possible explanation, consistent with our formulation in Sec. 2, is that weaker belief-tracking abilities may correspond to a larger update-error growth (*i.e.*, larger $m_\theta$, *cf.*, Asmp. 1), making smaller models more prone to quickly falling into BTRs, where even truncation cannot provide sufficient informative training signals.

A similar pattern holds across architecture types. As shown in Fig. 6c, we compare the effectiveness of $\mathbf{T^3}$ across LLaMA-3.1-8B-Instruct, Qwen-2.5-7B-Instruct, and DeepSeek-R1-Distill-LLaMA-8B.

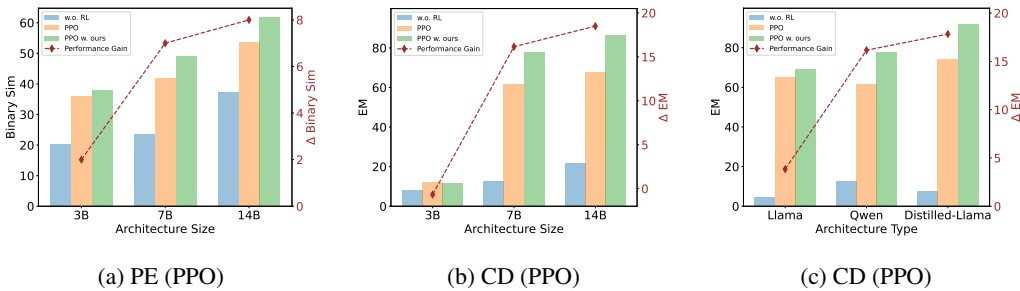

(a) PE (PPO)          (b) CD (PPO)          (c) CD (PPO)

Figure 6: Effectiveness of $\mathbf{T^3}$ on different sizes (a, b) and types (c) of LLM architectures. The "Performance Gain" denotes the improvement of $\mathbf{T^3}$ compared to the vanilla RL method.

We observe that LLaMA-8B-Instruct improves only marginally under $\mathbf{T^3}$, while its DeepSeek-distilled variant and Qwen-7B benefit more substantially. This echoes recent reports that Qwen exhibits stronger reasoning behaviors than LLaMA (Gandhi et al., 2025). Such differences may extend to belief-tracking abilities under partial observability. Notably, the distilled LLaMA variant with $\mathbf{T^3}$-equipped RL achieves the best overall performance, exhibiting the largest performance gains. We conjecture that distillation may improve the belief-tracking related capabilities, thereby enhancing the utility of $\mathbf{T^3}$ in preserving credit assignment. In our formulation, both scale- and architecture-dependent differences may be interpreted through variations in belief-tracking abilities and the associated $m_\theta$, which governs how easily trajectories get trapped in the BTR.

## 4 RELATED WORK

**Active Reasoning** requires LLMs to interact with external sources and actively acquire missing information to solve complex tasks. Prior work has improved LLMs' ability to handle ambiguity and incompleteness through making clarification and information-seeking actions. For example, Proactive CoT (Deng et al., 2023) prompts LLMs to identify ambiguous problems and generate clarification questions, while UoT (Hu et al., 2024) quantifies the contribution of each question in reducing uncertainty. However, challenges remain when transitioning from LLMs' single-turn success to multi-turn active reasoning (Kwan et al., 2024; Liang et al., 2024; Badola et al., 2025), even with several advanced strategies such as tree-based searching or post-training approaches, as highlighted in existing works (Zhou et al., 2025). In contrast, we leverage RL to incentivize active reasoning capabilities, and propose $\mathbf{T^3}$ to address key issues when applying RL in this setting.

**Credit Assignment and Multi-turn RL.** Credit assignment is crucial to long-horizon or multi-turn RL. Existing methods have extensively explored rule-based approaches (Yu et al., 2024; Dou et al., 2024; Zhang et al., 2025b) to shape intermediate rewards. Several recent works also proposed to measure the progress of stepwise actions toward overall task completion as intermediate rewards. Specifically, CURIO (Wan et al., 2025) constructs a potential function over an ideal belief state to assign intermediate rewards, assuming that the latent state space is finite and enumerable. Sotopia-RL (Yu et al., 2025) relies on reward labeling with proprietary LLMs. SPA-RL (Wang et al., 2025) trains reward models for intermediate rewards by enforcing a summation constraint with respect to the final outcome reward. In our studied active reasoning scenario, belief deviation under partial observability makes it difficult for outcome-based rewards to properly assign credit to key reasoning steps. Our proposed $\mathbf{T^3}$ mitigates this by halting the trajectory before the reasoning process becomes trapped in excessive belief deviation and the error accumulation overwhelms credit assignment.

## 5 CONCLUSION

In this work, we identify belief deviation and entry into the belief-trap region as a critical failure mode underlying instability and sub-optimality in RL for LLM-based active reasoning. To mitigate its harmful accumulation, we proposed $\mathbf{T^3}$, an early-truncation mechanism that halts belief-trapped trajectories. Empirical results on five active-reasoning tasks show that $\mathbf{T^3}$ consistently improves both training stability and final performance across multiple RL algorithms. Overall, our findings highlight belief deviation as a central bottleneck and show that controlling it provides a principled pathway toward building robust and generalizable active reasoning agents.

ACKNOWLEDGMENTS

We thank the reviewers for their constructive comments and suggestions. Deyu Zou, Yongqiang Chen, Haochen Yang, and James Cheng were supported by a CRF (No. C2005-24Y) from the RGC of Hong Kong. This work was a collaboration between Husky Data Lab at CUHK and ByteDance, supported by a ByteDance University Collaboration Project Grant.

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

CONTENTS

## LLM USAGE DISCLOSURE

In our work, we mainly use GPT-5 for writing enhancements, primarily to improve grammar and text clarity.

## REPRODUCIBILITY STATEMENT

We describe our dataset details in Appendix F.1. For additional training details, see Sec. 3.2 and Appendix F.3. For prompt templates, see Figures 10 to 15.

## A  NOTATION SUMMARY

| Symbol | Meaning | Domain / Notes |
|---|---|---|
| **Spaces, states, dynamics** | | |
| $\mathcal{S}, \mathcal{A}, \mathcal{O}$ | Latent state, action, observation spaces | Sets |
| $s^\star$ | Episode-wise fixed true latent state | $s^\star \in \mathcal{S}$ |
| $T(s' \mid s, a)$ | Transition function | Degenerate in our setting |
| $O(o \mid s, a)$ | Observation model | Assump. B.2; $O \geq \eta$ |
| $R, \gamma$ | Reward; discount factor | $\gamma \in (0, 1]$ |
| **Beliefs and policies** | | |
| $\Delta(\mathcal{S})$ | Probability simplex over $\mathcal{S}$ | Set |
| $b_t^\star, \ b_t$ | Oracle belief; agent belief at time $t$ | $b_t^\star, \ b_t \in \Delta(\mathcal{S})$ |
| $B^\star(b, a, o)$ | Oracle Bayesian belief update | Posterior under $O$ |
| $B_\theta(b, a, o)$ | Agent belief update | Parametrized by $\theta$ |
| $\pi(\cdot \mid b)$ | Belief-conditioned policy | Distribution on $\mathcal{A}$ |
| **Distances and potentials** | | |
| $d(b, b')$ | $\ell_1$ distance on beliefs | $d(b, b') = \sum_s \lvert b(s) - b'(s) \rvert \in [0, 2]$ |
| $\mathrm{TV}(P, Q)$ | Total variation distance | Probability measures; Assump. B.3 |
| $\Psi(b)$ | Truth-anchored potential | $\Psi(b) = -\log b(s^\star) \in [0, \infty)$; Def. B.1 |

| Symbol | Meaning | Domain / Notes |
|---|---|---|
| $\Psi_t,\ \Psi_t^\star$ | $\Psi(b_t);\ \Psi(b_t^\star)$ | Scalars |
| **Progress quantities** | | |
| $\mathcal{I}(b,a)$ | One-step oracle informativeness | Def. B.2 |
| $\mathcal{P}_\theta(b)$ | Agent expected one-step progress | Def. B.3 |
| $c_\theta(b)$ | Agent-Bayes update error | Def. B.4; Assump. B.4 |
| **Belief Trap Region (BTR)** | | |
| $\mathcal{R}_\theta$ | Belief trap region | Def. B.5; |
| $t_S$ | First time of reaching the BTR sufficient condition | Prop. B.1 and B.2 |
| **RL / GAE quantities** | | |
| $V_t := V(b_t)$ | Value function | $V_t = g(b_t(s^\star))$; Thm. B.1 |
| $\delta_t$ | TD-error | $\delta_t = r_t + \gamma V_{t+1} - V_t$ |
| $\lambda$ | GAE parameter | $\lambda \in (0,1]$ |
| $\widehat{A}_t$ | GAE estimator | $\widehat{A}_t = \sum_j (\gamma\lambda)^j \delta_{t+j}$ |
| **Model and technical constants** | | |
| $\eta$ | Observation non-degeneracy bound | $\eta \in (0,1]$; Assump. B.2 |
| $L_\pi$ | Policy sensitivity constant | Assump. B.3 |
| $m_\theta,\ c_0,\ U_0$ | Update-error growth parameters | $c_\theta(b) \geq m_\theta\Psi(b) - c_0$; Assump. B.4 |
| $\bar{B}$ | Technical constant | $\bar{B} = 2(-\log\eta \cdot L_\pi + 1/\eta)$; Prop. B.1 |
| $U$ | Sufficient BTR threshold | Prop. B.1 |
| $\Delta_1$ | Initial gap | $\Delta_1 = \Psi(b_1) - \Psi(b_1^\star)$; Prop. B.2 |
| $\mu$ | Oracle lower bound before $t_S$ | Prop. B.2 |
| $\delta$ | Technical constant | $\delta = m_\theta\mu - (c_0 + \bar{B})$; Prop. B.2 |
| **Auxiliary weights** | | |
| $S_{\text{pre}}(t)$ | Geometric prefix weight | $\sum_{j=0}^{t_S-t-1}(\gamma\lambda)^j$; Thm. B.1 |
| $S_{\text{tail}}^{\ominus}(t)$ | Geometric tail weight | $\sum_{j=t_S-t}^{T-t-2}(\gamma\lambda)^j$; Thm. B.1 |

# B MORE DETAILS ON THE THEORY

## B.1 DETAILED THEORETICAL SETUP

**Problem Formulation** We consider the *active reasoning* where an LLM agent interacts with an external environment to acquire missing information and infer the solution via a sequence of actions and observations (Zhou et al., 2025). This can be modeled as a Partially Observable Markov Decision Process (POMDP), defined by the tuple $(\mathcal{S}, \mathcal{A}, \mathcal{O}, T, O, R, \gamma)$, where $\mathcal{S}$ is the space of unobservable latent states, $\mathcal{A}$ the action space, $\mathcal{O}$ the observation space, $T$ the transition dynamics, $O$ the observation model, $R$ the reward function, and $\gamma$ the discount factor. At each step, the agent selects an action (question) $a_t \in \mathcal{A}$ based on a belief state $b_t \in \Delta(\mathcal{S})$, *i.e.*, a distribution over latent states summarizing the interaction history. Note that the true latent state $s^\star \in \mathcal{S}$ is **unobservable** to the agent ($s^\star$ is introduced solely for theoretical analysis); for analytical clarity, we assume $s^\star$ is **fixed** within an episode. The environment returns an observation $o_t \in \mathcal{O}$ via $O(\cdot \mid s^\star, a_t)$, and the agent updates its belief to $b_{t+1}$ accordingly.

We consider two agentic reasoners operating under the same interaction protocol but differing in their belief-update mechanisms: an *oracle* reasoner and an imperfect *LLM* reasoner. An ideal oracle reasoner would maintain an *oracle belief* distribution $b_t^* \in \Delta(\mathcal{S})$. Specifically, the oracle belief $b^\star$ is

recursively updated via Bayes' rule $B^\star$ upon taking action $a$ and observing $o$:

$$b_{t+1}^\star(s) := B^\star(b_t^\star, a, o) = \frac{O(o \mid s, a)b_t^\star(s)}{p_b(o \mid a)}, \tag{3}$$

where $p_b(o \mid a) := \sum_{s' \in \mathcal{S}} O(o \mid s', a)b_t^\star(s')$ is the Bayes-normalizer.

In contrast, an LLM agent does not perform exact Bayesian filtering. Instead, it maintains an *LLM belief* $b_t$, which represents its internal understanding of the latent state and what information remains missing. Given the action-observation pair $(a, o)$, the LLM belief evolves by $b_{t+1}(s) := B_\theta(b_t, a, o)$, where $\theta$ denotes LLM model parameters.

We compare the LLM agent's trajectory $(b_t, a_t, o_t)_{t \geq 1}$ with that of the oracle reasoner $(b_t^\star, a_t^\star, o_t^\star)_{t \geq 1}$. Specifically, the oracle samples action $a_t^\star \sim \pi(\cdot \mid b_t^\star)$ and receives observations from the environment generated via $o_t^\star \sim O(\cdot \mid s^\star, a_t^\star)$, updating its belief via $B^\star$ (Eq. 3). The LLM agent follows its own update rule $B_\theta$, sampling actions from $\pi(\cdot \mid b_t)$ and receiving observations via $O(\cdot \mid s^\star, a_t)$. Note that $s^\star$ denotes the truth latent state (fixed within an episode) used for analysis; both LLM and oracle agents do not observe $s^\star$. To quantify the discrepancy between beliefs, we use the $\ell_1$-distance: $d(b, b') := \sum_{s \in \mathcal{S}} |b(s) - b'(s)| \leq 2$, and denote $d_t := d(b_t, b_t^\star)$.

## B.2 Dynamics of Belief Trapping of LLM Agents in Active Reasoning

We begin by modeling *task progress* of active reasoning. Specifically, we introduce a truth-anchored potential function $\Psi : \Delta(\mathcal{S}) \mapsto \mathbb{R}^{\geq 0}$ that captures how concentrated the belief is on the true state $s^\star$.

**Definition B.1** (Truth-anchored potential). *For belief $b \in \Delta(\mathcal{S})$ and ground-truth state $s^\star$, define*

$$\Psi(b) := -\log b(s^\star).$$

*It holds that $\Psi(b) \in [0, \infty)$, with $\Psi(b) = 0$ iff $b(s^\star) = 1$ (task completion). Lower values of $\Psi(b)$ indicate higher confidence in the true state.*

Based on this, we assume that the oracle's belief $(b_t^\star)_{t \geq 1}$ is well-behaved and guaranteed to eventually converge to the truth.

**Assumption B.1** (Oracle Potential Convergence). *Along the oracle trajectory $(b_t^\star, a_t^\star, o_t^\star)_{t \geq 1}$, the potential $\Psi_t^\star := \Psi(b_t^\star)$ is bounded and convergent to zero. Specifically, there exists a deterministic nonincreasing sequence $(u_t)_{t \geq 1}$ with $u_1 = \Psi_1^\star$ and $u_t \searrow 0$ such that $\Psi_t^\star \leq u_t$ for all $t \geq 1$.*

To analyze the agent's behavior, we define several key quantities. Through the following definitions, we measure the expected information gain of an action under the *ideal* Bayesian update (Def. B.2), and the *actual* one-step progress when updating belief via the agent LLM (Def. B.3). We further quantify the discrepancy between the LLM agent's update and the Bayesian update (Def. B.4).

**Definition B.2** (One-Step Informativeness). *For belief $b$ and action $a$, define*

$$\mathcal{I}(b, a) := \Psi(b) - \mathbb{E}_{o \sim O(\cdot \mid s^\star, a)}\Big[\Psi\big(B^\star(b, a, o)\big)\Big].$$

*This captures the expected improvement of $\Psi$-progress when taking action $a$ from belief $b$.*

**Definition B.3** (One-step LLM-agent Progress). *The LLM agent's expected $\Psi$-progress given the current belief $b$:*

$$\mathcal{P}_\theta(b) := \Psi(b) - \mathbb{E}_{a \sim \pi(\cdot \mid b)}\mathbb{E}_{o \sim O(\cdot \mid s^\star, a)}\Big[\Psi\big(B_\theta(b, a, o)\big)\Big].$$

**Definition B.4** (LLM-Bayes update error). *For a belief $b$, define the conditional update error*

$$c_\theta(b) := \mathbb{E}_{a \sim \pi(\cdot \mid b)}\,\mathbb{E}_{o \sim O(\cdot \mid s^\star, a)}\Big[\Psi\big(B_\theta(b, a, o)\big) - \Psi\big(B^\star(b, a, o)\big)\Big].$$

We now state several technical assumptions required for our analysis.

**Assumption B.2.** *There exists $\eta \in (0, 1]$ such that $O(o \mid s, a) \geq \eta$ for all reachable $(o, s, a)$.*

**Assumption B.3** (Policy Sensitivity). *There exist $L_\pi \geq 0$ such that for any beliefs $b, b'$,*

$$\mathrm{TV}\big(\pi(\cdot \mid b), \pi(\cdot \mid b')\big) \leq L_\pi\, d(b, b'),$$

*where $\mathrm{TV}(P, Q) := \sup_{A \subseteq \mathcal{A}} |P(A) - Q(A)|$ denotes the total variation distance between probability distributions.*

**Assumption B.4** (Update-Error Growth). *There exist constants $m_\theta > 0$, $c_0 \geq 0$, and a threshold $U_0 \geq 0$ such that for all $b$ with $\Psi(b) \geq U_0$,*

$$c_\theta(b) \geq m_\theta \, \Psi(b) - c_0.$$

*That is, in high-uncertainty regimes, the LLM agent's update error grows at least linearly with $\Psi$.*

Accurately modeling belief states in active reasoning requires the agent to maintain a precise estimate of the underlying problem state and the remaining uncertainty, which is inherently challenging for LLMs. Assumption B.4 formalizes the imperfect belief modeling of LLM agents, which states that belief-update errors are amplified as the deviation increases. In high-uncertainty regimes, the update error grows at least linearly with $\Psi$. We next formalize the regime in which such misspecification dominates the oracle's informativeness:

**Definition B.5** (Belief Trap Region, BTR). *A set $\mathcal{R}_\theta \subseteq \Delta(\mathcal{S})$ is called a* belief trap region *for an agent parameterized by $\theta$ if it is* absorbing *and induces* non-positive progress*: for any belief $b \in \mathcal{R}_\theta$ and all subsequent times $t$ once entered, $\mathcal{P}_\theta(b) \leq 0$, and equivalently, $\mathbb{E}[\Psi(b_{t+1}) \mid b_t = b] \geq \Psi(b)$.*

Within BTRs, the potential sequence $\Psi_t$ becomes non-decreasing in expectation, *i.e.*, $\mathbb{E}[\Psi_{t+1} \mid b_t] \geq \Psi_t$. Consequently, once a trajectory enters the BTR, subsequent steps contribute little task progress and reinforce the stalled dynamics.

## B.3  DETAILED STATEMENT OF THEOREM 1

Next, we investigate the characteristics of the BTR as follows:

**Proposition B.1** (Sufficient Condition of entering BTR). *Under Assumptions B.2–B.4, define the constant $\bar{B} := 2\left(-L_\pi \log \eta + 1/\eta\right)$, the threshold $U := \max\{U_0, (\Psi_1^\star + \bar{B} + c_0)/m_\theta\}$, and let $t_S := \inf\{t : \Psi_t \geq U\}$ the first time the $\Psi$-potential reaches the threshold $U$. Then the following holds: if $t_S < \infty$, then for all $t \geq t_S$, $\mathcal{P}_\theta(b_t) \leq 0$, and equivalently, $\mathbb{E}\left[\Psi(b_{t+1}) \mid b_t\right] \geq \Psi(b_t)$.*

This result formalizes the *absorbing nature* of the belief-trap region: once the potential $\Psi$ exceeds the threshold $U$, the trajectory is locked into a regime where exploration is ineffective and the task progress no longer proceeds. Now we delve into the properties of the BTR entry time:

**Proposition B.2** (Upper bound of BTR entry time). *Strengthen Asmp. B.4 to global, i.e., $U_0 = 0$. Assume there exists $\mu > 0$ such that $\Psi_t^\star \geq \mu$ for all $t < t_S$. Assume $\delta := m_\theta \mu - (c_0 + \bar{B}) > 0$. Then the (expected) hitting time into BTR, denoted by $t_{\mathrm{BTR}} := \inf\{t : b_t \in \mathcal{R}_\theta\}$, obeys the upper bound*

$$t_{\mathrm{BTR}} \leq t_S \leq 1 + \left\lceil \log_{1+m_\theta} \frac{m_\theta \, U + \delta}{m_\theta \, \Delta_1 + \delta} \right\rceil.$$

Here $\Delta_1 := \Psi_1 - \Psi_1^\star$. The proofs for Proposition B.1 and Proposition B.2 are given in Appendix B.6 and Appendix B.7, respectively. This yields an explicit upper bound on the time to enter the trap: without corrective mechanisms, belief errors accumulate, so entering BTR becomes inevitable and can occur quickly once belief updates deteriorate.

## B.4  DETAILED STATEMENT OF THEOREM 2

**Theorem B.1** (BTR Induces Advantage Inversion). *Under the following assumptions: (i) the value function in policy optimization satisfies $V_t = g(b_t(s^*))$ for an increasing, differentiable $g$ with $\inf_x g'(x) \geq \kappa_V > 0$, and (ii) belief drop in BTRs: after entering the BTR, the belief exhibits a uniform negative drift, i.e., $\mathbb{E}[b_{k+1}(s^*) - b_k(s^*) \mid \mathcal{F}_k] \leq -\rho_b$ for $k \geq t_S$. Then, for any $t < t_S$, the expected advantage is bounded:*

$$\mathbb{E}[\widehat{A}_t] \leq \gamma \left(S_{pre}(t) - \kappa_V \rho_b S_{tail}^\ominus(t)\right), \tag{4}$$

*where $S_{pre}(t) = \sum_{j=0}^{t_S - t - 1} (\gamma\lambda)^j$ and $S_{tail}^\ominus(t) = \sum_{j=t_S - t}^{T - t - 2} (\gamma\lambda)^j$. Therefore, a sufficient condition for $\mathbb{E}[\widehat{A}_t] < 0$ is:*

$$\kappa_V \rho_b > S_{pre}(t)/S_{tail}^\ominus(t). \tag{5}$$

*In particular, when $\gamma\lambda \to 1$ (often used in practice for long-horizon agentic RL), the condition simplifies to $\kappa_V \rho_b > \Delta/L$, where $\Delta = t_S - t$ and $L = T - 1 - t_S$ are the prefix and tail lengths, respectively.*

The proof for Theorem B.1 is given in Appendix B.8. This theorem quantifies the credit assignment failure: a sufficiently long uninformative tail (large $L$) induces a negative drift that can dominate the positive contribution from the informative prefix, causing its overall gradient to point in the wrong direction and penalizing earlier exploratory actions.

## B.5 IMPORTANT LEMMAS

Before proving the propositions, we start by providing two important lemmas, and their proofs in Appendix B.10 and B.11.

**Lemma B.1** (Belief-Lipschitz Continuity of Informativeness). *Under Assumption B.2, for any fixed action $a \in \mathcal{A}$ and any beliefs $b, b' \in \Delta(\mathcal{S})$, we have*

$$\left| \mathcal{I}(b, a) - \mathcal{I}(b', a) \right| \leq \frac{1}{\eta} \|b - b'\|_1. \tag{6}$$

*Consequently, for any action distribution $q$,*

$$\left| \mathbb{E}_{a \sim q} \mathcal{I}(b, a) - \mathbb{E}_{a \sim q} \mathcal{I}(b', a) \right| \leq \frac{1}{\eta} \|b - b'\|_1. \tag{7}$$

**Lemma B.2** (Policy-Lipschitz Continuity of Informativeness). *Under Assumption B.2, for any fixed belief $b \in \Delta(\mathcal{S})$ and any two action distributions $q, q'$ on $\mathcal{A}$, we have*

$$|\mathbb{E}_{a \sim q} \mathcal{I}(b, a) - \mathbb{E}_{a \sim q'} \mathcal{I}(b, a)| \leq \Lambda \cdot \|q - q'\|_{\mathrm{TV}},$$

*where $\Lambda := -\log \eta$ and $\|q - q'\|_{\mathrm{TV}} := \sup_{A \subseteq \mathcal{A}} |q(A) - q'(A)|$ denotes the total variation norm.*

## B.6 PROOF OF PROPOSITION B.1

*Proof.* From Definitions B.2, B.3, and B.4, we have:

$$\mathcal{P}_\theta(b_t) = \mathbb{E}_{a_t \sim \pi(\cdot | b_t)}[\mathcal{I}(b_t, a_t)] - c_\theta(b_t). \tag{8}$$

Let $a_t \sim \pi(\cdot \mid b_t)$ and $a_t^\star \sim \pi(\cdot \mid b_t^\star)$. Leveraging the results in Lemma B.1 and B.2, we bound the difference in expected informativeness:

$$\left| \mathbb{E}_{a_t^\star}[\mathcal{I}(b_t^\star, a_t^\star)] - \mathbb{E}_{a_t}[\mathcal{I}(b_t, a_t)] \right| \tag{9}$$

$$\leq \left| \mathbb{E}_{a_t^\star}[\mathcal{I}(b_t^\star, a_t^\star)] - \mathbb{E}_{a_t}[\mathcal{I}(b_t^\star, a_t)] \right| + \left| \mathbb{E}_{a_t}[\mathcal{I}(b_t^\star, a_t)] - \mathbb{E}_{a_t}[\mathcal{I}(b_t, a_t)] \right| \tag{10}$$

$$\leq \Lambda \, \mathrm{TV}(\pi(\cdot \mid b_t^\star), \pi(\cdot \mid b_t)) + L_b \, d(b_t^\star, b_t) \tag{11}$$

$$\leq (\Lambda L_\pi + L_b) \, d_t. \tag{12}$$

From Assumption B.1, we have:

$$\mathbb{E}_{a_t^\star}[\mathcal{I}(b_t^\star, a_t^\star)] = \Psi(b_t^\star) - \mathbb{E}[\Psi(b_{t+1}^\star)] \leq \Psi_0. \tag{13}$$

Combining with Eq. 12 yields:

$$\mathbb{E}_{a_t}[\mathcal{I}(b_t, a_t)] \leq \Psi_0 + (\Lambda L_\pi + L_b) d_t. \tag{14}$$

Since $d_t \leq 2$, we obtain:

$$\mathbb{E}_{a_t}[\mathcal{I}(b_t, a_t)] \leq \Psi_0 + 2(\Lambda L_\pi + L_b) = K. \tag{15}$$

Now, from Assumption 1, if $\Psi(b_t) \geq U_0$, then:

$$c_\theta(b_t) \geq m_\theta \Psi(b_t) - c_0. \tag{16}$$

Substituting into Eq. 8 gives:

$$\mathcal{P}_\theta(b_t) \leq K - \big(m_\theta \Psi(b_t) - c_0\big). \tag{17}$$

Thus, if $\Psi(b_t) \geq (K + c_0)/m_\theta$ and $\Psi(b_t) \geq U_0$ (i.e., $\Psi(b_t) \geq U$), then $\mathcal{P}_\theta(b_t) \leq 0$, meaning:

$$\mathbb{E}[\Psi(b_{t+1}) \mid b_t] \geq \Psi(b_t). \tag{18}$$

Since $c_\theta(\cdot)$ is lower-bounded by a function that is nondecreasing in $\Psi$ (Assumption 1), this argument applies inductively for all $t \geq t_0$, confirming the supermartingale property and the stalling behavior. $\square$

## B.7 Proof of Proposition B.2

*Proof.* For simplicity, let $\Psi_t := \Psi(b_t)$ and $\Psi_t^\star := \Psi(b_t^\star)$. From the definitions of agent progress $\mathcal{P}_\pi(b)$ and update error $c_\theta(b)$, we have the one-step expectation:

$$\mathbb{E}[\Psi_{t+1} \mid \mathcal{F}_t] = \Psi_t - \mathbb{E}_{a_t \sim \pi(\cdot|b_t)}[\mathcal{I}(b_t, a_t)] + c_\theta(b_t). \tag{19}$$

For the oracle, it holds that:

$$\mathbb{E}[\Psi_{t+1}^\star \mid \mathcal{F}_t] = \Psi_t^\star - \mathbb{E}_{a_t^\star \sim \pi(\cdot|b_t^\star)}[\mathcal{I}(b_t^\star, a_t^\star)]. \tag{20}$$

Subtracting these two equations yields the fundamental drift identity for the gap $\Delta_t = \Psi_t - \Psi_t^\star$:

$$\mathbb{E}[\Delta_{t+1} - \Delta_t \mid \mathcal{F}_t] = \left(\mathbb{E}_{a_t^\star}[\mathcal{I}(b_t^\star, a_t^\star)] - \mathbb{E}_{a_t}[\mathcal{I}(b_t, a_t)]\right) + c_\theta(b_t). \tag{21}$$

From what have been shown in Eq. 12, we have,

$$\left|\mathbb{E}_{a_t^\star}[\mathcal{I}(b_t^\star, a_t^\star)] - \mathbb{E}_{a_t}[\mathcal{I}(b_t, a_t)]\right| \leq (\Lambda L_\pi + L_b)d_t \leq 2\left(\Lambda L_\pi + L_b\right) =: \bar{B}. \tag{22}$$

Substituting into 21 gives:

$$\mathbb{E}[\Delta_{t+1} - \Delta_t \mid \mathcal{F}_t] \geq -\bar{B} + c_\theta(b_t). \tag{23}$$

The strengthened Assumption 1 implies:

$$c_\theta(b_t) \geq m_\theta \Psi_t - c_0 = m_\theta(\Delta_t + \Psi_t^\star) - c_0. \tag{24}$$

Substituting into 23 yields:

$$\mathbb{E}[\Delta_{t+1} - \Delta_t \mid \mathcal{F}_t] \geq m_\theta \Delta_t + \left(m_\theta \Psi_t^\star - (c_0 + \bar{B})\right). \tag{25}$$

Rearranging terms:

$$\mathbb{E}[\Delta_{t+1} \mid \mathcal{F}_t] \geq (1 + m_\theta)\Delta_t + \left(m_\theta \Psi_t^\star - (c_0 + \bar{B})\right). \tag{26}$$

By *the law of total expectation*, we have,

$$\mathbb{E}\left[\mathbb{E}[\Delta_{t+1} \mid \mathcal{F}_t]\right] \geq \mathbb{E}\left[(1 + m_\theta)\Delta_t + \left(m_\theta \Psi_t^\star - (c_0 + \bar{B})\right)\right] \tag{27}$$

$$\mathbb{E}[\Delta_{t+1}] \geq (1 + m_\theta)\mathbb{E}[\Delta_t] + m_\theta \mathbb{E}[\Psi_t^\star] - (c_0 + \bar{B}). \tag{28}$$

Iterating this inequality gives:

$$\mathbb{E}[\Delta_T] \geq (1 + m_\theta)^{T-1}\Delta_1 + \sum_{k=1}^{T-1}(1 + m_\theta)^{T-1-k}\mathbb{E}\left[m_\theta \Psi_k^\star - (c_0 + \bar{B})\right]. \tag{29}$$

As assumed in the proposition, there exists $\mu > 0$ such that for all $k \geq 1$, $\Psi_k^\star \geq \mu$ almost surely. This implies $\mathbb{E}[\Psi_k^\star] \geq \mu$. Then:

$$\mathbb{E}\left[m_\theta \Psi_k^\star - (c_0 + \bar{B})\right] \geq m_\theta \mu - (c_0 + \bar{B}) =: \delta. \tag{30}$$

Substituting into Eq. 29:

$$\mathbb{E}[\Delta_T] \geq (1 + m_\theta)^{T-1}\Delta_1 + \delta \sum_{k=1}^{T-1}(1 + m_\theta)^{T-1-k} \tag{31}$$

$$= (1 + m_\theta)^{T-1}\Delta_1 + \delta \frac{(1 + m_\theta)^{T-1} - 1}{m_\theta}. \tag{32}$$

We now show that $\mathbb{E}[\Psi_T]$ exceeds $U$ in finite time. Recall:

$$\mathbb{E}[\Psi_T] = \mathbb{E}[\Delta_T] + \mathbb{E}[\Psi_T^\star] \geq \mathbb{E}[\Delta_T]. \tag{33}$$

A sufficient condition is therefore:

$$(1 + m_\theta)^{T-1}\Delta_1 + \delta \frac{(1 + m_\theta)^{T-1} - 1}{m_\theta} \geq U. \tag{34}$$

Since $\delta > 0$ and $1 + m_\theta > 1$, the left-hand side grows exponentially with $T$. Thus, for any $U > 0$, there exists a finite $T$ such that Eq. 34 holds. Specifically, we have:

$$(1 + m_\theta)^{T-1} \geq \frac{m_\theta U + \delta}{m_\theta \Delta_1 + \delta}. \tag{35}$$

Taking logarithms yields the explicit bound:

$$T \geq 1 + \left\lceil \frac{1}{\log(1 + m_\theta)} \log\left(\frac{m_\theta U + \delta}{m_\theta \Delta_1 + \delta}\right) \right\rceil. \tag{36}$$

This completes the proof.

$$\square$$

### B.8    PROOF OF THEOREM B.1

*Proof.* We decompose the advantage estimator: $\widehat{A}_t = \mathrm{Pre}(t) + \mathrm{Tail}(t)$, where

$$\mathrm{Pre}(t) = \sum_{j=0}^{t_S-t-1} q^j \delta_{t+j}, \quad \mathrm{Tail}(t) = \sum_{j=t_S-t}^{T-t-1} q^j \delta_{t+j}, \quad \text{and } q = \gamma\lambda.$$

For any $k < t_S$, the TD-error $\delta_k = \gamma V_{k+1} - V_k$ (since $r_k = 0$). Because $V_k \in [0,1]$,

$$\mathbb{E}[\delta_k \mid \mathcal{F}_k] = \gamma\mathbb{E}[V_{k+1} \mid \mathcal{F}_k] - V_k \le \gamma \cdot 1 - 0 = \gamma.$$

Taking full expectation and summing over the prefix yields:

$$\mathbb{E}[\mathrm{Pre}(t)] \le \gamma S_{\mathrm{pre}}(t). \tag{37}$$

We split the tail into the main part and the terminal step:

$$\mathrm{Tail}(t) = \underbrace{\sum_{j=t_S-t}^{T-t-2} q^j \delta_{t+j}}_{\mathrm{Tail}^-(t)} + q^{T-t-1}\delta_{T-1}.$$

For the terminal step, $\delta_{T-1} = R_T - V_{T-1}$, so $\mathbb{E}[\delta_{T-1} \mid \mathcal{F}_{T-1}] = 0$, and thus $\mathbb{E}[q^{T-t-1}\delta_{T-1}] = 0$.

Now, fix $k \in \{t_S, \ldots, T-2\}$. We analyze $\mathbb{E}[\delta_k \mid \mathcal{F}_k]$:

$$\mathbb{E}[\delta_k \mid \mathcal{F}_k] = \gamma\mathbb{E}[V_{k+1} - V_k \mid \mathcal{F}_k] + (\gamma-1)V_k \tag{38}$$
$$\le \gamma\mathbb{E}[V_{k+1} - V_k \mid \mathcal{F}_k] \quad \text{(since } V_k \ge 0 \text{ and } \gamma - 1 \le 0\text{)}. \tag{39}$$

By the calibration assumption, $V_{k+1} - V_k = g(b_{k+1}(s^*)) - g(b_k(s^*))$. Since $g$ is differentiable with $g' \ge \kappa_V > 0$, and since $\mathbb{E}[b_{k+1}(s^*) - b_k(s^*) \mid \mathcal{F}_k] \le -\rho_b$ by assumption, we have:

$$\mathbb{E}[V_{k+1} - V_k \mid \mathcal{F}_k] = \mathbb{E}[g'(\xi_k)(b_{k+1}(s^*) - b_k(s^*)) \mid \mathcal{F}_k] \tag{40}$$
$$\le \kappa_V \mathbb{E}[b_{k+1}(s^*) - b_k(s^*) \mid \mathcal{F}_k] \quad \text{(since } g'(\xi_k) \ge \kappa_V\text{)} \tag{41}$$
$$\le -\kappa_V \rho_b. \tag{42}$$

Therefore, $\mathbb{E}[\delta_k \mid \mathcal{F}_k] \le -\gamma\kappa_V\rho_b$. Taking full expectation and summing over the tail gives:

$$\mathbb{E}[\mathrm{Tail}^-(t)] \le -\gamma\kappa_V\rho_b S_{\mathrm{tail}}^{\ominus}(t). \tag{43}$$

Combining Eq. 37 and Eq. 43 proves the main bound Eq. 4. The inversion condition Eq. 5 follows directly by requiring the right-hand side of Eq. 4 to be negative.

From what have been proved above, we have:

$$\mathbb{E}[\widehat{A}_t] = \mathbb{E}[\mathrm{Pre}(t)] + \mathbb{E}[\mathrm{Tail}(t)] \le \mathbb{E}[\widehat{A}_t^{\mathrm{pre}}] - \gamma\kappa_V\rho_b S_{\mathrm{tail}}^{\ominus}(t).$$

Rearranging terms yields: $\mathbb{E}[\widehat{A}_t^{\mathrm{pre}}] \ge \mathbb{E}[\widehat{A}_t] + \gamma\kappa_V\rho_b S_{\mathrm{tail}}^{\ominus}(t)$.

$\square$

### B.9    PROOF OF PROPOSITION 1

*Proof.* Fix any $k$-step segment $(t+1, \ldots, t+k)$ that lies entirely outside the BTR, so that $g_s \ge \rho > 0$ for all $s \in \{t+1, \ldots, t+k\}$. By definition of the biased Gaussian-noise model, we have $d_s = g_s + \beta_s + \xi_s$, where $|\beta_s| \le M, \xi_s \sim \mathcal{N}(0, \sigma^2)$ independently across $s$. On a step $s$ outside the BTR, a local false truncation event occurs when the proxy falls below the threshold $\Delta_{\min}$ (*cf.*, Def. 2) despite $g_s \ge \rho$:

$$\mathcal{E}_s := \{d_s < \Delta_{\min}\} = \{g_s + \beta_s + \xi_s < \Delta_{\min}\}.$$

Using $g_s \ge \rho$ and $|\beta_s| \le M$, we obtain $g_s + \beta_s \ge \rho - M$. Hence

$$\Pr(\mathcal{E}_s) = \Pr(g_s + \beta_s + \xi_s < \Delta_{\min}) \le \Pr(\rho - M + \xi_s < \Delta_{\min}) = \Pr(\xi_s < \Delta_{\min} - (\rho - M)).$$

Define the margin $a := \rho - M - \Delta_{\min}$. By the assumption $\Delta_{\min} < \rho - M$, we have $a > 0$ and therefore,

$$\Pr(\mathcal{E}_s) \ \leq \ \Pr(\xi_s < -a).$$

Since $\xi_s \sim \mathcal{N}(0, \sigma^2)$, the standard concentration inequality gives, for any $a > 0$, we have

$$\Pr(\xi_s \leq -a) \ \leq \ \exp\Big( - \tfrac{a^2}{2\sigma^2} \Big).$$

Applying this with $a = \rho - M - \Delta_{\min} > 0$ yields

$$\Pr(\mathcal{E}_s) \ \leq \ \exp\Big( - \tfrac{(\rho - M - \Delta_{\min})^2}{2\sigma^2} \Big). \tag{44}$$

Recall that the $\mathbf{T^3}$ rule with window size $k$ triggers at the end of a $k$-step segment only if all $k$ steps in the window are classified as "non-informative". For a non-BTR segment $(t+1, \ldots, t+k)$, activating $\mathbf{T^3}$ therefore corresponds to the intersection of the $k$ single-step events $\mathcal{E}_{t+1}, \ldots, \mathcal{E}_{t+k}$:

$$\mathcal{E}_{t+1,\ldots,t+k} := \bigcap_{s=t+1}^{t+k} \mathcal{E}_s.$$

By independence of the noises $\{\xi_s\}$ across $s$ and because each $\mathcal{E}_s$ is determined by $\xi_s$, we have

$$\Pr(\mathcal{E}_{t+1,\ldots,t+k}) = \prod_{s=t+1}^{t+k} \Pr(\mathcal{E}_s).$$

Applying the single-step bound (Eq. 44) uniformly yields

$$\Pr(\mathcal{E}_{t+1,\ldots,t+k}) \ \leq \ \exp\Big( - \tfrac{k(\rho - M - \Delta_{\min})^2}{2\sigma^2} \Big).$$

To ensure that the false-truncation probability on any $k$-step non-BTR segment is at most $\delta \in (0, 1)$, it suffices to require

$$\exp\Big( - \tfrac{k(\rho - M - \Delta_{\min})^2}{2\sigma^2} \Big) \ \leq \ \delta,$$

which is equivalent to

$$k\,(\rho - M - \Delta_{\min})^2 \ \geq \ 2\sigma^2 \log(1/\delta).$$

$\square$

### B.10 PROOF OF LEMMA B.1

*Proof.* We begin by showing the closed form of one-step informativeness $\mathcal{I}(b, a)$. Combing Definitions B.1, B.2 and Eq. 3, we have,

$$\mathcal{I}(b, a) = \Psi(b) - \mathbb{E}_{o \sim O(\cdot | s^\star, a)} \big[ \Psi(B^\star(b, a, o)) \big] \tag{45}$$

$$= -\log b(s^\star) - \mathbb{E}_{o \sim O(\cdot | s^\star, a)} \Big[ -\log \Big( \frac{O(o \mid s^\star, a) b(s^\star)}{p_b(o \mid a)} \Big) \Big] \tag{46}$$

$$= \mathbb{E}_{o \sim O(\cdot | s^\star, a)} \Big[ \log \frac{O(o \mid s^\star, a)}{p_b(o \mid a)} \Big]. \tag{47}$$

For fixed $a$, Let $P(o) := O(o \mid s^\star, a)$, and $Q_b(o) := p_b(o \mid a) = \sum_s b(s) O(o \mid s, a)$. Then we have:

$$\mathcal{I}(b, a) = \mathbb{E}_{o \sim P} \Big[ \log \frac{P(o)}{Q_b(o)} \Big] = \underbrace{\mathbb{E}_P[\log P(o)]}_{\text{constant in } b} - \mathbb{E}_P[\log Q_b(o)]. \tag{48}$$

By the non-degeneracy assumption (Assumption B.2), $O(o \mid s, a) \geq \eta$ for all reachable $o, s$. Consequently, for any belief $b$ and any observation $o$,

$$Q_b(o) = \sum_{s \in \mathcal{S}} b(s) O(o \mid s, a) \geq \sum_{s \in \mathcal{S}} b(s) \cdot \eta = \eta. \tag{49}$$

Thus, $Q_b(o) \geq \eta$ and $Q_{b'}(o) \geq \eta$ hold for all $o$.

For any $x, y \geq \eta > 0$, we have the elementary bound

$$| \log x - \log y | = \left| \int_y^x \frac{1}{t} \, dt \right| \leq \frac{|x - y|}{\min\{x, y\}} \leq \frac{|x - y|}{\eta}. \tag{50}$$

Applying this with $Q_b(o)$ and $Q_{b'}(o)$ yields:

$$| \log Q_b(o) - \log Q_{b'}(o) | \leq \frac{|Q_b(o) - Q_{b'}(o)|}{\eta} \quad \text{for all } o. \tag{51}$$

Taking expectation under $P$ and properties of expectation, we get:

$$|\mathbb{E}_P[\log Q_b(o)] - \mathbb{E}_P[\log Q_{b'}(o)]| \leq \mathbb{E}_P\left[ | \log Q_b(o) - \log Q_{b'}(o) | \right] \tag{52}$$

$$\leq \mathbb{E}_P\left[ \frac{|Q_b(o) - Q_{b'}(o)|}{\eta} \right] \tag{53}$$

$$\leq \frac{1}{\eta} \| Q_b - Q_{b'} \|_1. \tag{54}$$

Since $\mathcal{I}(b, a) = \text{const} - \mathbb{E}_P[\log Q_b(o)]$, it follows that

$$|\mathcal{I}(b, a) - \mathcal{I}(b', a)| \leq \frac{1}{\eta} \| Q_b - Q_{b'} \|_1. \tag{55}$$

We have

$$|Q_b(o) - Q_{b'}(o)| = \left| \sum_{s \in \mathcal{S}} (b(s) - b'(s)) O(o \mid s, a) \right| \leq \sum_{s \in \mathcal{S}} |b(s) - b'(s)| O(o \mid s, a). \tag{56}$$

Summing over $o$ gives:

$$\| Q_b - Q_{b'} \|_1 = \sum_{o \in \mathcal{O}} |Q_b(o) - Q_{b'}(o)| \leq \sum_{o \in \mathcal{O}} \sum_{s \in \mathcal{S}} |b(s) - b'(s)| O(o \mid s, a) \tag{57}$$

$$= \sum_{s \in \mathcal{S}} |b(s) - b'(s)| \sum_{o \in \mathcal{O}} O(o \mid s, a) \tag{58}$$

$$= \| b - b' \|_1. \tag{59}$$

Combining this with Eq. 55 yields the pointwise bound:

$$|\mathcal{I}(b, a) - \mathcal{I}(b', a)| \leq \frac{1}{\eta} \| b - b' \|_1. \tag{60}$$

For any action distribution $q$, by the linearity of expectation:

$$|\mathbb{E}_{a \sim q} \mathcal{I}(b, a) - \mathbb{E}_{a \sim q} \mathcal{I}(b', a)| \leq \mathbb{E}_{a \sim q} |\mathcal{I}(b, a) - \mathcal{I}(b', a)| \leq \mathbb{E}_{a \sim q} \left[ \frac{1}{\eta} \| b - b' \|_1 \right] = \frac{1}{\eta} \| b - b' \|_1. \tag{61}$$

$\square$

## B.11 PROOF OF LEMMA B.2

*Proof.* For fixed $b$, define $f(a) := \mathcal{I}(b, a)$. We first show that $f$ is bounded. By non-degeneracy, $O(o \mid s, a) \geq \eta$ for all $o, s, a$. Consequently, for any $a$,

$$p_b(o \mid a) = \sum_{s \in \mathcal{S}} b(s) O(o \mid s, a) \geq \eta \quad \text{and} \quad O(o \mid s^\star, a) \geq \eta.$$

By Eq. 47, we have

$$0 \leq \mathcal{I}(b, a) = \mathbb{E}_{o \sim O(\cdot \mid s^\star, a)} \left[ \log \frac{O(o \mid s^\star, a)}{p_b(o \mid a)} \right] \leq \mathbb{E}_{o \sim O(\cdot \mid s^\star, a)}[\log(1/\eta)] = -\log \eta.$$

Hence, $\| f \|_\infty \leq -\log \eta$, where $\| \cdot \|_\infty$ denotes the supremum norm $\| f \|_\infty := \sup_{a \in \mathcal{A}} |f(a)|$.

The result now follows from a standard property of the total variation norm: for any bounded function $f$,

$$|\mathbb{E}_{a \sim q} f(a) - \mathbb{E}_{a \sim q'} f(a)| \leq \| f \|_\infty \cdot \| q - q' \|_{\text{TV}} \leq (-\log \eta) \cdot \| q - q' \|_{\text{TV}}.$$

$\square$

## C EMPIRICAL VERIFICATION OF THE THEORY

### C.1 EMPIRICAL VERIFICATION OF ASSUMPTION 1

A direct empirical validation of Assumption 1 is inherently challenging, as neither the oracle Bayesian update $B^\star$ nor the LLM agent's internal belief state $b_t$ is directly observable. To address this, we design a controlled study on the PE task that enables practical and theory-aligned approximations of all relevant quantities $U_0, m_\theta, c_0$.

**(i) Approximating the potential $\Psi$.** Each interaction round in PE provides the model's explicit estimate of the latent user-preference vector, denoted by $w_t$ (here we use $w$ to denote the preference vector, same as $v$ in main text). We define

$$d(w_t) := \|w_t - w^\star\|_2^2,$$

and use $d(w_t)$ to serve as an operational proxy of the potential, *i.e.*, $\hat{\Psi}_t := d(w_t)$. This proxy preserves the essential properties of the theoretical potential: it is non-negative and equals zero if and only if the task is solved. Note that $w^\star$ is not available to the agent.

**(ii) Approximating the oracle Bayesian update $B^\star$.** Although the true Bayesian posterior is inaccessible, we construct a principled surrogate update rule $\hat{B}$ following a standard linear-Gaussian update. Specifically, given the model's query $a_t := (A, B)$ where $A, B$ denote the movie pair to compare and the observed feedback $o_t$, we define

$$w'_{t+1} := \hat{B}(w_t, a_t, o_t) = w_t + K_t\, m_t\big(o_t - m_t^\top w_t\big), \qquad K_t = \frac{\sigma_0^2}{\sigma_0^2\|m_t\|_2^2 + \sigma^2}.$$

Here, $m_t \in \mathbb{R}^d$ is the movie-attribute *difference vector* for the pair of movies selected by the LLM's query, *i.e.*, $m_t = \mathrm{attr}(A) - \mathrm{attr}(B)$. The binary observation $o_t \in \{-1, +1\}$ corresponds to the user's response and is given by $o_t = \mathrm{sign}(m_t^\top w^\star)$. The terms $\sigma_0^2$ and $\sigma^2$ denote prior and observation noise variances; following standard practice, we set both to $1.0$. In contrast, the LLM agent updates its estimate via

$$w_{t+1} := B_\theta(w_t, a_t, o_t),$$

which reflects the internal belief dynamics induced by its parameters $\theta$.

**(iii) Constructing observable samples of the update-error term.** Using the above approximations, we instantiate the update-error quantity via

$$\hat{c}_\theta(b_t) := d(w_{t+1}) - d(w'_{t+1}) \approx c_\theta(b_t).$$

We totally collect over **150k** samples of $(\hat{\Psi}_t, \hat{c}_\theta(b_t))$ using rollouts from the Qwen-2.5 series models, which provide a sufficiently rich empirical basis for inspecting the assumption.

**(iv) Estimating $m_\theta$, $U_0$, $c_0$ via lower-envelope fitting.** Since Assumption 1 concerns only a *lower bound* relationship, we estimate the empirical lower envelope using a principled two-step procedure:

(a) *Lower-envelope extraction via binning.* According to Asmp. 1, belief deviation of the LLM agent will be further amplified once it progresses into a high-$\hat{\Psi}$ region. Hence we empirically select a proper value of $\hat{U}_0$ such that large belief deviations are observed. We then partition the range $[\hat{U}_0, \hat{\Psi}_{\max}]$ into $B$ equal-width bins $[\psi_{b-1}, \psi_b)$, where $\hat{\Psi}_{\max}$ represents maximum observed $\hat{\Psi}_t$ in data. For each bin $b$, we compute:

$$x_b := \mathbb{E}[\hat{\Psi}_t \mid \hat{\Psi}_t \in \text{bin } b], \qquad y_b := \mathrm{Quantile}_{0.1}\big(\hat{c}_\theta(b_t) \mid \hat{\Psi}_t \in \text{bin } b\big),$$

where $y_b$ captures the empirical 10th-percentile lower envelope within the bin.

(b) *Linear estimation on the active region.* Restricting to the active region $\hat{\Psi}_t \geq \hat{U}_0$, we fit a linear model to the extracted lower-envelope points:

$$y_b \approx \hat{m}_\theta\, x_b - \hat{c}_0.$$

The resulting $(\hat{m}_\theta, \hat{c}_0)$ provide empirical estimates of the coefficients in Assumption 1.

We visualize the whole procedure and the fitted linear model in Fig. 7. The above procedure yields an interpretable empirical characterization of the lower-bound growth pattern required by Assumption 1.

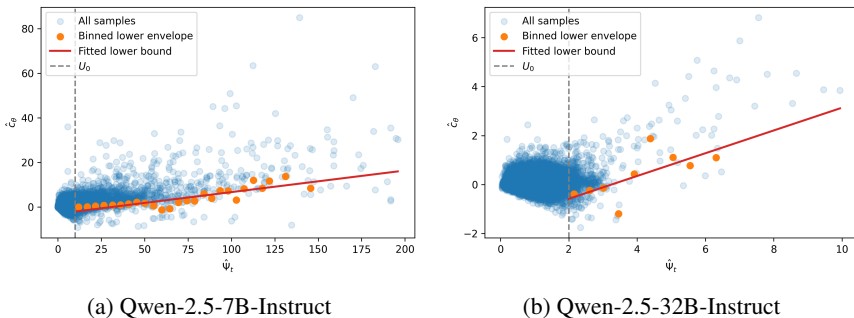

(a) Qwen-2.5-7B-Instruct  (b) Qwen-2.5-32B-Instruct

Figure 7: Empirical visualization of Assumption 1 on the PE task. The dashed vertical line marks the empirically determined threshold $\hat{U}_0$. Blue points show all samples, while orange points represent the binned lower envelope, obtained by partitioning the range of $\{\hat{\Psi}_t \geq \hat{U}_0\}$ into equal-width bins and taking the 10th percentile of $\hat{c}_\theta$ within each bin. The red line is a linear fit to these lower-envelope points. For (a), we empirically select $\hat{U}_0 = 10$, and obtain the linear fit: $\hat{c}_\theta = 0.0969 \times \hat{\Psi} - 3.0478$. For (b), similarly, we select $\hat{U}_0 = 2$ and obtain $\hat{c}_\theta = 0.4655 \times \hat{\Psi} - 1.5158$.

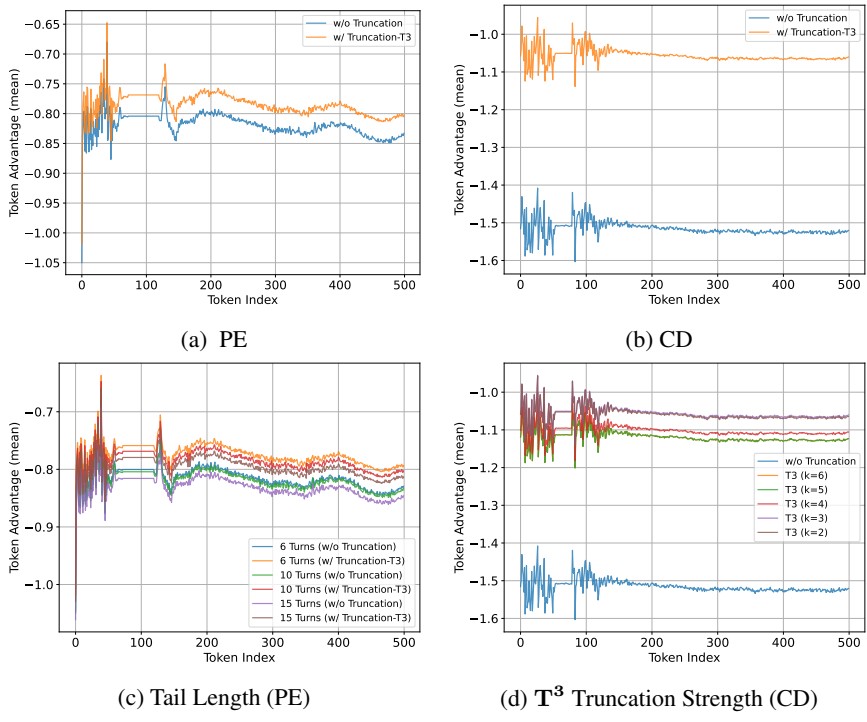

(a) PE  (b) CD

(c) Tail Length (PE)  (d) $\mathbf{T}^3$ Truncation Strength (CD)

Figure 8: Empirical verification of Theorem 2 and Corollary 1. (a-b) Without truncation, early-token advantages exhibit a clear negative drift, while $\mathbf{T}^3$ consistently elevates them across PE and CD tasks. (c) Longer uninformative tails (higher maximum interaction turns, from 6 to 15) cause stronger suppression of early advantages. (d) Stronger $\mathbf{T}^3$ truncation (smaller $k$) yields cleaner, less-biased early advantages.

## C.2   VERIFICATION OF THEOREM 2 AND COROLLARY 1

To empirically validate the credit assignment pathology formalized in Theorem 2 and the mitigating effect of $\mathbf{T}^3$ stated in Corollary 1, we designed a controlled experiment to isolate the impact of the uninformative trajectory tail on the advantage estimates of preceding exploratory actions.

**Experimental Setup.** Given a fixed policy optimized via standard PPO paradigm, we generated two sets of rollouts: one using the standard method (*w/o Truncation*) and one using the $\mathbf{T}^3$ truncation rule (*w/ Truncation*). To precisely measure the contamination effect of the uninformative tail without the

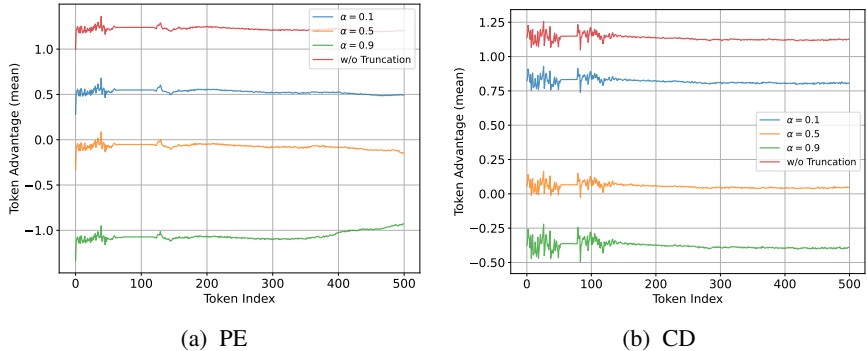

(a) PE  (b) CD

Figure 9: Empirical verification of the effect of false positive on (a) PE and (b) CD tasks. More aggressive false-positive truncation (larger $\alpha$) systematically reduces the advantages of early exploratory actions, reflecting the removal of positive future return contributions.

confounding factor of successful outcomes, we filtered and exclusively analyzed rollouts that resulted in a *failure* (i.e., a final reward of $0$). We then computed the Generalized Advantage Estimation (GAE) for each token in the first 500 tokens of these failed trajectories. Finally, we calculated the mean advantage at each token index across all rollouts within each condition.

**Main Results.** The results across the CD and MR datasets are presented in Fig. 8a and 8b. In the *w/o Truncation* condition, the mean advantage of early tokens is suppressed, while applying the $\mathbf{T^3}$ truncation rule (*w/ Truncation*) consistently elevates the mean advantage of the early tokens. This suggests that the uninformative tail inside the BTR introduces a negative drift that systematically corrupts the advantage estimates of the preceding exploratory actions, and shows that the $\mathbf{T^3}$ early-truncation mechanism effectively alleviates this issue, preserving the integrity of the gradient signal during policy optimization.

**Effect of Tail Length and Truncation Strength:** We further vary the effective tail length and truncation strength. As shown in Fig. 8c, longer uninformative tails in the *w/o Truncation* setup led to a more severe suppression of early-token advantages. Fig. 8d exhibits that stronger (more aggressive) truncation in the *w/ Truncation* setup resulted in higher and less corrupted advantage estimates for the preserved trajectory prefix. This is consistent with the theoretical outcome of this work.

### C.3    COMPLEMENTARY ANALYSIS OF FALSE-POSITIVE TRUNCATION AND ITS IMPACT

Since $\mathbf{T^3}$ leverages observable surrogates of the BTR to construct the truncation condition, the frequency of false positives is empirically limited. However, premature (*false-positive*) truncation can, in principle, remove useful exploratory steps and harms optimization. We provide both an analytical discussion and a diagnostic experiment.

**Analytical perspective.** Under the standard GAE decomposition, the advantage of an early token $t$ aggregates future TD-errors: $A_t = \sum_{u=t}^{T}(\gamma\lambda)^{u-t}\delta_u$. Theorem 2 characterizes the "uninformative tail" regime in which the expected TD-errors $\delta_u$ are negative; failing to truncate such tails induces a downward drift on $A_t$. A premature truncation corresponds to the opposite scenario: the trajectory has not yet entered the belief-trap region, and truncating at this point may discard future steps whose TD-errors $\delta_u$ could have been positive. Consequently, $A_t$ may be reduced due to the loss of these potentially informative and reward-contributing steps.

**Diagnostic experiment.** To make this effect concrete, we conducted a controlled diagnostic experiment. We fixed a trained vanilla-PPO policy and generated a set of full rollouts. To focus our study on the effect of false positives, we filtered the rollouts to those with a final reward of 1, ensuring that the retained trajectories contain genuinely informative future signals and do not enter the BTR. On these trajectories, we simulated false-positive truncation as follows: With probability $\alpha$, the trajectory is forcibly truncated at turn 3 (the maximum allowed turn is 10). With probability $1-\alpha$, the trajectory proceeds normally to completion. This creates a clean setting in which any degradation can be attributed *solely* to premature truncation. For each early-stage token position $t = 1{:}500$, we computed the mean GAE advantage across rollouts for different $\alpha$ values.

**Results.** We present the results for the CD and PE datasets in Fig. 9. As expected, more aggressive false-positive truncation systematically reduces the advantages of early exploratory actions, confirming that premature (false-positive) truncation negatively impacts credit assignment.

# D  COMPLEMENTARY EMPIRICAL ANALYSIS

In this section, we present complementary experimental results to provide further insights.

## D.1  RATIONALE FOR SELECTING BINARY SIMILARITY THRESHOLD IN PE

In the PE task, the reward is derived from the cosine similarity between the model-predicted preference vector and the ground-truth preference. We convert the similarity into a binary reward by activating it only when the similarity exceeds a prescribed threshold. To understand the effect of this threshold, we evaluate several settings $\{0.85, 0.88, 0.90, 0.95\}$ using Qwen-2.5-7B-Instruct trained with PPO.

Table 5 summarizes the results. Lower thresholds (e.g., below $0.80$) cause the reward to activate (*i.e.*, a 1 reward) almost continuously, which diminishes the discriminative value of high-quality predictions. Conversely, very high thresholds (e.g., above $0.95$) make activations extremely rare, preventing PPO from learning effectively. Mid-range thresholds between $0.85$ and $0.90$ consistently yield stable training dynamics and strong downstream performance. We use $0.88$, which lies within this empirically robust region, in the main experiments of the PE task.

Table 5: Effect of the binary-similarity threshold on PE performance (BinarySim). All results use Qwen-2.5-7B-Instruct trained with PPO.

| Threshold | 0.85 | 0.88 | 0.90 | 0.95 |
|---|---|---|---|---|
| PPO (vanilla) | 55.33 | 42.00 | 33.67 | 4.33 |
| PPO + $\mathbf{T^3}$ | 63.00 | 49.00 | 37.67 | 3.67 |

## D.2  EFFECT OF REFERENCE-SET SIZE ON REDUNDANCY-INDUCED STALLING

**Empirical Verification.** To further examine the role of redundancy in inducing belief-trap regions (BTR) in the PE task as mentioned in Sec. 3.3.2, we investigate how the frequency of truncation varies with the size of the reference set $S$. We evaluate truncation ratios across different reference-set sizes $S \in \{10, 15, 20, 25, 30\}$ for the Qwen-2.5-Instruct model family. Table 6 reports the results.

Table 6: Truncation ratio (%) under different reference-set sizes $S$ for Qwen-2.5-Instruct models. Larger $S$ corresponds to more potentially redundant comparisons.

| $S$ | 10 | 15 | 20 | 25 | 30 |
|---|---|---|---|---|---|
| 3B | 41.67 | 39.67 | 46.67 | 44.33 | 50.00 |
| 7B | 50.67 | 53.67 | 54.00 | 56.67 | 56.67 |
| 14B | 23.33 | 30.33 | 27.00 | 33.00 | 33.33 |
| 32B | 38.00 | 39.67 | 39.33 | 50.33 | 46.33 |

Across all model scales, the truncation ratio exhibits a general upward trend as $S$ increases from 10 to 30. This pattern suggests that larger reference sets may introduce additional noisy or redundant pairwise comparisons, which in turn make epistemic progress harder to achieve and increase the likelihood of entering a redundancy-induced BTR.

## D.3  T3 ON PE-LIKE TASKS WITHOUT ACCESS TO THE GROUND TRUTH

The proxy rule for the PE/MR task described in Sec. 3.1 relies on the ground-truth preference vector $v^\star$. However, the truncation mechanism does *not* require access to the ground-truth. Instead, we employ a fully belief-driven truncation rule that relies solely on the agent's internal preference estimates. Let

$\hat{v}_t$ denote the model's predicted preference vector at round $t$. We define an epistemic-stalling signal via a $k$-step moving average of update magnitudes:

$$\text{stall}_t = \mathbb{I}\left[ \left( \frac{1}{k} \sum_{j=t-k}^{t-1} \|\hat{v}_{j+1} - \hat{v}_j\|_2 \right) < \varepsilon \right], \tag{62}$$

where $k$ is the sliding-window length and $\varepsilon$ is a truncation threshold. The threshold is obtained from the empirical distribution of the $k$-step moving-average updates $\bar{\Delta}_t^{(k)}$ computed from offline rollouts. Specifically, $\varepsilon$ is set to a chosen quantile (e.g., 60%, 75%, 85%) of this distribution, ensuring that the criterion is *entirely ground-truth-free*. A trajectory is truncated once Eq. 62 is triggered, *i.e,* the agent's belief updates become small for consecutive steps, indicating epistemic stalling.

Table 7 summarizes the results on the PE dataset. Despite the absence of oracle information, the belief-based truncation retains strong performance, closely matching or surpassing the oracle-based $\mathbf{T^3}$ reported in the main paper.

Table 7: Performance of $\mathbf{T^3}$ on the PE task without access to $v^\star$. Thresholds $\varepsilon$ correspond to quantiles of offline $\bar{\Delta}_t^{(k)}$ statistics. BinarySim accuracy is reported for Qwen-2.5-7B-Instruct trained with PPO. `vanilla` and `T3-gt` represent vanilla-PPO and $\mathbf{T^3}$ in the main text (with access to the ground-truth $v^\star$), respectively.

| Quantile | 60% | 75% | 85% | vanilla | T3-gt |
|---|---|---|---|---|---|
| $\varepsilon$ | 0.18 | 0.28 | 0.36 | – | – |
| BinarySim | 44.33 | 50.67 | 49.00 | 42.00 | 49.00 |

### D.4 Exploration of adaptive T3 truncation rule

**Adaptive $\mathbf{T^3}$ via online threshold selection.** Motivated by extending $\mathbf{T^3}$ beyond fixed, offline-chosen thresholds, we further investigate an adaptive variant in which the truncation threshold evolves alongside the policy. For the PE task, the belief-based stalling criterion is employed the same as Appendix D.3 and Eq. 62 with $k = 4$. To obtain $\varepsilon$ adaptively, every 6 training steps we collect a batch of fully untruncated rollouts under the current policy and compute the empirical distribution of the $k$-step moving-average update magnitudes $\bar{\Delta}_t^{(k)}$. The threshold is then updated according to a fixed quantile $\alpha$ of this distribution:

$$\varepsilon \leftarrow \text{Quantile}_\alpha \left( \{ \bar{\Delta}_t^{(k)} \}_{\text{online}} \right).$$

This mechanism yields a dynamically adjusted truncation threshold that tracks the scale of the model's ongoing belief updates.

Table 8 reports the performance across quantiles $\alpha$. The results exhibit non-monotonic dependence on $\alpha$. Notably, at $\alpha = 0.6$, the adaptive variant achieves a substantial improvement, outperforming both the PPO baseline and the oracle-based $\mathbf{T^3}$ result reported in the main text. These results highlight the potential for extending the $\mathbf{T^3}$ principle to adaptive thresholding, and we leave a more in-depth exploration to future work.

Table 8: Adaptive $\mathbf{T^3}$ on the PE dataset. The threshold $\varepsilon$ is updated online from the $\alpha$-quantile of the current $\bar{\Delta}_t^{(k)}$ distribution.

| $\alpha$ | 20% | 40% | 60% | 80% | 90% | vanilla | T3-gt |
|---|---|---|---|---|---|---|---|
| BinarySim | 43.67 | 44.33 | **60.33** | 43.67 | 39.67 | 42.00 | 49.00 |

# E  POTENTIAL FUTURE WORK

## E.1  MORE GENERAL-PURPOSE PROXY DESIGN

**Task-agnostic surrogate signals for epistemic stalling.** In main experiments, since the structure of hypothesis spaces and notions of progress differ across tasks, instantiating $\mathbf{T^3}$ naturally leverages *task-level structure* to define observable proxies that track epistemic progress. However, guided by the $\mathbf{T^3}$ principle, we can further reduce the reliance on task-specific knowledge via utilizing *general-purpose* truncation detectors. We explore two broad, task-agnostic families of surrogate signals as follows.

**(i) Semantic redundancy signals.** In multi-turn LLM-agent settings, epistemic stalling frequently manifests as semantic redundancy, where the model repeatedly issues circular queries or revisits previously resolved informational subgoals, as shown in prior studies (Zhou et al., 2025; Yuan et al., 2025). Such redundancy is often detectable via embedding similarity, clustering, *etc*.

Building on this intuition, we have several successful explorations in this direction: *i)* In the SP task, the truncation based on question-semantic similarity (*cf.*, Sec. 3.3.3) yields consistent performance gains. *ii)* Moreover, for tasks with continuous latent spaces, such as the PE task, tracking the convergence of the model's internal preference vector estimate provides an effective proxy for redundancy: truncation is triggered when the estimate ceases to change meaningfully (*cf.*, Appendix D.3 and D.4). This convergence reflects an epistemic "stall" analogous to query redundancy in dialog scenarios such as the SP. Our experiments show the effectiveness of these redundancy-based surrogates.

**(ii) Internal state signals.** Recent empirical analyses suggest that hidden representations of Transformer and LLM models could encode intermediate judgment or reasoning states (Lu et al., 2025; Zhou et al., 2024). Although the precise hidden-state signatures corresponding to epistemic stalling remain an open question, characterizing such patterns (*e.g.*, consecutive high similarity of hidden states) is a promising direction for future work. Such signals may be especially valuable in open-domain tasks where a structured hypothesis space is not readily defined.

# F  SETUP DETAILS

## F.1  DATASET DETAILS AND PROMPT TEMPLATES

In this section, we present more details for the datasets and tasks evaluated in this work. See dataset statistics in Table 9.

**SituationPuzzles (SP).** This task introduces a challenging active reasoning task where the LLM player must uncover a coherent narrative from an initially puzzling scenario. Each puzzle begins with a brief, paradoxical statement. The solver interacts iteratively with a judge by asking binary yes-no questions, gathering feedback from the judge to constrain the solution space. The goal is to formulate a complete and plausible explanation that resolves the apparent contradiction. We directly use this dataset from the AR-Bench (Zhou et al., 2025). In our experiments, we utilize a Qwen2.5-14B-Instruct model to provide the interactive feedback.

The prompt template for the SituationPuzzles dataset can be seen in Fig. 11. For SituationPuzzles, put a specific puzzle to solve into {puzzle} of the prompt. The prompt template for the judge LLM is shown in Fig. 13. The judge will receive {surface} and {bottom} to understand the whole puzzle, and give yes-no feedback according to the player LLM's question.

**GuessNumbers (GN).** Adapted from the original dataset proposed by AR-Bench (Zhou et al., 2025) which the player must crack a 4-digit secret (digits are unique in 0-9), our newly constructed $\text{GN}(a, b)$ is a series of reasoning tasks that involve the LLM agent's interactive deduction with external sources: the target is a $a$-digit number, where each digit is sampled from a set of $b$ unique symbols without repetition. This yields $P(b, a) = b!/(b-a)!$ possible targets.

At each step, the LLM agent makes a guess and receives structured feedback in the form of xAyB, where $x$ denotes the number of digits that are both correct in value and position (denoted as "A"), and $y$ denotes the number of digits that are correct in value but placed in the wrong position (denoted as

"B"). The agent is expected to actively perform reasoning based on accumulated observations and interact with an external source to efficiently reduce uncertainty and locate the correct answer.

To control for randomness in the first move, which plays a minor role in evaluating the LLM agent's ability to understand and update based on observations, we fix the first guess to a deterministic number that differs from the answer. This means we need $(a, b, g_0, x_0, y_0)$ to specify a question for the LLM player, where $g_0$ denotes the initial guess, and $(x_0, y_0)$ denotes the corresponding initial feedback of the form $\texttt{x}_0\texttt{Ay}_0\texttt{B}$.

We group data items by their tuple $(a, b, x_0, y_0)$, since items sharing the same $(a, b, x_0, y_0)$ correspond to tasks with similar uncertainty reduction dynamics and reasoning logic patterns. Specifically, our constructed dataset covers all data items of the following sub-groups: $(3, 4, 0, 3)$, $(3, 4, 2, 0)$, $(3, 4, 1, 2)$, $(3, 5, 1, 2)$, $(3, 5, 0, 3)$, $(3, 5, 1, 0)$, $(3, 5, 2, 0)$, $(4, 4, 0, 4)$, and $(4, 5, 3, 0)$. These configurations are carefully selected to ensure diversity in task complexity: varying $(a, b)$ controls the size of the hypothesis space, while varying $(x_0, y_0)$ shapes the initial reasoning landscape by introducing distinct patterns of partial evidence. Finally, we perform a randomized train-test split over the obtained set for training and evaluation.

The prompt template for the GuessNumbers dataset can be seen in Fig. 12. For GuessNumbers, we need to first specify $\{\texttt{num\_digits}\}$ and $\{\texttt{num\_uniques}\}$, corresponding to $(a, b)$ mentioned above, and then specify the initial guess in $\{\texttt{initial\_guess}\}$, and the resulting initial feedback in $\{\texttt{initial\_feedback\_same\_pos}\}$ and $\{\texttt{initial\_feedback\_diff\_pos}\}$.

**CircuitDecoding (CD).** Adapted from Badola et al. (2025), in this dataset, each instance presents a collection of unknown Boolean circuits, each taking a fixed number of binary inputs and producing a binary output. There are a set of ground-truth circuits which are drawn from a finite candidate set of logical structures, and the player must identify which candidates correspond to the hidden circuits. To achieve this, the solver engages in a multi-turn interaction protocol: at each turn, the player must query one circuit with a binary input configuration of their choice, and receives the corresponding output. These queries serve as informative probes, allowing the player to iteratively eliminate inconsistent candidates and refine their hypotheses. The task requires strategic planning to maximize information gain under limited query budgets, and finally the solver must output the candidate indices of all underlying circuits. In our experiments, we adopt the prompt template shown in Fig. 10, where the LLM solver aims to figure out $\{\texttt{num circuits}\}$ hidden ground-truth circuits from $\{\texttt{num candidates}\}$ candidates specified as: $\{\texttt{candidate\_list\_str}\}$.

**PreferenceEstimation (PE).** Adapted from Badola et al. (2025), this dataset targets the problem of interactive preference elicitation, where the agent must infer a latent user preference vector governing utility over movies. Specifically, each movie is associated with a list of attribute scores $(s_1, \cdots, s_n)$, where $n$ is the total dimensions of attributes. In this task, the user evaluates a movie as a weighted sum of its attribute scores $\sum_{i=1}^{n} w_i^{\star} s_i$, with the weights $(w_1^{\star}, \cdots, w_n^{\star})$ forming the hidden preference vector to be discovered. At the beginning of an interaction episode, the agent is presented with a set of reference movies annotated by their attribute values. At each round, the agent outputs both its current vector guess and a pairwise comparison query between two reference movies. The user provides feedback ("Yes", "No", or "Equal") according to the weighted sum scores of the two mentioned movies. Through multiple turns, the agent iteratively updates its estimate of the preference vector by reasoning over past user feedback.

The prompt template for the PreferenceEstimation dataset is illustrated in Fig. 14. The LLM player is given $\{\texttt{len\_seen}\}$ reference movies for raising pairwise questions, to iteratively refine its guess on the $\{\texttt{len\_attributes}\}$-dimensional hidden user preference vector.

**MovieRecommendation (MR).** Building upon the preference estimation setup, this dataset further evaluates the generalization ability of an agent's inferred user model. After completing several rounds of interaction as mentioned in the PE task, the agent is tasked with recommending from a set of unseen movies. Each unseen movie is described by the same attribute dimensions, but the agent has not encountered them during training or interaction. In the final turn, the agent applies its preference vector guess to score each candidate unseen movie, and is required to select the movie that the user is most likely to prefer as its recommendation. This task thus demands transferring preference inference to out-of-distribution recommendation, and evaluates reasoning consistency, robustness, and generalization in interactive recommender systems.

The prompt template for this task is shown in Fig. 15. The agent is expected to leverage its estimated preference vector to make a personalized recommendation from {unseen_movie_list}.

Table 9: Dataset Statistics in this work.

|  | Train | Test |
|---|---|---|
| SituationPuzzles (SP) | 400 | 100 |
| GuessNumbers (GN) | 1526 | 382 |
| CircuitDecoding (CD) | 1000 | 300 |
| PreferenceEstimation (PE) | 700 | 300 |
| MovieRecommendation (MR) | 700 | 300 |

## F.2 BASELINE DETAILS

Here we introduce RL algorithms used in our experiments. Formally, given an actor model $\pi_\theta$, the likelihood of a response $y$ to a query $x$ under the policy $\pi_\theta$ is modeled as $\pi_\theta(y|x) = \prod_{t=1}^{|y|} \pi_\theta(y_t|x, y_{<t})$. Given a query-response pair $(x, y)$, a verifier $r$ generates its reward $r(x, y) \in [0, 1]$.

**Proximal Policy Optimization (PPO)** (Schulman et al., 2017) employs the following objective for policy optimization:

$$\mathcal{J}_{\text{PPO}}(\theta) = \mathbb{E}_{x \sim \mathcal{D}, \, y \sim \pi_{\theta_{\text{old}}}(\cdot|x)} \left[ \frac{1}{|y|} \sum_{t=1}^{|y|} \min \left( w_t(\theta)\widehat{A}_t, \, \text{clip}\left(w_t(\theta), 1 - \varepsilon, 1 + \varepsilon\right) \widehat{A}_t \right) \right], \quad (63)$$

where the importance ratio of the token $y_t$ is defined as $w_t(\theta) = \frac{\pi_\theta(y_t|x, y_{<t})}{\pi_{\theta_{\text{old}}}(y_t|x, y_{<t})}$, the advantage $\widehat{A}_t$ of $y_t$ is typically computed via Generalized Advantage Estimation (GAE) (Schulman et al., 2015) with temporal-difference errors, and $\varepsilon$ is the clipping range of importance ratios.

**Group Relative Policy Optimization (GRPO)** (Shao et al., 2024) proposes computing the relative advantage of each response within a group of responses of the same query using the following objective (omitting the KL regularization term):

$$\mathcal{J}_{\text{GRPO}}(\theta) = \mathbb{E}_{x, \, \{y_i\}_{i=1}^G} \left[ \frac{1}{G} \sum_{i=1}^{G} \frac{1}{|y_i|} \sum_{t=1}^{|y_i|} \min \left( w_{i,t}(\theta)\widehat{A}_{i,t}, \, \text{clip}\left(w_{i,t}(\theta), 1 - \varepsilon, 1 + \varepsilon\right) \widehat{A}_{i,t} \right) \right],$$
$$(64)$$

where $\{y_i\}_{i=1}^G \sim \pi_{\theta_{\text{old}}}(\cdot|x)$ and $G$ is the group size. The importance ratio $w_{i,t}(\theta)$ and advantage $\widehat{A}_{i,t}$ of token $y_{i,t}$ are defined as:

$$w_{i,t}(\theta) = \frac{\pi_\theta(y_{i,t}|x, y_{i,<t})}{\pi_{\theta_{\text{old}}}(y_{i,t}|x, y_{i,<t})}, \quad \widehat{A}_{i,t} = \frac{r(x, y_i) - \text{mean}\left(\{r(x, y_i)\}_{i=1}^G\right)}{\text{std}\left(\{r(x, y_i)\}_{i=1}^G\right)}, \quad (65)$$

respectively, where all the tokens in $y_i$ share the same advantage.

**Group Sequence Policy Optimization (GSPO)** (Zheng et al., 2025) extends GRPO by defining the importance ratio at the sequence level with length normalization, with sequence-level clipping, rewarding, and optimization. The objective is:

$$\mathcal{J}_{\text{GSPO}}(\theta) = \mathbb{E}_{x, \, \{y_i\}_{i=1}^G} \left[ \frac{1}{G} \sum_{i=1}^{G} \min \left( s_i(\theta)\widehat{A}_i, \, \text{clip}(s_i(\theta), 1 - \varepsilon, 1 + \varepsilon)\widehat{A}_i \right) \right], \quad (66)$$

where

$$s_i(\theta) = \left( \frac{\pi_\theta(y_i|x)}{\pi_{\theta_{\text{old}}}(y_i|x)} \right)^{1/|y_i|} = \exp \left( \frac{1}{|y_i|} \sum_{t=1}^{|y_i|} \log \frac{\pi_\theta(y_{i,t}|x, y_{i,<t})}{\pi_{\theta_{\text{old}}}(y_{i,t}|x, y_{i,<t})} \right).$$

---

**Input Prompts for the CircuitDecoding dataset**

Welcome to the Circuit Deduction Challenge!

## The Setup:
- There are `{num_circuits}` circuits, labeled `{circuit_labels}`.
- Each circuit accepts `{num_inputs}` binary inputs (0 or 1) and produces a single binary output (0 or 1).
- Each circuit is drawn from a fixed candidate list of `{num_candidates}` possible logical structures, each associated with an index: `{candidate_list_str}`

## Your Goal:
Identify which circuits from the candidate list correspond exactly to circuits `{circuit_labels}`.

## How to Play:
You can interact with me for several turns to determine the true underlying circuits:
1. At each turn, query one circuit with any binary input of your choice.
2. Use the specified format for your query. For example, to query circuit A with inputs `x0=1, x1=0, x2=1`, ask: `<interact>A(1, 0, 1)</interact>`.
3. You must make only one query at each turn. I will return the binary output for that circuit on the given input.
4. Ask strategic queries that maximize information gain. Your goal is to minimize the number of turns by leveraging the feedback at each step to narrow down the candidate possibilities.

## Final Submission:
Once you are confident, submit your final answer by providing the indices of the identified circuits from the candidate list inside `<answer>` and `</answer>`. For example, if A corresponds to candidate 13 and B corresponds to 6, your answer must be: `<answer>13, 6</answer>`.
Please start with your first query.

Figure 10: Prompt Template for CircuitDecoding.

## F.3 SUPPLEMENTARY IMPLEMENTATION DETAILS

Here we provide additional implementation details. The maximum number of interaction turns is set at 10 for GuessNumbers, 15 for SituationPuzzles, 10 for CircuitDecoding, 10 for PreferenceEstimation, and 5 for MovieRecommendation. For RL training, we define task-specific rewards aligned with their evaluation metrics: for GuessNumbers, the reward is *Exact Match* (binary $\{0, 1\}$, given only at the final step); for SituationPuzzles, the reward is the *F1-word / character* score (continuous in $[0, 1]$, computed against the ground-truth answer); for CircuitDecoding and MovieRecommendation, the reward is also *Exact Match*; and for PreferenceEstimation, the reward is *Binary Similarity* between the predicted and ground-truth preference vectors. All rewards are provided only at the terminal step of each trajectory, consistent with the outcome-based RL setting.

Training for GuessNumbers and SituationPuzzles is conducted on a single node equipped with 8 H100 GPUs, while CircuitDecoding and PreferenceEstimation/MovieRecommendation are trained on a single node with 8 B200 GPUs, based on the implementations of Verl (Sheng et al., 2025). All training tasks are conducted for 200 steps with the actor model optimized using a learning rate of $1.0 \times 10^{-6}$. For distributed training, we adopt Fully Sharded Data Parallelism (FSDP), using BFloat16 precision throughout both training and evaluation. For efficient LLM rollouts, we adopt vLLM [†] with a tensor parallel size of 1. The rollout sampling uses a temperature of 1.0 for SituationPuzzles and 0.6 for GuessNumbers, and a top-p value of 0.95 for all datasets.

For the PPO baseline, we use Generalized Advantage Estimation (GAE) with parameters $\lambda = 1$ and $\gamma = 1$. The KL divergence regularization coefficient $\beta$ and clip ratio $\varepsilon$ are set to 0.001 and 0.2. For GRPO training, we sample 5 responses per prompt, and the rollout parameters, KL divergence coefficient, and the clip ratio are consistent with the PPO setting. For the GSPO algorithm, we do not use the KL divergence constraint, and the clip ratio $\varepsilon_{low}$ and $\varepsilon_{high}$ are set to 0.0003 and 0.0004, respectively, while others keep consistent with GRPO training.

---

[†] https://docs.vllm.ai/en/latest/

---

**Input Prompts for the SituationPuzzles dataset**

Let's play a situation puzzle game. I'll give you a puzzle. You can interact with me for several turns during the question phase to reach the final answer. For each turn, you will:
- Review all previous questions and feedback.
- Ask me a yes-or-no question inside `<interact>` and `</interact>`.
- I will answer your latest question with "Yes", "No", or "Unknown".
- Repeat the process until you are confident in the answer.
If you believe you have confidently determined the correct solution, present your answer inside `<answer>` and `</answer>`.

Now, here's the puzzle:
Puzzle: {`puzzle`}

---

Figure 11: Prompt Template for SituationPuzzles.

---

**Input Prompts for the GuessNumbers dataset**

Let's play a number guessing game. The rules are as follows: I have a secret {`num_digits`}-digit number in mind, composed of digits from 1 to {`num_uniques`}, with no repeated digits. You will take turns guessing the number, using feedback after each guess to progressively narrow down the possibilities.
For each turn, you will:
- Review all previous guesses and feedback.
- Think through your reasoning process inside `<think>` and `</think>`. The reasoning should show how your belief about the secret number evolves based on the accumulated evidence.
- Make a strategic guess inside `<interact>` and `</interact>`, based on your current belief.
- Receive feedback of your latest guess describing: how many digits are present in the answer and in the correct positions, and how many digits are present in the answer but in the different positions.
- Repeat the process until you are confident in the answer. If you believe you have confidently found the correct number, present your answer inside `<answer>` and `</answer>`.
Game start. Now it is your turn:

`<think>`No prior knowledge. Start with a random guess that covers diverse digits to gather information.`</think>`
`<interact>`{`initial_guess`}`</interact>`

The feedback of your latest guess: {`initial_feedback_same_pos`} digits are present in the answer and in the correct positions, {`initial_feedback_diff_pos`} digits are present in the answer but in the different positions.
Now it is your turn:

---

Figure 12: Prompt Template for GuessNumbers.

---

**Input Prompts for the Judge LLM in the SituationPuzzles dataset**

You are the referee of a game where players are shown a `<Surface>` and you are given the `<Bottom>`. You need to understand the entire story based on both the `<Surface>` and `<Bottom>`. Players will ask questions based on the `<Surface>`, and you need to judge whether their guesses are correct. Please strictly adhere to answering with only three specified responses: Yes, No, or Unknown, without any explanation.

## Judging Rules
- If the player's question matches the given `<Surface>` and `<Bottom>`: Please only answer "Yes" without any explanation.
- If the player's question contradicts the given story: Please only answer "No" without any explanation.
- If the answer to the player's question cannot be found in the `<Surface>` and `<Bottom>`, and cannot be deduced through reasoning: Please only answer "Unknown" without any explanation.
- If the player directly ask for the answer, please only answer "This is not a question, please propose your next question."
- If the player does not propose a question or question that not for solve the puzzle, please only answer "This is not a question, please propose your next question."

## Important Notes
1. Fully understand the cause, process, and outcome of the entire story, and make logical inferences.
2. If a conclusion cannot be drawn from the provided story or through reasonable inference, answer "Unknown".
3. Strictly adhere to answering with only the three specified responses: Yes, No, or Unknown. Do not provide additional explanations.
4. Carefully check whether the player ask for the answer, if a player do so, please only answer "This is not a question, please propose your next question."

## Examples
### Example 1: The Hiccuping Man
`<Surface>`
A man walks into a bar and asks the bartender for a glass of water. The bartender suddenly pulls out a gun and points it at him. The man smiles and says, "Thank you!" then calmly leaves. What happened?
`<Bottom>`
The man had hiccups and wanted a glass of water to cure them. The bartender realized this and chose to scare him with a gun. The man's hiccups disappeared due to the sudden shock, so he sincerely thanked the bartender before leaving.
Possible questions and corresponding answers:
Q: Does the man have a chronic illness? A: Unknown
Q: Was the man scared away? A: No
Q: Did the bartender want to kill the man? A: No
Q: Did the bartender intend to scare the man? A: Yes
Q: Did the man sincerely thank the bartender? A: Yes

## Question Content
### `<Surface>`
{surface}
### `<Bottom>`
{bottom}
Now, please judge the following player question:
{question}
Answer with only one of the three specified responses: Yes, No, or Unknown, without any explanation.

Figure 13: Prompt Template for the Judge LLM in SituationPuzzles.

---

**Input Prompts for the PreferenceEstimation task**

You are a movie recommendation agent. Your goal is to infer the hidden user preference vector (w1,...,w{len_attributes}) through interaction.

## Setup:
- You are given {len_seen} movies with scores on {len_attributes} attributes (indexed 1...{len_attributes}):
{seen_movie_sample}
- User satisfaction = w1*attr1 + ... + w{len_attributes}*attr{len_attributes}, where each wi in [0, 1]. The user always answers consistently.

## Interaction Rules (per round):
1. Reflect on all past feedback and reason about how it changes your estimate of the preference vector.
   - Think about which attributes gained or lost importance.
   - Adjust your estimate strategically.
2. Output both your updated guess and a new pairwise query in the exact format:
`<interact>`
Guess: w1,w2,...
Question: Would you prefer `option_1` over `option_2`?
`</interact>`
   - Guess must be comma-separated numbers in [0,1].
   - `option_1` and `option_2` must be movie names only.
The user replies with one of: "Yes" (prefer `option_1`), "No" (prefer `option_2`), or "Equal".

## Final Stage:
Once you are confident about the user preference after several turns, output your final preference vector as:
`<answer>w1,w2,...,w{len_attributes}</answer>`
Please Start with your first `<interact>` block.

Figure 14: Prompt Template for PreferenceEstimation.

---

**Input Prompts for the MovieRecommendation task**

Final Turn: Now you have reached the last turn. Instead of asking a new question, use your most recent preference guess to score the following unseen movies and recommend the best one.
{unseen_movie_list}

Here is an example of how to proceed:
Preference vector (guess): 0.2,0.7,0.5
Example Unseen movies:
Movie_A: [0.6,1.0,0.8]
Movie_B: [1.2,0.3,0.4]
Movie_C: [0.5,0.8,0.9]
Scoring:
Movie_A = 0.2*0.6 + 0.7*1.0 + 0.5*0.8 = 1.22
Movie_B = 0.2*1.2 + 0.7*0.3 + 0.5*0.4 = 0.65
Movie_C = 0.2*0.5 + 0.7*0.8 + 0.5*0.9 = 1.11
Best = Movie_A
`<answer>Movie_A</answer>`

Your goal:
Now do the same with your own latest preference vector and the given unseen movies. After scoring, return the final answer enclosed within `<answer>` and `</answer>`. The answer must be exactly one of the unseen movie names.

Figure 15: Prompt Template for MovieRecommendation.

