# OpenReview forum: "Reducing Belief Deviation in Reinforcement Learning for Active Reasoning of LLM Agents"
_ICLR.cc/2026/Conference — ICLR 2026 Oral_

### Official Review · Reviewer_4y4o · 2025-10-14

**Soundness:** 3
**Presentation:** 3
**Contribution:** 2
**Rating:** 8
**Confidence:** 3

**Summary:**

This paper proposes a novel method, T3 (Truncating Belief-Trapped Trajectories), to mitigate belief deviation in LLM-based agents during multi-turn active reasoning under reinforcement learning (RL). The core idea is to detect when an agent enters a “Belief Trap Region” (BTR) where further reasoning becomes uninformative and truncate the trajectory early to preserve credit assignment for the earlier informative steps. Theoretical analysis establishes the adverse effect of BTR on gradient estimation, and experiments on five tasks (from AR-Bench and Multi-Turn Puzzles) demonstrate significant gains in stability, efficiency, and performance across multiple RL methods (PPO, GRPO, GSPO).

**Strengths:**

Novel Theoretical Insight: Identifies belief deviation and BTR as key bottlenecks in multi-turn RL for reasoning.

Simple & Effective Method: T3 integrates seamlessly into existing RL algorithms with minimal changes.

Rigorous Theory: Theorems formally show how uninformative trajectory tails can invert gradients and harm exploration.

Impressive Results: Up to 30% performance gain, and 34% reduction in token usage across various benchmarks.

Robustness: T3 generalizes well to OOD settings, LLM scales (3B–14B), and architectures (Qwen, LLaMA, DeepSeek).

**Weaknesses:**

(W1) Truncation proxy design remains task-specific and heuristic :While the theoretical formulation of T3 is grounded in the notion of epistemic stalling (i.e., halted belief progress), the practical implementation relies on task-specific heuristic proxies such as:
repeated “unknown” feedbacks in Situation Puzzles,
invalid guesses outside the hypothesis set in Guess Numbers,
similarity drops in Movie Recommendation.
These conditions work well for the selected benchmarks, but lack generality. In tasks with ill-defined or open-ended hypothesis spaces, such proxies may fail to accurately detect BTR (Belief Trap Region) entry, leading to either missed truncation or premature stopping.
my suggestion: The authors could explore more general-purpose or learnable truncation detectors, such as classifiers over internal LLM hidden states or CoT consistency metrics, to broaden T3’s applicability across domains
 (W2) Limited to sparse outcome-based reward setting:
T3 is explicitly designed for environments where only the final step yields a non-zero reward. While this aligns with many current RLHF-style settings, it raises concerns regarding:tasks with dense or shaped intermediate rewards, environments where step-wise feedback is available or desirable..
In such scenarios, truncating trajectories could remove valid informative signals from later steps, inadvertently harming credit propagation rather than helping it. T3’s compatibility with these more general reward formulations remains unexplored. Suggestion: Discuss how T3 would behave in dense-reward or hybrid reward settings, or propose adaptations for preserving mid-trajectory feedback signals.
(W4) Lack of analysis on false positive truncation and its impact.While the authors provide some ablations on hyperparameters (e.g., window size k), they do not explicitly model the cost of mis-triggered truncation or propose mitigation strategies.
What if the truncation proxy fires prematurely (false positive)?
Could this discard useful exploratory steps that would have led to reward?
Does it increase variance in gradient estimation during early-stage training?

**Questions:**

How does T3 behave under dense reward or shaped intermediate feedback?
Have you considered extending T3 beyond outcome-only reward settings?

Can the proxy condition for BTR be learned, rather than hand-crafted?
For instance, via classification over hidden states or CoT traces?

What is the impact of false-positive truncation in early training stages?
Does it increase gradient variance or harm exploration?

---

> ### Author Response · Authors · 2025-11-23
>
> We appreciate reviewer 4y4o for acknowledging the theoretical contributions and strong empirical performance of our approach. We also thank you for the insightful suggestions, to which we respond below.
>
> # Weaknesses & Questions
> ## WQ1. Task-specific proxies; suggestion of exploring more general-purpose / learnable truncation detectors
> ### 1.1 Task-specific proxies
> We appreciate the reviewer’s insightful comments. We agree that our current experiments instantiate different observable proxies for different tasks. Our intention in this paper is not to propose a single universal proxy, but a **general principle: T3 truncates when epistemic progress stalls.** For LLMs, the latent belief state is not directly accessible, and it is therefore impossible to pinpoint BTR entry exactly. The implementation of T3 accordingly relies on observable surrogates of epistemic progress, which naturally vary with the task’s available interaction signals rather than with the T3 principle itself.
>
> More specifically, T3 requires a representation of the hypothesis state ($\mathcal{H}$), a criterion for epistemic progress $d(\mathcal{H}\_\tau,\mathcal{H}\_{\tau+1})<\Delta_{\min}$, and a patience hyperparameter $k$ for truncation. In practice, since the structure of hypothesis spaces and notions of progress differ across tasks, obtaining these components naturally relies on **task-level meta-knowledge** about what observable signals best reflect changes in $\mathcal{H}$. From the five diverse tasks in this work, it could be relatively easy to find the surrogates for applying T3 and obtain significant improvements.
>
> ### 1.2 suggestion of exploring more general-purpose / learnable detectors
> We agree that extending T3 with more general-purpose detectors (e.g., hidden-state classifiers, CoT-consistency signals, or state-entropy measures) is a promising direction, especially for tasks with unstructured or open-ended hypothesis spaces, where it is challenging to characterize and establish criteria for $\mathcal{H}$. **In practice, several task-agnostic families of detectors could further relieve the reliance on task-specific knowledge  while still indicating the entry into a belief-trap region:**
>
> **i) Semantic Redundancy Signals.** In LLM-agent systems, epistemic stalling often manifests as semantic redundancy: the model repeatedly asks questions that are paraphrastic variants of earlier ones or revisits previously explored informational subgoals, as reported in previous works [1][2]. Such redundancy can be captured through embedding similarity, query clustering, etc., which do not rely on a structured hypothesis space. We have **several successful explorations** as follows:
> - In the ablation studies in Sec. 3.3.3, we explored using Question Semantic Similarity as the truncation condition in the SP task, and the results also show improvements;
>
> - Moreover, in the response to W3 and W4 of the reviewer K1QM https://openreview.net/forum?id=r8hzDA3pUY&noteId=2UZZ2wTc0l, for tasks with continuous spaces like Preference Estimation, we experiment on leveraging redundancy-style signals: truncating when the preference vector estimated by the agent stops changing meaningfully; it reflects an epistemic “stall” analogous to semantic redundancy in the SP task. **We also obtained a remarkable performance gain, even largely surpassing the results reported in the original paper.**
>
> **ii) Internal State Signals.** Previous empirical studies demonstrate that hidden-state representations of Transformer and LLM models could encode intermediate judgment or reasoning states [3][4]. While the precise hidden-state signatures of epistemic stalling remain an open question, characterizing such patterns (e.g., consecutive high similarity of hidden states) is a promising  direction for future work. These internal consistency signals may provide a task-agnostic, model-internal indicator of belief trapping especially for open-ended tasks.
>
> We have added a discussion section to clarify this point and position these approaches as valuable future work.
>
> ### References in this part
>
> [1] Zhou, Zhanke, et al. "From Passive to Active Reasoning: Can Large Language Models Ask the Right Questions under Incomplete Information?." Forty-second International Conference on Machine Learning.
>
> [2] Yuan, Siyu, et al. "Agent-R: Training Language Model Agents to Reflect via Iterative Self-Training." arXiv preprint arXiv:2501.11425 (2025).
>
> [3] Lu, Wenquan, et al. "Latent Chain-of-Thought? Decoding the Depth-Recurrent Transformer." arXiv preprint arXiv:2507.02199 (2025).
>
> [4] Zhou, Zhenhong, et al. "How Alignment and Jailbreak Work: Explain LLM Safety through Intermediate Hidden States." EMNLP (Findings). 2024.

---

> ### Author Response · Authors · 2025-11-23
>
> ## WQ2. Limited to sparse outcome-based reward setting
>
> We thank the reviewer for this insightful observation. **Our theoretical analysis and the design of T3 focus on outcome-based sparse rewards, which is prevalent** in modern LLM reasoning settings, where intermediate steps do not provide environmentally grounded feedback (e.g., [1][2][3]).
>
> We agree that in dense-reward environments the situation becomes more nuanced. Truncation is beneficial when the tail carries no additional epistemic information, but if intermediate rewards encode meaningful signals, a truncation rule must be explicitly redesigned to preserve such feedback. **Extending T3 to these regimes systematically would therefore be non-trivial and require new theoretical machinery, since our theoretical analysis of the truncation mechanism fundamentally relies on the outcome-based reward structure.**
>
> In the refined manuscript, we have clarified this scope and added a discussion section on potential (non-rigorous) adaptations. **For example,** in dense-reward environments, a natural adaptation could be hybrid truncation: we may truncate the trajectory when epistemic stalling is detected, but replace the truncated tail with a bootstrapped value estimate. This preserves informative dense feedback while still preventing the late-stage credit contamination that T3 aims to avoid. Another potential way is epistemic-progress shaping, i.e., to incorporate the LLM’s epistemic progress as an auxiliary reward to align dense feedback with belief updates.
>
> We view this as a promising direction for future work.
>
> ### References
>
> [1] Guo, Daya, et al. "Deepseek-r1: Incentivizing reasoning capability in llms via reinforcement learning." arXiv preprint arXiv:2501.12948 (2025).
>
> [2] Jin, Bowen, et al. "Search-r1: Training llms to reason and leverage search engines with reinforcement learning." arXiv preprint arXiv:2503.09516 (2025).
>
> [3] Team, Tongyi DeepResearch, et al. "Tongyi DeepResearch Technical Report." arXiv preprint arXiv:2510.24701 (2025).

---

> ### Author Response · Authors · 2025-11-23
>
> ## WQ3. Lack of analysis on false positive truncation and its impact
>
> We thank the reviewer for highlighting the importance of analyzing false-positive truncation. T3 adopts observable surrogates of BTR to construct the truncation conditions, hence there tend to be limited false positive cases. Nevertheless, we agree that premature truncation (false positives) can, in principle, discard useful exploratory steps.
>
> **Analytically**, through the standard GAE decomposition,  the advantage of an early token $t$ aggregates future TD-errors $A_t=\sum_{u=t}^{T}(\gamma\lambda)^{u-t}\delta_u$. Our Theorem 2 characterizes the “uninformative tail” regime where the expected TD-errors become negative, so not truncating introduces a downward drift on $A_t$. A false-positive truncation corresponds to the opposite case: the trajectory has not yet entered the belief-trap region, and future $\delta_u$ are often non-negative on successful rollouts. Truncating prematurely therefore removes these positive contributions, reducing $A_t$ by discarding potentially useful exploratory steps that would have led to reward.
>
> **Diagnostic experiment.** To make the above analysis more concrete, we added a small diagnostic experiment. We fixed a trained vanilla-PPO policy and generated a set of rollouts. To ensure that the full trajectory contains informative signals and does not enter the belief-trap region, we retained only trajectories with final reward = 1. On these trajectories, we then simulated false-positive truncation:
> - with probability $\alpha$ the trajectory was forcibly truncated at the 3th turn (max allowed turn = 10);
> - and with probability $1-\alpha$ it proceeded normally.
> This creates a clean setting in which any observed degradation is attributable solely to premature truncation. For each early-stage token position $t=1:500$, we computed the mean GAE advantage across rollouts under each $\alpha.$
>
> **Results.** We have presented the results in CD and PE datasets here (**this figure:** https://postimg.cc/YjtjsYBh). As expected, more aggressive false-positive truncation
> systematically reduces the advantages of early exploratory actions, reflecting the removal of positive future return contributions.

---

### Official Review · Reviewer_DC2y · 2025-10-31

**Soundness:** 2
**Presentation:** 3
**Contribution:** 3
**Rating:** 8
**Confidence:** 4

**Summary:**

The paper presents a method to provide early stop during the RL if the hypothesis space did not shrink by a meaningful amount. The paper proposes $T^3$ (Truncating Belief-Trapped Trajectories), a mechanism that uses task-specific proxy signals to detect and halt trajectories entering the "Belief Trap Region (BTR)". Experiments across five diverse reasoning tasks demonstrate that $T^3$ consistently improves varied RL algorithms (PPO, GRPO, GSPO) in both final performance and token efficiency.

**Strengths:**

1. The paper provides a strong theoretical grounding for why RL fails in long-horizon active reasoning, formally characterizing the BTR and proving how it inverts expected advantage in GAE, thereby corrupting gradient estimates.
2. $T^3$ is a straightforward, drop-in enhancement for standard RL algorithms (PPO, GRPO, GSPO) that yields significant gains with small change.
3. The authors include thorough analyses, including out-of-distribution generalization tests (where $T^3$ maintains robustness) and ablations of different proxy truncation conditions, confirming the importance of detecting BTR entry rather than indiscriminate truncation.

**Weaknesses:**

1. Dependency on Task-Specific Proxies: While the theoretical $T^3$ condition is general, its practical implementation relies on manually designing task-specific proxies (e.g., repetitive queries for Situation Puzzles, failure to reduce hypothesis space for Guess Numbers)999999999. Finding these proxies for novel, less-structured tasks might be challenging.

**Questions:**

1. In Figure 3, the first and fourth figures have different y-axis label than the second and third figures, why?
2. If we want to extend this to more real world cases, how would $T^3$ fit in these use cases?

---

> ### Author Response · Authors · 2025-11-23
>
> We express our gratitude to the reviewer DC2y for recognizing the strong theoretical characterization and thorough empirical analysis of this work. In response to the valuable feedback received, we provide the following.
>
> # Weaknesses
> ## W1 Dependency on Task-Specific Proxies
> ### 1.1 `Dependency on Task-Specific Proxies`
> We appreciate the reviewer’s insightful comments. We agree that our current experiments instantiate different observable proxies for different tasks. Our intention in this paper is not to propose a single universal proxy, but a **general principle: T3 truncates when epistemic progress stalls**. For LLMs, the latent belief state is not directly accessible, and it is therefore impossible to pinpoint BTR entry exactly. The implementation of T3 accordingly relies on observable surrogates of epistemic progress, which naturally vary with the task’s available interaction signals rather than with the T3 principle itself.
>
> More specifically, T3 requires a representation of the hypothesis state ($\mathcal{H}$), a criterion for epistemic progress $d(\mathcal{H}\_\tau,\mathcal{H}\_{\tau+1})<\Delta_{\min}$, and a patience hyperparameter $k$ for truncation. In practice, since the structure of hypothesis spaces and notions of progress differ across tasks, obtaining these components naturally relies on **task-level meta-knowledge** about what observable signals best reflect changes in $\mathcal{H}$. From the five diverse tasks in this work, it could be relatively easy to find the surrogates for applying T3 and obtain significant improvement.
>
> ### 1.2 `Finding these proxies for novel, less-structured tasks might be challenging.` (discussed together with Q1 in Questions Section)
> We agree that surface signals of epistemic progress may be less explicit in less-structured tasks. However, as discussed above, T3 is fundamentally a meta-procedure: it only requires some observable surrogates of epistemic stalling, and does not need careful and sophisticated construction. As shown across five benchmarks, **any surrogate that meaningfully reflects whether belief updating has stalled can be plugged into T3**. In practice, several task-agnostic families of detectors could further relieve the reliance on task-specific knowledge  while still indicating the entry into a belief-trap region:
>
> **i) Semantic Redundancy Signals**. In LLM-agent systems, epistemic stalling often manifests as semantic redundancy: the model repeatedly asks questions that are paraphrastic variants of earlier ones or revisits previously explored informational subgoals, as reported in previous works [1][2]. **Such redundancy can be captured through embedding similarity, query clustering, etc., which do not rely on a structured hypothesis space. We have several successful explorations as follows:**
> - In the ablation studies in Sec. 3.3.3, we explored using Question Semantic Similarity as the truncation condition in the SP task, and the results also show improvements;
> - Moreover, in the response to W3 and W4 of the reviewer K1QM https://openreview.net/forum?id=r8hzDA3pUY&noteId=2UZZ2wTc0l, for tasks with continuous spaces like Preference Estimation, we experiment on leveraging redundancy-style signals: truncating when the preference vector estimated by the agent stops changing meaningfully; it reflects an epistemic “stall” analogous to semantic redundancy in the SP task. **We also obtained a remarkable performance gain, even largely surpassing the results reported in the original paper.**
>
> **ii) Internal State Signals**. Previous empirical studies demonstrate that hidden-state representations of Transformer and LLM models could encode intermediate judgment or reasoning states [3][4]. While the precise hidden-state signatures of epistemic stalling remain an open question, characterizing such patterns (e.g., consecutive high similarity of hidden states) is a promising  direction for future work. These internal consistency signals may provide a task-agnostic, model-internal indicator of belief trapping especially for open-ended tasks.
>
> We have incorporated the aforementioned discussion in our revised manuscript to further strengthen the discussion on the future extensions of our work.
>
> # Questions
> ## Q1 Typos of Figure 3
>
> Thank you for your careful observation. In Fig. 3a and 3d, the y-axis should read “Response length” and we have corrected this in the revision.
>
> ## Q2 How would T3 fit in more real-world cases?
> We thank the reviewer for raising this insightful question. Please refer to the second bullet of W1 in the Weakness Section, where we discuss T3 in real-world cases or less-structured tasks in detail.

---

> > ### Comment · Reviewer_DC2y · 2025-11-24
> >
> > Thank you for your information, will keep my rating

---

> > > ### Author Response · Authors · 2025-11-25
> > >
> > > Thank you, reviewer DC2y, for your insightful comments, prompt response, and continued positive evaluation of our work!

---

> ### Author Response · Authors · 2025-11-23
>
> ### References
>
> [1] Zhou, Zhanke, et al. "From Passive to Active Reasoning: Can Large Language Models Ask the Right Questions under Incomplete Information?." Forty-second International Conference on Machine Learning.
>
> [2] Yuan, Siyu, et al. "Agent-R: Training Language Model Agents to Reflect via Iterative Self-Training." arXiv preprint arXiv:2501.11425 (2025).
>
> [3] Lu, Wenquan, et al. "Latent Chain-of-Thought? Decoding the Depth-Recurrent Transformer." arXiv preprint arXiv:2507.02199 (2025).
>
> [4] Zhou, Zhenhong, et al. "How Alignment and Jailbreak Work: Explain LLM Safety through Intermediate Hidden States." EMNLP (Findings). 2024.

---

### Official Review · Reviewer_K1QM · 2025-10-31

**Soundness:** 2
**Presentation:** 3
**Contribution:** 3
**Rating:** 6
**Confidence:** 3

**Summary:**

The paper formalizes belief deviation in multi‑turn, active reasoning and defines a Belief‑Trap Region (BTR) where epistemic progress stalls. It proves that long, uninformative “tails” can flip the sign of early‑step advantages (advantage inversion), then proposes T³, an early‑truncation rule triggered by simple proxy tests of stalled progress in the reasoning trace. T³ is a drop‑in wrapper for PPO/GRPO/GSPO and shows consistent gains across five interactive tasks, smoother learning, and shorter responses.

**Strengths:**

Crisp failure‑mode lens. Clear POMDP framing with a truth‑anchored potential Ψ and a precise BTR definition; theory links tail length to advantage inversion (Theorem 2)

Simple, general mechanism. T³ uses observable stalling proxies (e.g., non‑shrinking hypothesis sets, repeated Unknown judgments) so it can integrate with standard RL without changing optimizers

The proposed method shows broad, consistent empirical gains. Improvements on CircuitDetection (CD), SituationPuzzles (SP), GuessNumbers (GN), PreferenceEstimation (PE), MovieRecommendation (MR) with 14/18 metric wins; OOD robustness; works across model sizes/types.

Stability & efficiency: Learning curves are smoother and responses shorter, implying fewer wasted tokens.

**Weaknesses:**

Theory–practice gap. Assumptions (e.g., value calibration to truth probability; linear update‑error growth) are strong and not empirically validated against measured beliefs/values.

Privileged proxies. PE/MR use oracle preference similarity to trigger truncation, limiting deployability without ground truth.
“Shorter is better” confound. Random truncation sometimes helps (Table 3), so more direct evidence of advantage inversion would strengthen the causal story.

Tuning guidance. Heuristics for window size/thresholds are task‑specific; no adaptive rule is provided.
Comparisons & scope. Limited real‑world tasks and reliance on an LLM judge in SP; baselines could be more apples‑to‑apples vs. open, strong models trained under similar budgets.

**Questions:**

No‑oracle proxies: How would you instantiate T³ on PE/MR‑like tasks without access to the true preference vector (e.g., entropy/disagreement/self‑consistency signals)?
Direct test of Theorem 2: Can you measure early‑token advantages with/without truncation and show the predicted sign flips as tail length grows?

Adaptive T³: Any online procedure to target a truncation‑ratio band (or validation return) and auto‑tune window/thresholds? Sensitivity to the SP judge identity/temperature?

Actionable feedback

Add a one‑page algorithm box for T³ (inputs, trigger, where truncation enters GAE/PPO).
Provide equal‑length ablations to disentangle truncation vs. better credit assignment; include γ/λ/KL sensitivity.
Add a small calibration check: value vs. correctness reliability plots. Report token & wall‑clock to fixed reward thresholds.

---

> ### Author Response · Authors · 2025-11-23
>
> We are grateful to reviewer K1QM for acknowledging the crisp failure‑mode lens, simple & general mechanism, and empirical gains of this work. We summarize your comments and provide our detailed responses below.
>
> # Weaknesses & Questions
> ## 1. Theory–practice gap: Assumptions not empirically validated
> ### 1.1 Assumption 1 – empirical verification
>
> We need to note that a direct empirical validation is challenging because both the oracle Bayesian update $B^\star$ and the agent’s internal belief $b_t$ are not directly accessible in LLMs. To address this concern, we designed an experiment on the PE task that allows us to construct practical and theoretically aligned approximations of the quantities involved in Assumption 1.
>
> **i) Approximating the potential $\Psi$.** In PE, each interaction round provides the model’s explicit estimate of the latent preference vector, denoted by $w_t$. Since the ground-truth preference vector $w^\star$ is known, we define: $d(w_t) := \Vert w_t - w^\star\Vert_2^2,$
> and use $d(w_t)$ as an observable proxy for the task potential: $\hat \Psi_t:=d(w_t)\approx \Psi(b_t).$ This proxy retains the key properties of our potential function: it is non-negative, and equals zero iff. task completion.
>
> **ii) Approximating the oracle Bayesian update $B^\star$.** While the true Bayesian posterior over the latent user preference is inaccessible, we can construct a standard preference-update rule $\hat B$ based on traditional machine learning. Given the model’s query $a_t$ and the feedback $o_t$, this update produces:
> $$\hat{w}\_{t+1} := \hat{B}(w_t, a_t, o_t),$$
> which we use as a principled surrogate for the oracle update $B^\star$. More specifically, we leverage:
> $$\hat w_{t+1}= w_t + K_t m_t (o_t -m_t^\top w_t),
>  \qquad
>  K_t = \frac{\sigma_0^2}{\sigma_0^2 \Vert m_t\Vert_2^2 + \sigma^2}.$$
> Here, $m_t$ is the movie-attribute difference vector for the pair of movies selected by the LLM’s query, i.e., $m_t = \mathrm{attr}(A)-\mathrm{attr}(B)$. The binary observation $o_t \in \lbrace-1,+1\rbrace$ represents the user’s feedback to this query and is given by $o_t = \operatorname{sign}(m_t^\top w^\star)$. The parameters $\sigma_0^2$ and $\sigma^2$ denote the prior and observation noise variances, respectively. We set them to $1.0$.
>
> In contrast, the LLM’s own update under parameters $\theta$ yields:
> $w_{t+1} := B_\theta(w_t, a_t, o_t),$
> as the LLM-generated estimate at the next step.
>
>
> **iii) Constructing observable samples of the update-error quantity.** We empirically instantiate the update error term in Eq. 2 of the paper: $$\hat c\_\theta(b_t):= d(w\_{t+1}) - d(\hat{w}\_{t+1})\approx c_\theta (b_t).$$
>
> We collect over **150k** data points by rollouts generated by the Qwen-2.5 series models, obtaining samples of $\lbrace(\hat \Psi_t, \hat c_\theta(b_t) )\rbrace$. Plotting these pairs allows us to examine Asp1.
>
> **iv) Visualization and Estimation of $m_\theta, U_0, c_0$.**  While individual samples exhibit variance (as expected for high-dimensional belief updates), Asp 1 concerns only the lower bound of this relationship. Therefore, our goal is to estimate the lower-envelope trend of the obtained sample points. We follow a principled and standard two-step procedure:
> - (a) Extracting the **empirical lower envelope via binning**.
>   - Empirical selection of $\hat U_0$: According to Asp 1, belief deviation of the LLM agent will be further amplified once it progresses into an uncertain region. Hence we empirically select a proper value of $\hat U_0$ such that large belief deviations are observed.
>   - Determining $m_\theta$ and $c_0$: With $\hat U_0$, we partition the range of $\hat\Psi_t$ into $B$ equal-width bins $[\hat U_0,\psi_1),\ [\psi_1,\psi_2),\ \ldots,\ [\psi_{B-1},\Psi_\max).$ For each bin $b$, we collect all points $(\hat\Psi, \hat c_\theta)$ with $\hat\Psi\in[\psi_{b-1},\psi_b)$.  We then compute: the bin center $$x_b := \mathbb{E}[\hat\Psi \mid \hat\Psi\in\text{bin }b],$$ and the 10th percentile of the update error within the bin,
> $$y_b := \mathrm{Quantile}_{0.1}\big(\hat c\_\theta\big).$$
>
> - (b) **Linear estimation** of $\hat m_\theta, \hat c_0$ on the active region $\hat\Psi_t \ge \hat U_0$. We fit a linear model using the obtained $\lbrace (x_b,y_b) \rbrace$: $$y_b \approx \hat m_\theta x_b - \hat c_0.$$
>
> **Results.** The resulting visualization and linear estimation of 7B and 32B can be seen **in this figure** (https://postimg.cc/rD5kTZ5n).
>
> Together, these visual and quantitative results provide strong empirical support for Assumption 1: beyond moderate uncertainty, the belief-update error of LLMs grows at least linearly with the deviation of their belief state, matching the theoretical growth condition required for the existence of the Belief-Trap Region.

---

> ### Author Response · Authors · 2025-11-23
>
> ### 1.2 Other important assumptions
> - Non-degenerate observation (Assumption 3)
>
> In our finite-observation setting, this is a mild assumption: one may take $\eta = \min_{(o,s,a)} O(o\mid s,a) > 0$ over reachable triples.
>
> - Lipschitz policy (Assumption 4)
>
> The Lipschitz-policy assumption is widely used in RL theory analysis, where policies are often assumed or enforced to be Lipschitz continuous to obtain convergence and robustness guarantees (e.g., [1][2][3]). In parallel, recent work on Transformer-based language models explicitly studies and regularizes their Lipschitz constants and input-space smoothness (e.g.,[4][5][6]), indicating that the Lipschitz policy is a reasonable regularity assumption.
>
> - Value calibration (in Thm2)
>
> This type of assumption is prevalent in RL theory analysis. For example, [7] stated `Given a ρ-POMDP…, for any finite time horizon T, the optimal value function is (locally + vector) Lipschitz-continuous`. [8] proposed `Lipschitz Continuity of Q-Functions`. [9] stated `...prove that the value functions (over belief vectors) are piecewise linear and convex…`.
>
> In summary, we hope this clarification makes clear that Assumption 1 could be empirically verified and the other adopted assumptions are mild or prevalent in RL analysis; constants in the assumptions are not additional model hyperparameters, but serve as technical devices used to connect the formal theory to the belief-trapping behaviour qualitatively observed in prior work (e.g., [10]) and in our experiments.
>
> ### References in this part
>
> [1] Pirotta M, Restelli M, Bascetta L. Policy gradient in lipschitz markov decision processes. Machine Learning, 2015.
>
> [2] Wang, Yue, and Shaofeng Zou. "Policy gradient method for robust reinforcement learning." International conference on machine learning. PMLR, 2022.
>
> [3] Nie, Buqing, et al. "Improve robustness of reinforcement learning against observation perturbations via l∞ lipschitz policy networks." Proceedings of the AAAI Conference on Artificial Intelligence. Vol. 38. No. 13. 2024.
>
> [4] Castin, Valérie, Pierre Ablin, and Gabriel Peyré. "How Smooth Is Attention?." International Conference on Machine Learning. PMLR, 2024.
> [5] Ye, Wenqian, et al. "Mitigating transformer overconfidence via Lipschitz regularization." Uncertainty in Artificial Intelligence. PMLR, 2023.
>
> [6] Newhouse, Laker, et al. "Training Transformers with Enforced Lipschitz Constants." arXiv preprint arXiv:2507.13338 (2025).
>
> [7] Fehr, Mathieu, et al. "rho-POMDPs have Lipschitz-continuous epsilon-optimal value functions." Advances in neural information processing systems 31 (2018).
>
> [8] Lecarpentier, Erwan, et al. "Lipschitz lifelong reinforcement learning." Proceedings of the AAAI Conference on Artificial Intelligence. Vol. 35. No. 9. 2021.
>
> [9] Zhang, Hao. "Partially observable Markov decision processes: A geometric technique and analysis." Operations Research 58.1 (2010): 214-228.
>
> [10] Zhou, Zhanke, et al. "From Passive to Active Reasoning: Can Large Language Models Ask the Right Questions under Incomplete Information?." Forty-second International Conference on Machine Learning.

---

> ### Author Response · Authors · 2025-11-23
>
> ## 2. Direct test of Theorem 2
> We thank the reviewer for raising this important point. To directly examine when and how the credit-assignment pathology of Theorem 2 appears in reality, we conducted a controlled diagnostic experiment.
>
>
> **Experimental Setup.** We fixed a trained vanilla-PPO policy and generated two sets of rollouts: (i) standard rollouts without truncation, and (ii) rollouts using the proposed T3 rule (the truncation condition follows Sec. 3.1 of page 5). To isolate the effect of the uninformative tail, we selected failed trajectories (final reward = 0), and computed GAE for each token. This setup removes confounding effects of the success signals to the advantage analysis and allows a clean measurement of the uninformative tail’s influence on early-token advantages. For each early-stage token position $t=1:500$, we computed the average advantage across rollouts for both conditions.
>
> **Main results.** The main results are shown in **(a)(b) of this figure** (https://postimg.cc/1nwm8YZD).  Across both CD and MR datasets, we observe that without truncation, early-token advantages are systematically suppressed, showing the negative drift predicted by Theorem 2. Applying T3 consistently elevates these early advantages, indicating that truncation mitigates this drift and preserves the intended gradient signal.
>
> **Effect of Tail Length and Truncation Strength.** We further varied the effective tail length and truncation strength and presented the results in **(c)(d) of this figure** (https://postimg.cc/1nwm8YZD). Longer uninformative tails caused stronger suppression, whereas more aggressive truncation yielded less-biased early advantages, which fully aligned with the theory.
>
> We have made these empirical findings explicit in the revision.

---

> ### Author Response · Authors · 2025-11-23
>
> ## 3. T3 on PE/MR‑like tasks without access to the ground truth
>
> We thank the reviewer for pointing out the potential confusion. **We need to clarify that the T3 condition does not need to access the true preference vector $w^*$ in PE/MR.**
> Specifically,  we employ a **fully no-oracle truncation rule** derived from the agent’s own internal belief dynamics in PE/MR:
>
> $$stall_t=\left(\frac{1}{k} \sum_{j=t-k+1}^t\lVert \hat{w}_{j+1}-\hat{w}_j\rVert_2\right)<\varepsilon, $$
>
> where $\hat{w}_j$ is the predicted preference vector by the agent, $k$ is the sliding-window length with a little abuse of notation, and $\varepsilon$ is the truncation threshold. Specifically, we compute the $k$-step moving-average updates $\bar{\Delta}_t$​ over offline rollouts and set $\varepsilon$ to a **quantile** (e.g., 50%, 80%) of this distribution, **without using any oracle labels**. A trajectory is truncated when the agent’s preference estimate stops making meaningful changes for $k=4$ consecutive rounds, indicating epistemic stalling. This proxy depends only on the model’s belief updates and is completely ground-truth-free. The results of the PE dataset are shown in the table below, where we evaluated Qwen-7B-Instruct using PPO, and T3-gt represents the T3 reported in the paper.
>
> | Quantile   | 60%   | 75%   | 85%   | vanilla | T3-gt |
> |------------|-------|-------|-------|----------|-------|
> | $\varepsilon$| 0.18  | 0.28  | 0.36  | –        | –     |
> | Binary Sim | 44.33 | 50.67 | 49.00    | 42.00       | 49.00    |
>
> We sweep over $\varepsilon$ in 0.18, 0.28, 0.36, corresponding to 60%, 75%, 85% quantiles mentioned above.
> Using this belief-based truncation rule without access to ground truth, T3 still achieves strong gains on PE, and **confirms that T3 does not need to rely on the oracle vector**.
>
> ## 4. Exploration of adaptive T3 truncation rule.
>
> We thank the reviewer for providing novel insights for extension of T3 principles. The systematic adaptive T3 version is an important future work. Here we provide some preliminary exploration:
>
> **Experiments on adaptive T3.** Following the reviewer’s suggestion, we instantiate T3 on the PE dataset using the belief-based, no-ground-truth proxy, as discussed in WQ3: $$stall_t=\left(\frac{1}{k} \sum_{j=t-k+1}^t\lVert \hat{w}_{j+1}-\hat{w}_j\rVert_2\right)<\varepsilon, $$ where $\hat w_t$ is the agent’s internal preference estimate. We set the sliding window $k=4$, same as WQ3.
>
> **Adaptive selection of $\varepsilon$.** Instead of using a fixed $\varepsilon$, **we update it online in a data-driven manner**. Specifically, every 6 training steps, we sample a batch of fully untruncated rollouts from the current policy, and compute the empirical distribution of the $k$-step moving-average update magnitudes $\bar \Delta_t$. We then set $\varepsilon$ to a fixed quantile $\alpha$ (e.g., 60% or 80% quantile) of this distribution:
> $$ \varepsilon \leftarrow \text{Quantile}\_\alpha\left(\lbrace \bar\Delta_t \rbrace\_{\text{online}}\right). $$ This yields an adaptive T3 threshold that automatically adjusts to the model’s current update scale and training dynamics.
>
> **Outcome.** We sweep over the quantile $\alpha$ and present the results in the following table. The performance varies non-monotonically with the increase of $\alpha$. Notably, at $\alpha = 0.6$, **the model even substantially surpasses both our baseline (0.42) and the T3 result reported in the paper (0.49), highlighting the promise of the adaptive T3 variant.**
>
> | $\alpha$  | 20%   | 40%   | 60%   | 80%   | 90%   | vanilla | T3-gt |
> |------------|-------|-------|-------|-------|-------|---------|-------|
> | Binary Sim | 43.67 | 44.33 | **60.33** | 43.67 | 39.67 | 42.00      | 49    |
>
> These results indicate that the T3 principle is compatible with adaptive thresholding and remains effective without the access to the ground-truth. We have supplemented this important extension in the revised manuscript.

---

> ### Author Response · Authors · 2025-11-23
>
> ## 5. Reliance on an LLM judge in SP; Sensitivity to the SP judge identity/temperature
> We thank the reviewer for raising this point. While in SP we include a LLM-judge-based proxy (consecutive “unknown” responses made by the LLM judge), **our method does not need to rely on an LLM judge**. As described in Section 3.3.3, we also evaluate a judge-free proxy for T3 condition:
> - Sim-$\alpha$ (semantic-redundancy detection): Truncate when the cosine similarity between the embedding of the current query and any previous query exceeds a threshold $\alpha$. This proxy detects circular or redundant questioning and does not use the judge at all.
>
> - Importantly, Sim-$\alpha$ is relatively robust to the choice of $\alpha$: with $\alpha$ in $0.90, 0.93$, both variants yield strong improvements (as seen in Table 3), demonstrating that T3’s effectiveness does not depend on the identity or temperature of the LLM judge.
>
> We have clarified this potentially confusing point in the revision.
>
> ## 6. Concern of `Shorter is better confound`
>
> We need to clarify that as discussed in Sec. 2, T3 achieves better performance through mitigating the negative effects of BTR to credit assignment.
>
> Furthermore, our gains **cannot be attributed to trajectory shortening alone**. In the controlled study of Sec. 3.3.3 on the CD dataset, we ran random truncation ($\beta = 0.1$) and T3 truncation ($k = 3$). Although these two configurations exhibit **very similar initial averaged response-length** at the beginning of training, their training dynamics differ: 1) T3 achieves much higher final accuracy (77.83% vs. 69.00%; Table 3), and 2) over training, T3 achieves longer CoT length (See **in this figure:** https://postimg.cc/xccfwPx9). These observations indicate that T3’s improvement does not stem from producing shorter trajectories, but rather **from selectively removing BTR-induced tails that misallocate credit**, unlike random truncation that indiscriminately removes informative content.
>
> We have clarified this in the revision and add the above evidence showing that “shorter is better” alone does not account for T3’s effect.
>
>
> ## 7. Actionable Feedback Responses
>
> - Algorithm box
>
> Thank you for the suggestion. We have added a concise algorithm box summarizing the inputs, stall-detection trigger, and how truncation integrates into PPO/GAE.
>
> - `disentangle truncation vs. better credit assignment`
>
> As explained in the response to WQ6, we would respectfully clarify that T3 is explicitly designed as a credit-assignment correction mechanism: it removes belief-trap tails which contaminate early-token advantages under outcome-based RL. This point is highlighted in the Intro section of the paper (line 85, page2). Therefore, improving credit assignment is the intended mechanism of T3 rather than a confound.
>
> - `Equal-length ablations`
>
> Please refer to WQ6 of the Weaknesses & Questions Section discussed above.
>
> - γ/λ/KL sensitivity / Calibration check
>
> Our settings follow widely-used PPO/GRPO/GSPO configurations in LLM agentic reasoning, e.g., [11][12][13]. A full sensitivity study is valuable but beyond our current scope; we will add a short discussion on robustness.
>
> - Token/wall-clock to fixed reward thresholds.
>
> As seen in Sec. 3.3.1, we already reported some token-to-reward-threshold metrics: under PPO on CD, to reach a reward level of 0.65, our method consumes 66.4% of the total tokens compared to vanilla on average; under GSPO on GN, to reach 0.96, it requires 76.3% of the tokens.
>
> ### References in this part
>
> [11] Shi, Yaorui, et al. "Search and Refine During Think: Facilitating Knowledge Refinement for Improved Retrieval-Augmented Reasoning." The Thirty-ninth Annual Conference on Neural Information Processing Systems. 2025.
>
> [12] Jin, Bowen, et al. "Search-r1: Training llms to reason and leverage search engines with reinforcement learning." arXiv preprint arXiv:2503.09516 (2025).
>
> [13] Zheng, Chujie, et al. "Group sequence policy optimization." arXiv preprint arXiv:2507.18071 (2025).

---

> > ### Comment · Reviewer_K1QM · 2025-11-27
> >
> > After reading the rebuttal, I am increasing my score. The authors directly addressed the main weaknesses and my own confuision: they provided a clear empirical validation of Assumption 1 and justified the other assumptions as reasonable ones oftne made in the RL literature. They also included a controlled and convincing test of Theorem 2 showing tail-induced advantage drift and its correction by T3. The adaptive T3 variant is also very nice and interesting extension. They also clarified my misudnerstanding about the fact that there's no LLM-judge needed, and included an algorithm box plus token-efficiency evidence. With these concerns resolved, the paper’s soundness and contribution are stronger, and I update my rating to 8 (accept)

---

> > > ### Author Response · Authors · 2025-11-27
> > > **Thank you and a small note**
> > >
> > > Dear Reviewer K1QM,
> > >
> > > Thank you very much, Reviewer K1QM, for your insightful suggestions, thoughtful follow-up, and for letting us know that you are updating the rating! We truly appreciate the time and effort you spent reviewing our work!
> > >
> > > Just a small note in case it's helpful: on our side, the rating in the system may not have been updated to reflect your final assessment. If it's not too much trouble, would you mind double-checking it at your convenience? This would help ensure the records are consistent with your valuable conclusions.
> > >
> > > Thank you again for your constructive feedback, which helped a lot in strengthening our work!
> > >
> > > Best regards,
> > > Authors

---

> > > > ### Comment · Reviewer_K1QM · 2025-11-27
> > > >
> > > > Ooops! Apologies, done now!

---

> > > > > ### Author Response · Authors · 2025-11-27
> > > > >
> > > > > Thank you for the prompt follow-up! And once again, we are grateful for your time and continued engagement in the review process!
> > > > >
> > > > > Cheers,
> > > > > Authors

---

### Official Review · Reviewer_nLrS · 2025-11-02

**Soundness:** 2
**Presentation:** 2
**Contribution:** 1
**Rating:** 4
**Confidence:** 4

**Summary:**

The paper targets multi-turn active reasoning with LLM agents, where belief tracking often drifts, which produces redundant actions that poison RL credit assignment. It proposes T3 - early truncation of rollouts once belief-trap are detected via simple proxies (e.g., non-shrinking hypothesis sets, repeated “unknown” feedback, off-manifold guesses, or declining preference similarity). T3 is claimed to be a drop-in rule for PPO/GRPO/GSPO that preserves credit for informative prefixes and removes noisy tails. Across five tasks, it improves stability and OOD robustness, increasing performance gains with fewer rollout tokens.

**Strengths:**

the $T^3$ framework provides a useful lens to connect information gain with progress in active reasoning (though its usefulness is a bit questionable, see more below), and the proxy-based T3 is simple to implement and, empirically, shows consistent gains across tasks/algorithms

**Weaknesses:**

- while the BTR lens motivates truncation, Def. 2’s T3 condition does not give actionable guidance on setting $ k, \alpha, \beta $ beyond "detect stalling". The task-specific proxies in Sec. 3.1 are essentially heuristic and could be understood without Section 2. In other words, the T3 condition does not concretely guide how to pick the metric $d(\cdot,\cdot)$, the window size $k$, or the threshold $\Delta_{\min}$, nor does it yield principled settings for the concrete proxies ("unknown" counts in SP, hypothesis-set shrinkage in CD/GN, similarity trends in PE/MR).There is no quantitative link between $ \Delta \Psi$ and $ d (H_\tau, H_{\tau +1} ) $; can the authors articulate decision rules (e.g., choose $k$ to control an estimated upper bound on false truncations via concentration of an estimator of $\Delta \Psi$)?

- as to the two theorems established
  - assumption 1 appears a bit strong: it posits systematic error growth with uncertainty across all actions/observations, yet the experiments do not validate (or estimate) $m_\theta, c_0, U_0$.
   - the extra constants/conditions ($\eta$, $L_\pi$-Lipschitz policy, “non-degenerate observations”) appear task/model-dependent and unverified; the bound $U$ seems not computable from observable quantities.
  - theorem 2 motivates truncation but does not establish when actual policy gradients are meaningfully biased on the reported tasks.
  - the sufficient condition shows the possibility of a sign flip, but does not quantify when it occurs for real critics/discounts.

- as to training dynamics/stability: figures show improved stability in several cases, but GRPO+T3 still collapses later in SP. The rationale for pairing PPO (CD/PE), GRPO (SP), GSPO (GN) is not explained; differences in optimizer/entropy/clip may confound stability claims. If T3 is “algorithm-agnostic,” a matched comparison across all tasks/algorithms would strengthen its stabilization role.

- as to OOD analysis consistency, CD OOD uses Qwen-14B (Table 2) while the main CD uses 7B; can the authors justify the switch? The narrative “too many references induce redundancy $\Rightarrow$ more BTR” in PE is plausible but unsubstantiated; no proxy trajectory statistics (e.g., stall frequency, hypothesis-shrinkage rates) are shown to support the causal link.

- authors need to be explicit about what default $\alpha$ and $\beta$, $k$ values are used for all 5 reasoning tasks.

- regarding ratio of early-truncation vs. performance, the trends are confusing: authors report SP with $\alpha=0.9$, where the truncation ratio converges to 1 and performance increases, whereas PE with $\beta=0.8$ also $\rightarrow 1$ but performance decreases. Similarly, for CD, higher ratios ($k=1,2$) appear to hurt exploration, unlike SP. Is there actually a consistent correlation between performance and the truncation ratio? Readers need the rationale and a principled conclusion, not case-by-case explanations. A controlled sweep reporting correlation between ratio and performance, conditioned on proxy type, would clarify when "more truncation" helps vs. harms.

- in addition, regarding the summary claim in lines 418-420, can authors elaborate on this alignment: what theoretical quantities map to the chosen proxy thresholds and settings, and under what observable conditions (e.g., provable relation between $|H_t|$-shrinkage and $\Psi$-decrease in CD/GN)?

- some claims are casual: "This suggests that distillation can effectively boost the belief-tracking capability under finite state spaces"; that size/type differences “can be attributed to $m_\theta$" (lines 450-452) is speculative with current evidence. The first statement is based on one distilled LLaMA baseline on one task. It should be labelled as a hypothesis; broader study across tasks/models is needed.

- $m_\theta$ is introduced in Assumption 1 as a slope in the belief-update error lower bound, but later used to explain size/type effects qualitatively. can authors elaborate on it? How can we connect $m_\theta$ to measurable quantities (e.g., proxy stall rates) and estimate it empirically?

- minor presentation issues

  - authors need to list default ($k, \alpha, \beta$) per task and how you tuned them

  - define the “ratio of early truncation” in text (denominator, per-episode vs. per-step?)

  - in Fig. 3a,d the y-axis should read “Response length”

  - Table 1 should be placed near Sec. 3.3.1 analysis

**Questions:**

- is the ratio of early truncation ever defined/formalized in the text?
- what are variables $b, a, o$ in fig 1?
- How would section 2 choose ($k, \alpha, \beta$)? For instance, can we map $d(H_\tau, H_{\tau + 1})$ to an empirical estimator of $\Delta \Psi$ and pick $k$ so that the false-truncate probability is < δ (with a concentration bound)?

- Why does GRPO+T3 still collapse later in SP? Is this due to proxy false-truncation starving exploration, or value-function drift? Any mitigation (pro

- for binary similarity threshold 0.88 (PE): why 0.88?

---

> ### Author Response · Authors · 2025-11-23
>
> We thank reviewer nLrS for their time in providing insightful comments, and we will respond to their concerns as follows. (eg, W3 means the third Weakness)
>
> # Weaknesses
> ## W1. Concerns about T3 methodology
> ### 1.1 Concern that task-specific proxies are heuristic
>
> We appreciate the reviewer’s insightful comments. We agree that our current experiments instantiate different observable proxies for different tasks. Our intention in this paper is not to propose a single universal proxy, but a **general principle: T3 truncates when epistemic progress stalls**. For LLMs, the latent belief state is not directly accessible, and it is therefore impossible to pinpoint BTR entry exactly. The implementation of T3 accordingly relies on observable surrogates of epistemic progress, which naturally vary with the task’s available interaction signals rather than with the T3 principle itself.
>
> In addition, in the response to reviewer DC2y (https://openreview.net/forum?id=r8hzDA3pUY&noteId=Aa6nNGohNX), following the principle of T3, **we can further reduce the reliance on task-specific knowledge on hypothesis space by utilizing semantic redundancy signals** that the model repeatedly asks questions that are paraphrastic variants of earlier ones or revisits previously explored informational subgoals. **We have some successful explorations as follows:**
>
> - In the ablation studies in Sec. 3.3.3, we explored using Question Semantic Similarity as the truncation condition in the SP task, and the results also show improvements;
> - More interestingly, in the response to W3 and W4 of the reviewer K1QM (https://openreview.net/forum?id=r8hzDA3pUY&noteId=2UZZ2wTc0l), for tasks with continuous spaces like Preference Estimation, we experiment on leveraging redundancy-style signals: truncating when the preference vector estimated by the agent stops changing meaningfully; it reflects an epistemic “stall” analogous to semantic redundancy in the SP task. We also obtained a remarkable performance gain, **even largely surpassing the results reported in the original paper**.
>
> More specifically, T3 requires a representation of the hypothesis state ($\mathcal{H}$), a criterion for epistemic progress $d(\mathcal{H}\_\tau,\mathcal{H}\_{\tau+1})<\Delta_{\min}$, and a patience hyperparameter $k$ for truncation. In practice, since the structure of hypothesis spaces and notions of progress differ across tasks, obtaining these components naturally relies on task-level meta-knowledge about what observable signals best reflect changes in $\mathcal{H}$. From the five diverse tasks in this work, it could be relatively easy to find the surrogates for applying T3 and obtain significant improvement, **following the same principle of T3**.
>
> ### 1.2 T3 `does not concretely guide` $d(\cdot,\cdot), k, \Delta_\min$
> As discussed above, the T3 principle does not prescribe a unique choice of $k$, $d(\cdot,\cdot)$, or $\Delta_{\min}$ because these quantities necessarily depend on the task’s observable structure: different environments expose different forms of epistemic progress. T3 specifies what needs to be detected (stalling), but leaves the concrete instantiation to the task’s available signals. This is analogous to how advantage estimation or reward shaping depend on environment-specific observables while the underlying RL principle remains task-agnostic.
>
> ### 1.3 `no quantitative link between` $\Delta \Psi$ and $d(H_\tau, H_{\tau+1})$
> Please refer to W7 of the Weaknesses Section, where we give an explicit quantitative link between $\Delta \Psi$ and $d(H_\tau, H_{\tau+1})$ in tasks like CD/GN.
> ### 1.4 `...can the authors articulate decision rules…`
> Please refer to Q3 of the Questions Section, where we give a derivation leveraging concentration inequality, which controls an estimated upper bound on false truncations via $k$ and $\Delta_\min$ (both are T3 params from Def 2).

---

> ### Author Response · Authors · 2025-11-23
>
> ## W2.1 Assumption 1 not validated by experiments
> We appreciate this point. We need to note that a direct empirical validation is challenging because both the oracle Bayesian update $B^\star$ and the agent’s internal belief $b_t$ are not directly accessible in LLMs. To address this concern, we designed an experiment on the PE task that allows us to construct practical and theoretically aligned approximations of the quantities involved in Assumption 1.
>
> **i) Approximating the potential $\Psi$.** In PE, each interaction round provides the model’s explicit estimate of the latent preference vector, denoted by $w_t$. Since the ground-truth preference vector $w^\star$ is known, we define: $d(w_t) := \Vert w_t - w^\star\Vert_2^2,$
> and use $d(w_t)$ as an observable proxy for the task potential: $\hat \Psi_t:=d(w_t)\approx \Psi(b_t).$ This proxy retains the key properties of our potential function: it is non-negative, and equals zero iff. task completion.
>
> **ii) Approximating the oracle Bayesian update $B^\star$.** While the true Bayesian posterior over the latent user preference is inaccessible, we can construct a standard preference-update rule $\hat B$ based on traditional machine learning. Given the model’s query $a_t$ and the feedback $o_t$, this update produces:
> $$\hat{w}\_{t+1} := \hat{B}(w_t, a_t, o_t),$$
> which we use as a principled surrogate for the oracle update $B^\star$. More specifically, we leverage:
> $$\hat w_{t+1}= w_t + K_t m_t (o_t -m_t^\top w_t),
>  \qquad
>  K_t = \frac{\sigma_0^2}{\sigma_0^2 \Vert m_t\Vert_2^2 + \sigma^2}.$$
> Here, $m_t$ is the movie-attribute difference vector for the pair of movies selected by the LLM’s query, i.e., $m_t = \mathrm{attr}(A)-\mathrm{attr}(B)$. The binary observation $o_t \in \lbrace-1,+1\rbrace$ represents the user’s feedback to this query and is given by $o_t = \operatorname{sign}(m_t^\top w^\star)$. The parameters $\sigma_0^2$ and $\sigma^2$ denote the prior and observation noise variances, respectively. We set them to $1.0$.
>
> In contrast, the LLM’s own update under parameters $\theta$ yields:
> $w_{t+1} := B_\theta(w_t, a_t, o_t),$
> as the LLM-generated estimate at the next step.
>
>
> **iii) Constructing observable samples of the update-error quantity.** We empirically instantiate the update error term in Eq. 2 of the paper: $$\hat c\_\theta(b_t):= d(w\_{t+1}) - d(\hat{w}\_{t+1})\approx c_\theta (b_t).$$
>
> We collect over **150k** data points by rollouts generated by the Qwen-2.5 series models, obtaining samples of $\lbrace(\hat \Psi_t, \hat c_\theta(b_t) )\rbrace$. Plotting these pairs allows us to examine Asp1.
>
> **iv) Visualization and Estimation of $m_\theta, U_0, c_0$.**  While individual samples exhibit variance (as expected for high-dimensional belief updates), Asp 1 concerns only the lower bound of this relationship. Therefore, our goal is to estimate the lower-envelope trend of the obtained sample points. We follow a principled and standard two-step procedure:
> - (a) Extracting the **empirical lower envelope via binning**.
>   - Empirical selection of $\hat U_0$: According to Asp 1, belief deviation of the LLM agent will be further amplified once it progresses into an uncertain region. Hence we empirically select a proper value of $\hat U_0$ such that large belief deviations are observed.
>   - Determining $m_\theta$ and $c_0$: With $\hat U_0$, we partition the range of $\hat\Psi_t$ into $B$ equal-width bins $[\hat U_0,\psi_1),\ [\psi_1,\psi_2),\ \ldots,\ [\psi_{B-1},\Psi_\max).$ For each bin $b$, we collect all points $(\hat\Psi, \hat c_\theta)$ with $\hat\Psi\in[\psi_{b-1},\psi_b)$.  We then compute: the bin center $$x_b := \mathbb{E}[\hat\Psi \mid \hat\Psi\in\text{bin }b],$$ and the 10th percentile of the update error within the bin,
> $$y_b := \mathrm{Quantile}_{0.1}\big(\hat c\_\theta\big).$$
>
> - (b) **Linear estimation** of $\hat m_\theta, \hat c_0$ on the active region $\hat\Psi_t \ge \hat U_0$. We fit a linear model using the obtained $\lbrace (x_b,y_b) \rbrace$: $$y_b \approx \hat m_\theta x_b - \hat c_0.$$
>
> **Results.** The resulting visualization and linear estimation of 7B and 32B can be seen **in this figure** (https://postimg.cc/rD5kTZ5n).
>
> Together, these visual and quantitative results provide strong empirical support for Assumption 1: beyond moderate uncertainty, the belief-update error of LLMs grows at least linearly with the deviation of their belief state, matching the theoretical growth condition required for the existence of the Belief-Trap Region.

---

> ### Author Response · Authors · 2025-11-23
>
> ## W2.2 Extra constants or conditions (eg $L_\pi$ Lipschitz) unverified
> We appreciate the reviewer’s comment and agree that the constants $\eta$, $L_\pi$, and $U$ are task/model-dependent and not directly observable. Our intention is not to treat them as tunable or empirically estimated quantities, but to characterize the existence of the belief-trap region (BTR) for theoretical analysis. The implementation of the T3 method does not rely on those constants.
> - **Non-degenerate observations**
>
> In our finite-observation setting, this is a mild assumption: one may take $\eta = \min_{(o,s,a)} O(o\mid s,a) > 0$ over reachable triples.
> - **Lipschitz policy**
>
> The Lipschitz-policy assumption is widely used in RL theory analysis, where policies are often assumed or enforced to be Lipschitz continuous to obtain convergence and robustness guarantees (e.g., [1][2][3]). In parallel, recent work on Transformer-based language models explicitly studies and regularizes their Lipschitz constants and input-space smoothness (e.g.,[4][5][6]), indicating that the Lipschitz policy is a reasonable regularity assumption. In our theory, this assumption is to describe the LLM agent’s sensitivity of its action to its internal belief of the latent states, and never compute $L_\pi$ in practice.
> - **The bound $U$ in Theorem 1**
>
> Theorem 1 provides an existence result: under these regularity conditions and a growth assumption on belief-update error, there exists some level set $\{\Psi \ge U\}$ that behaves as a BTR. We agree that $U$ is not computable directly from observations in our current setup and it is also intractable for LLMs; we do not attempt to estimate it. Instead, our empirical analysis uses observable proxies (e.g., hypothesis-set shrinkage, similarity trends) to approximate entry into the theoretical BTR.
>
> We hope this clarification makes clear that these constants are not additional model hyperparameters, but serve as technical devices used to connect the formal theory to the belief-trapping behaviour qualitatively observed in prior work (e.g., [7]) and in our experiments.
>
> ### References in this part
> [1] Pirotta M, Restelli M, Bascetta L. Policy gradient in lipschitz markov decision processes. Machine Learning, 2015.
>
> [2] Wang, Yue, and Shaofeng Zou. "Policy gradient method for robust reinforcement learning." International conference on machine learning. PMLR, 2022.
>
> [3] Nie, Buqing, et al. "Improve robustness of reinforcement learning against observation perturbations via l∞ lipschitz policy networks." Proceedings of the AAAI Conference on Artificial Intelligence. Vol. 38. No. 13. 2024.
>
> [4] Castin, Valérie, Pierre Ablin, and Gabriel Peyré. "How Smooth Is Attention?." International Conference on Machine Learning. PMLR, 2024.
> [5] Ye, Wenqian, et al. "Mitigating transformer overconfidence via Lipschitz regularization." Uncertainty in Artificial Intelligence. PMLR, 2023.
>
> [6] Newhouse, Laker, et al. "Training Transformers with Enforced Lipschitz Constants." arXiv preprint arXiv:2507.13338 (2025).
>
> [7] Zhou, Zhanke, et al. "From Passive to Active Reasoning: Can Large Language Models Ask the Right Questions under Incomplete Information?." Forty-second International Conference on Machine Learning.

---

> ### Author Response · Authors · 2025-11-23
>
> ## W2.3 & 2.4 Theorem 2 not validated by experiments
> We thank the reviewer for raising this important point. First, in response to `...when actual policy gradients are meaningfully biased…` or `...quantify when it occurs…`, we note that the precise characterization in real environments depends on many factors, including the value-network approximation error, rollout horizon, discounting, and the task-specific epistemic dynamics, etc.. Since our focus is to demonstrate the existence of BTR and reveal its influence on policy training, a fine-grained characterization is beyond the scope of the work.
>
> To further evaluate the negative drift predicted by Theorem 2,  we conducted a controlled experiment to empirically verify the phenomenon predicted by Theorem 2 on our reported tasks, showing how the negative drift manifests in practice, and how T3 mitigates it:
>
> **Experimental Setup.** We fixed a trained vanilla-PPO policy and generated two sets of rollouts: (i) standard rollouts without truncation, and (ii) rollouts using the proposed T3 rule (the truncation condition follows Sec. 3.1 of page 5). To isolate the effect of the uninformative tail, we selected failed trajectories (final reward = 0), and computed GAE for each token. This setup removes confounding effects of the success signals to the advantage analysis and allows a clean measurement of the uninformative tail’s influence on early-token advantages. For each early-stage token position $t=1:500$, we computed the average advantage across rollouts for both conditions.
>
> **Main results.** The main results are shown in **(a)(b) of this figure** (https://postimg.cc/1nwm8YZD).  Across both CD and MR datasets, we observe that without truncation, early-token advantages are systematically suppressed, showing the negative drift predicted by Theorem 2. Applying T3 consistently elevates these early advantages, indicating that truncation mitigates this drift and preserves the intended gradient signal.
>
> **Effect of Tail Length and Truncation Strength.** We further varied the effective tail length and truncation strength and presented the results in **(c)(d) of this figure** (https://postimg.cc/1nwm8YZD). Longer uninformative tails caused stronger suppression, whereas more aggressive truncation yielded less-biased early advantages, which fully aligned with the theory.
>
> We have made these empirical findings explicit in the revision.
>
> ## W3 Concerns of training dynamics/stability
> - **GRPO+T3 still collapses later in SP.**
>
> We appreciate the reviewer’s careful observation. We have two potential explanations for why GRPO + T3 still shows some late-stage collapse in SP, despite the clear stability improvement over the baseline.
>
> First, several recent studies suggest that GRPO exhibits training instability and reward collapse after many training steps in outcome-based RL settings (e.g., [8][9]). This behavior has been observed without truncation mechanisms, indicating that part of the degradation may stem from GRPO instability rather than T3.
>
> Second, the **SP task itself** poses a unique challenge.
> SP rewards are continuous: partially correct hypotheses yield stable medium scores, while fully solving the puzzle requires capturing many fine-grained details. This reward structure tends to reinforce an early, overconfident but incorrect hypothesis, creating an epistemic bottleneck.
> T3 mitigates the credit contamination caused by late-stage uninformativeness, but SP may exhibit a different failure mode: an early epistemic stalling induced by medium-reward attractors.
>
> Thus, some late-stage degradation still occurs, which may be attributed to GRPO’s instability and SP’s unique task structure.
>
>
> - `The rationale for pairing PPO (CD/PE), GRPO (SP), GSPO (GN) is not explained … would strengthen its stabilization role.`
>
> We need to clarify that **all three RL algorithms (PPO, GRPO, GSPO) are evaluated on all five tasks, as shown in Table 1**. Each task, therefore, includes a matched comparison across algorithms, **under identical optimizer/entropy/clipping settings**, isolating the effect of T3 within each algorithm family. The only reason the training-curve figures display a single algorithm per task is due to space constraints. This is a visualization choice rather than an experimental restriction.
>
> Thus, T3 is indeed **algorithm-agnostic**, and our conclusions on stability are drawn **within-algorithm** comparisons (with vs. without T3), not across different RL algorithms.
>
>
> ### References in this part
>
> [8] Jin, Bowen, et al. "Search-r1: Training llms to reason and leverage search engines with reinforcement learning." arXiv preprint arXiv:2503.09516 (2025).
>
> [9] Liu, Zongkai, et al. "CPGD: Toward Stable Rule-based Reinforcement Learning for Language Models." arXiv preprint arXiv:2505.12504 (2025).

---

> ### Author Response · Authors · 2025-11-23
>
> ## W4 Concerns of OOD analysis
> - **CD OOD uses Qwen-14B (Table 2) while the main CD uses 7B**
>
> We have presented the results of 7B in CD OOD in the following table.
>
> |        | PPO vanilla | PPO w/ T3 |
> |-------------|-------------|-----------|
> | $S=10$      | 61.67       | 77.83     |
> | $S=15$      | 56.83       | 70.17     |
> | $S=20$      | 41.67       | 50        |
> | $S=25$      | 32.83       | 37.5      |
> | $S=30$      | 27.83       | 31.83     |
> | $C=2$       | 61.67       | 77.83     |
> | $C=3$       | 60.33       | 72.17     |
> | $C=4$       | 48.92       | 52.33     |
>
> - **Verification of  “too many references induce redundancy ⇒ more BTR” in PE**
>
> We thank the reviewer for pointing out the need for empirical support of the redundancy-induced BTR in PE. To directly examine this claim, we conducted an additional diagnostic experiment measuring the truncation ratio under different reference-set sizes $S\in\lbrace10,15,20,25,30\rbrace$ across the Qwen-2.5-Instruct model series. The truncation condition is the same as what is used in Sec.3.1. The results are as follows:
>
> | | 10    | 15    | 20    | 25    | 30    |
> |-------|-------|-------|-------|-------|-------|
> | 3B    | 41.67 | 39.67 | 46.67 | 44.33 | 50.00 |
> | 7B    | 50.67 | 53.67 | 54.00 | 56.67 | 56.67 |
> | 14B   | 23.33 | 30.33 | 27.00 | 33.00 | 33.33 |
> | 32B   | 38.00 | 39.67 | 39.33 | 50.33 | 46.33 |
>
>
> The results show a consistently increasing truncation ratio as $S$ grows from 10 to 30, indicating that larger reference contexts lead to more frequent epistemic stalling events detected by T3. This aligns with our explanation: excessively large reference sets introduce noisy or redundant comparisons, thereby elevating the likelihood of entering BTR.
>
> We have added this empirical result in the revised manuscript to support our claim.
>
>
> ## W5 Clarification for the default $\alpha, \beta, k$ values
>
> We need to clarify that **the parameters $\alpha$ and $\beta$, referenced in Sec. 3.3.3, are not part of the default T3 condition. They are used only in ablations** that explore alternative stalling proxies beyond our main definition. None of the five tasks use $\alpha$ and $\beta$ in the default T3 pipeline. The default $k$ values for each task are listed in Sec. 3.1 ($k$ is set to 1 for GN, 5 for SP, 3 for CD, 2 for PE & MR).
>
> We have revised the manuscript to avoid any confusion between the core parameters and the ablation-specific ones.

---

> ### Author Response · Authors · 2025-11-23
>
> ## W6 Concerns for the analysis of early-truncation ratio vs. performance
>
> We appreciate the reviewer’s comment. We provide clarifications and respond point-by-point below.
>
> ### 6.1 Clarification of roles of $\alpha$, $\beta$ truncation
> First, as mentioned in W5, we need to clarify that the parameters $\alpha$ and $\beta$, referenced in Sec. 3.3.3, are **not part of the default T3 condition**. In fact, they are separate hyperparameters to ablate different truncation conditions:
> - $\beta$ (random truncation) is introduced as an *adversarial* baseline. Its role is to illustrate that effective truncation requires meaningfully detecting the BTR entry, **rather than simply indiscriminate cutting**. Therefore, the behavior of $\beta$-based truncation should not be interpreted as representative of BTR-based truncation.
> - $\alpha$ (semantic-similarity proxy), in contrast, is a meaningful surrogate that approximates belief-update stalling. This is to demonstrate the **flexibility** of T3: any surrogate that **meaningfully captures belief stalling** can be plugged into the T3 rule and still produce gains.
> ### 6.2 Meaning of the truncation ratio
> In this work, the truncation ratio at training step $t$ is defined as:
> $$
> \text{ratio}_t =\frac{\text{ num of rollouts truncated at step } t}{\text{ total rollouts at step } t}
> $$
>
> This is a diagnostic quantity that dynamically evolves throughout training. It is indirectly influenced by the T3 parameters (e.g., window size $k$) but it is **not directly controlled in the experiments**. Therefore, in the current work, the truncation ratio should be interpreted as a **diagnostic measure, not as a hyperparameter**.
> Specifically, in the original paper (Sec.3.3.3), we sweep over the params of the BTR-detection truncation (the window size $k$ or the similarity threshold $\alpha$) and examine how different settings affect both performance and the resulting truncation-ratio dynamics.
>
> Our observation (as seen in Sec 3.3.3): for tasks with an unbounded latent state space such as SP and PE, effective settings tend to produce truncation ratios that rise quickly to a high level. In contrast, for tasks with a finite and enumerable latent space, such as CD, we find that effective settings tend to keep low-to-moderate truncation ratios during training, as overly aggressive truncation removes informative exploratory steps.
>
> Additionally, to explicitly control the ratio, one would need an **adaptive adjustment of T3**’s parameters during training. We have done **some successful explorations**, in response to W4 of the reviewer K1QM https://openreview.net/forum?id=r8hzDA3pUY&noteId=2UZZ2wTc0l.
>
> ### 6.3 ` Is there actually a consistent correlation between performance and the truncation ratio?`
> **Our answer is that such consistency cannot exist.** The BTR phenomenon itself is task- and model-dependent, e.g., the structure of the hypothesis space, the belief tracking capabilities of LLMs, etc. Therefore, the “optimal” truncation strength is essentially different across tasks, as stated in Sec.3.3.3. The correct perspective is not cross-task consistency, but: **Within each task, more accurate detection of BTR entry leads to better performance.**
>
> We have clarified these points in the revised manuscript.

---

> ### Author Response · Authors · 2025-11-23
>
> ## W7  Correlations between theoretical quantities with the proxy thresholds and settings
>
> We thank the reviewer for raising this important point. For tasks in CD or GN, since an explicit, finite, and enumerable candidate set $H_t$ (of possible circuits / numbers) is accessible, it’s natural and standard to model the agent’s internal belief as approximately uniform over this feasible set:
> $b_t(s) = {1}/{|H_t|}, s\in H_t.$ In other words,  the agent treats all remaining candidates symmetrically and only eliminates some of them when contradicted by the feedback. We also require the mild condition $s^* \in H_t$ for convenient analysis, since otherwise the truth-anchored potential $\Psi(b_t) = -\log b_t(s^*)$ diverges.
>
> Then, it follows immediately that $\Psi(b_t) = -\log b_t(s^\star) = \log |H_t|$. Thus the change in potential is $\Psi(b_{t+1}) - \Psi(b_t) = \log |H_{t+1}| - \log |H_t|.$ Therefore, hypothesis-space refinement is equivalent to progress in the $\Psi$-potential: $|H_{t+1}| < |H_t| \Longleftrightarrow\Psi(b_{t+1}) < \Psi(b_t).$ This establishes that in CD/GN the T3 is a provably exact surrogate of the underlying potential decrease.
>
> **Relation to Definition 2.** Our formalization in Definition 2 intentionally uses a task-agnostic metric $d(\mathcal{H}\_t, \mathcal{H}\_{t+1})$, to quantify incremental changes in the agent’s hypothesis space. This abstraction covers CD/GN cases while remaining applicable to continuous or implicit hypothesis spaces in other tasks. The threshold $\Delta{\min}>0$ further quantifies the notion of minimum informative progress. Meanwhile, the sliding window of length $k$ reflects the temporal nature of belief-trap regions (BTR): these regions are characterized not by a single noisy step but by persistent non-positive progress. Under these params, T3 provides a principled and robust observable condition of the theoretical BTR.
>
> ## W8.1 Concerns that some claims are casual
> We thank the reviewer for pointing this out. We agree that the current evidence on DeepSeek-Distill-LLaMA-8B does not fully justify a general conclusion. Our intention was to highlight an interesting hypothesis:  distilled capability from the frontier reasoning model may implicitly assist with belief-tracking capability under finite state spaces, thereby enhancing the utility of T3.
>
> In the refined manuscript, we have softened the statement. Additionally, we conducted preliminary experiments using  DeepSeek-Distill-Qwen-7B on the CD task without hyperparameter tuning. The results are given below:
>
> |              | w.o. training | PPO vanilla | PPO w. T3 |
> |--------------|---------|-------------|-----------|
> | EM   | 7.17    | 84.33       | 86.17     |
>
> Interestingly, Qwen-7B model with distilled capability can achieve much better performance than one without distilled reasoning capability (61.67 in the original paper). We believe it is an interesting future direction to conduct a fine-grained analysis on how RLVR, as well as distillation, will influence belief tracking capability.
>
> ## W8.2&W9 Elaborating on $m_\theta$
> We appreciate the point. This is related to empirical verification of Assumption 1. Please refer to W2.1.
>
> ## W10 minor presentation issues
>
> We thank the reviewer for these detailed corrections. For 1) default $\alpha,\beta,k$ per task, refer to W5 of the rebuttal. For 2), please refer to the second bullet of W6. For 2), 3) and 4), we have fixed them accordingly in the revised manuscript.

---

> ### Author Response · Authors · 2025-11-23
>
> # Questions
> ## Q1 Defining the ratio of early truncation
> Please refer to W6 of the weakness section.
>
> ## Q2 Variable meaning in Fig 1
> $b,a,o$ refer to the belief, action, and the observation,respectively. We have clarified them accordingly in the revised manuscript.
>
> ## Q3
> As discussed before, we target a general principle: T3 truncates when epistemic progress stalls. In LLM active reasoning, since 1) it is intractable to inspect LLMs’ belief states and closely examine the exact BTR entry, and 2) BTR is task-specific, we rely on observable surrogates of epistemic progress, and therefore, **task-specific knowledge** is required to obtain those observable surrogates. Details could be found in our response to your previous concern (W1): https://openreview.net/forum?id=r8hzDA3pUY&noteId=dxiGNB46Bi.
>
> We also thank the reviewer’s advice and agree that concentration inequalities offer a useful tool for analytically linking the false-truncate probability with the T3 parameters. Below, we outline how this connection arises under a simple and standard noise model. We denote the true absolute progress at step $t$ as $g_t := \Psi(b_t) - \Psi(b_{t+1}),$ so that in non-BTR regions the update is consistently informative, i.e., $g_t \ge \rho > 0.$ for some $\rho>0$. The T3 proxy $d_t:=d(\mathcal H\_t,\mathcal H\_{t+1})$ could be viewed as a noisy estimator of this progress: $d_t = g_t + \xi_t,$ with $\xi_t \sim \text{Gaussian}(0,\sigma^2).$ According to Definition 2 of the paper, T3 is triggered at turn $t$ if $d_t < \Delta_{\min}.$ Therefore, in a non-BTR region, a false truncation occurs when $$d_t < \Delta_{\min} \Longleftrightarrow g_t + \xi_t < \Delta_{\min} \Longleftrightarrow \xi_t < \Delta_{\min} - g_t.$$ Since $g_t \ge \rho$, we upper-bound the event by $$\Pr(\text{false truncate at } t)\le \Pr(\xi_t < \Delta_{\min} - \rho).$$ If $\Delta_{\min} < \rho$, we can leverage concentration inequality: $$\Pr(\xi_t < \Delta_{\min} - \rho) \le \exp\left(-\frac{(\rho - \Delta_{\min})^2}{2\sigma^2}\right).$$
>
> Under approximate independence across turns (an assumption for simplicity), the probability that a length-$k$ window is entirely misclassified as stalled is at most $$\exp\left( -\frac{k(\rho - \Delta_{\min})^2}{2\sigma^2} \right).$$
> Thus, to guarantee that the false-truncate probability over a window is below $\delta$, we have relationships between $k, \Delta_{\min}$, and $\delta$ as: $k (\rho - \Delta_{\min})^2 \ge 2\sigma^2 \log(1/\delta).$
>
> ## Q4 Why does GRPO+T3 still collapse later in SP?
> Please refer to W3 of the Weakness section.
>
> ## Q5 Binary similarity threshold in PE: why 0.88
> We thank the reviewer for the question. For the PE task, the reward signal is based on the cosine similarity between the LLM-estimated preference vector and the ground-truth. We selected 0.88 randomly.
> To examine the sensitivity to this parameter, we evaluated thresholds $\lbrace0.85, 0.90, 0.95\rbrace$ and presented the results below (Qwen-7B-Instruct, PPO). Lower thresholds (e.g., <0.8) cause the binary reward to activate almost constantly, making the signal overly dense. Conversely, higher thresholds (e.g., >0.95) make activations extremely rare, resulting in an overly sparse reward that prevents PPO from learning effectively. Mid-range thresholds (0.85-0.90) consistently produced stable improvements, and 0.88 lies in this robust region. We have clarified this rationale in the revision.
>
> |              | 0.85 | 0.88 | 0.9  | 0.95 |
> |--------------|------|------|------|------|
> | PPO vanilla  | 55.33 | 42.00    | 33.67 | 4.33 |
> | PPO w. T3    | 63.00    | 49.00    | 37.67 | 3.67 |

---

> ### Author Response · Authors · 2025-11-27
> **A gentle reminder**
>
> Dear Reviewer nLrS,
>
> Thank you again for your thorough and constructive comments on our work! We have provided point-to-point responses to your questions and concerns. We understand that it may take some time to read through our responses. But as the discussion period has only one week left, to facilitate us better address your remaining concerns, could you kindly take a look at our responses below and let us know if you still have any questions? Thank you!
>
> Best regards,
> Authors

---

### Author Response · Authors · 2025-11-23
**Rebuttal Summary**

Dear reviewers,

Many thanks for your time and efforts in providing us with valuable feedback to improve our paper. We are grateful that the reviewers think this work offers a clear theoretical understanding of why RL struggles in long-horizon active reasoning, and proposes a simple yet broadly effective truncation mechanism with consistent empirical gains across benchmarks and RL algorithms. In addition, the reviewers raise many insightful questions and suggestions, and we will address them in our response to each reviewer.

Submission 8182 Authors

---

### Comment · Reviewer_4y4o · 2025-11-27
**I have receieve author's comment.**

My questions are well sovled.

---

> ### Author Response · Authors · 2025-11-27
> **Thank you!**
>
> Dear Reviewer 4y4o,
>
> Thank you so much for your acknowledgement and continued positive recommendation of our work!
>
> Best regards,
> Authors

---

### Author Response · Authors · 2025-12-04
**Rebuttal Summary for AC (Part 1 of 2)**

We thank the reviewers for their constructive comments and suggestions. We also appreciate the substantial load ACs are carrying this year. Below we provide a summary of the key points raised by reviewers, our responses, and our discussions with the reviewers.

We first summarize the **core contribution and novelty** of our work:
- We provide the first clear, unified theoretical explanation `(as recognized by Reviewers K1QM, DC2y, 4y4o)` of why outcome-based RL becomes systematically suboptimal in long-horizon active-reasoning tasks (with empirical verification), identifying a fundamental credit-assignment pathology (Thm. 2) caused by uninformative tails in Belief-Trap Region (Thm. 1 and Def. 1).
- Based on this insight, we introduce T3, a **simple yet general truncation principle** `(also agreed by all reviewers)` that triggers when epistemic progress stalls. T3 is not only theoretically grounded but also empirically effective across tasks, model scales, and RL algorithms `(as acknowledged by all reviewers)`.
- Together, our work establishes a **theory-practice** framework that diagnoses a ubiquitous phenomenon in active-reasoning RL and delivers a broadly applicable solution.

The reviewers have recognized the strength of this paper as **novel, clear theoretical understanding** of why RL struggles in long-horizon active reasoning, and that it proposes a **simple yet broadly effective** truncation mechanism with **consistent** empirical gains across benchmarks and RL algorithms.


## Reviewers’ Acknowledgement during the Discussion Period

This paper received an initial average score of 6.5 (4, 6, 8, 8). **Three reviewers acknowledged that their concerns have been addressed by our rebuttal and updated their scores to (8, 8, 8), while Reviewer nLrS who gave an initial score of 4 hasn’t responded to our rebuttal before responses was disabled due to the OpenReview bug.**

Specifically, two reviewers replied **before the bug was reported (see response history):**

-  Reviewer K1QM **(score 6 → 8)** stated that the rebuttal `directly addressed the main weaknesses and my confusion`… `With these concerns resolved, the paper’s soundness and contribution are stronger, and I update my rating to 8 (accept)` at Nov 27, 5:22 AM, EST (https://openreview.net/forum?id=r8hzDA3pUY&noteId=G6jrpcHznE).

- Reviewer DC2y **(score 8 → 8)** confirmed the original rating (8) and noted satisfaction with the additional clarification (https://openreview.net/forum?id=r8hzDA3pUY&noteId=5pn4syHyTy).

Reviewer 4y4o **(score 8 → 8)** replied to us during the “bug period”, confirmed the original rating (8), and noted `My questions are well solved` (https://openreview.net/forum?id=r8hzDA3pUY&noteId=WoZCEC8ERU).

Reviewer nLrS hasn’t responded before the disabling of the comment function, but the major concerns of Reviewer nLrS, including `empirical verifications of assumption 1 and theorem 2`, `generality of the truncation conditions`, and `analysis of false-positive truncation` are also shared by other Reviewers such as K1QM and 4y4o, who acknowledged that these concerns have been resolved by our rebuttal. Regarding the remaining questions of Reviewer nLrS, we also provided direct responses including theoretical analysis and empirical results, and we are confident that the reviewer’s remaining concerns are also addressed.

---

> ### Author Response · Authors · 2025-12-04
> **Rebuttal Summary for AC (Part 2 of 2)**
>
> ## Summary of Responses and New Results
>
> While we provided point-to-point responses to all reviewers, below we summarize common questions raised by reviewers as well as our responses.
>
> ### Summary of New Empirical Results
>
>
> #### 1. Proxy-design concerns (all reviewers)
>
> Multiple reviewers (nLrS, K1QM, DC2y, 4y4o) raised the question of whether T3 relies on task-specific heuristics. In response to them, we first clarified as follows: Our intention in this paper is not to propose a single universal proxy, but a **general principle: T3 truncates when epistemic progress stalls**. In practice, since the structure of hypothesis spaces and notions of progress differ across tasks, instantiating T3 naturally relies on **task-level meta-knowledge**.
>
> Moreover, we provided **empirical explorations on general-purpose detection** manners, which would further reduce reliance on task-specific knowledge. See Appendix E.1 of the paper, or Section 1.2 of the rebuttal to DC2y.
>
> `Reviewer Reactions`: We appreciate that Reviewers K1QM, DC2y, and 4y4o are satisfied with our responses and additional results, and have therefore resolved the concern.
>
> #### 2. Validation of core assumption and theory (nLrS, K1QM)
>
> We provided empirical tests of Assumption 1 (update-error behavior) and Theorem 2 (tail-induced advantage drift), in response to reviewer nLrS (bullet 2 of Weakness) and K1QM (`Theory-practice gap`, `Direct test of Theorem 2`). Please kindly refer to results in Appendix C.1 & C.2 in the revised paper, or  W2.1 & W2.3 of the rebuttal to nLrS, or Sec 1.1 & Sec 2 of the rebuttal to K1QM.
>
> `Reviewer Reactions`: We are grateful  that Reviewer K1QM kindly responded that `directly addressed the main weaknesses`.
>
>
> #### 3. Use of oracle in PE/MR tasks (K1QM)
>
> We added experiments showing that T3 on PE/MR-like tasks preserves its effectiveness **without access to ground-truth**, in response to reviewer K1QM (`PE/MR use oracle preference similarity to trigger truncation`).
>
> See Section 3 of the rebuttal to K1QM or Appendix D.3 of the paper.
>
> `Reviewer Reactions`: We are grateful that Reviewer K1QM kindly responded that `directly addressed the main weaknesses`.
>
> #### 4. Adaptive T3 (K1QM)
>
> We have made successful explorations on adaptive T3, which adjusts T3 parameters online, in response to reviewer K1QM (`no adaptive rule is provided`).
> See Section 4 of the rebuttal to K1QM or Appendix D.4 of the paper.
>
> `Reviewer Reactions`: We are grateful that Reviewer K1QM kindly responded that `The adaptive T3 variant is also very nice and interesting extension.`
>
> #### 5. Analysis on false-positive truncation (4y4o)
>
> We provided empirical analysis on false positive truncation and its impact, in response to reviewer 4y4o (`Lack of analysis on false positive truncation`). See Section 3 of the rebuttal to 4y4o or Appendix C.3 of the paper.
>
> `Reviewer Reactions`: We are grateful that Reviewer 4y4o responded that `My questions are well sovled.`
>
>
> #### 6. Additional analyses requested by nLrS
>
> - Results of OOD performance for the 7B model in the CD task, in response to bullet 4 of Weakness. See W4 of the rebuttal to nLrS.
>
> - Results of empirical verification of “too many references induce redundancy ⇒ more BTR” in PE, in response to bullet 4 of Weakness. See W4 of the rebuttal to nLrS or Appendix D.2 of the revised paper.
>
> - Experiments showing the rationale of selecting 0.88 as the threshold in the PE task, in response to bullet 5 of Question. See Q5 of the rebuttal to nLrS or Appendix D.1.
>
> ### Summary of New Theoretical Results
>
> #### 1. The constants in assumptions not measurable (nLrS, K1QM)
>
> - For the core Assumption 1, we provided detailed empirical tests and practical estimates of the relevant quantities (Appendix C.1; Rebuttal W2.1 to nLrS).
>
> - For the remaining assumptions, we added clarifications and supporting references, noting that (i) the constants appear as standard technical devices required to connect the formal theory to the observed belief-trapping behavior, and (ii) these assumptions are mild and commonly adopted in RL theoretical analyses. (Rebuttal W2.2 to nLrS and Section 1.2 to K1QM)
>
> `Reviewer Reactions`: We are grateful that Reviewer K1QM **explicitly acknowledged this**, stating that the rebuttal provided `a clear empirical validation of Assumption 1 and justified the other assumptions as reasonable ones oftne made in the RL literature`.
>
>
> #### 2. Relations between T3 and the theory (nLrS, 4y4o)
>
> We provide **theoretical relations between T3 and BTR-formalism**, characterizing how adjusting T3 components directly controls the probability of false positives with an exponential rate,  in response to reviewer nLrS (bullets 1, 7 of Weakness and bullet 3 of Question).
>
> See Proposition 1 (page 5) of the revised paper or W1, W7 and Q3 of the rebuttal to nLrS.

---

### Meta-Review · Area_Chair_REhb · 2026-01-06

**Summary:**

The paper presents a timely and significant contribution to active reasoning in LLMs, introducing a theoretically grounded method ($T^3$) to mitigate "stalling" during Chain-of-Thought generation. The submission garnered strong support from the majority of reviewers, who praised the authors’ comprehensive rebuttal for effectively bridging the "theory-practice gap" through new empirical validations of the belief deviation assumptions and the successful integration of non-oracle proxies. While the reliance on task-specific heuristics for proxy design remains a valid limitation noted by a few reviewers, the consensus is that the method's robust performance, theoretical novelty, and direct applicability to the critical problem of efficient inference-time compute firmly justify acceptance. Given the high reviewer enthusiasm and the relevance of the topic, I would like to recommend this paper for accpetance.

**Reviewer Concerns:**

**Resolved concerns**

- Reviewers nLrS and K1QM questioned the validity of Assumption 1 and Theorem 2. The authors added Appendix C.1 and C.2, providing direct empirical tests measuring update errors and advantage drift.

- Reviewer K1QM noted that the truncation proxies for PE and MR tasks relied on ground-truth (oracle) preference vectors, which would not be available in real deployment. The authors introduced experiments using non-oracle proxies (e.g., entropy, disagreement, self-consistency) in Appendix D.3, demonstrating that $T^3$ remains effective without ground-truth access.

- Reviewers nLrS and 4y4o were concerned about the risk and impact of truncating useful trajectories (false positives). The authors provided a detailed analysis in Appendix C.3 and derived Proposition 1, which theoretically links $T^3$ parameters to the probability of false positives

- Reviewer K1QM requested an adaptive version of T3 to auto-tune window sizes and thresholds. The authors presented an Adaptive T3 exploration in Appendix D.4.

- Reviewer nLrS questioned the method's performance on OOD data and smaller models. The authors provided additional results showing OOD robustness and performance gains for 7B models.

**Partially addressed concerns**

- Reliance on task-specific heuristics (proxy design). The authors clarify that their intention in this paper is not to propose a single universal proxy, but a general principle. While reviewers accepted this defense, the fundamental requirement to engineer task-specific proxies remains.

**Reviewer Scores:**

Reviewer K1QM has replied and increased the score.
Reviewer nLrS is likely to increase the score as the authors have explicitly addressed most of the concerns.
Other two reviewers are likely to maintian their scores.

---

### Decision · Program_Chairs · 2026-01-26

Accept (Oral)